# Reduced temporal and spatial stability of neural activity patterns predict cognitive control deficits in children with ADHD

Zhiyao Gao [1] ✉, Katherine Duberg[1], Stacie L. Warren [2], Li Zheng [3], Stephen P. Hinshaw[4,5], Vinod Menon [1,6,7,8] ✉ & Weidong Cai [1,6,8] ✉

This study investigates the neural underpinnings of cognitive control deficits in attention-deficit/hyperactivity disorder (ADHD), focusing on trial-level variability of neural coding. Using fMRI, we apply a computational approach to single-trial neural decoding on a cued stop-signal task, probing proactive and reactive control within the dual control model. Reactive control involves suppressing an automatic response when interference is detected, and proactive control involves implementing preparatory strategies based on prior information. In contrast to typically developing children (TD), children with ADHD show disrupted neural coding during both proactive and reactive control, characterized by increased temporal variability and diminished spatial stability in neural responses in salience and frontal-parietal network regions. This variability correlates with fluctuating task performance and ADHD symptoms. Additionally, children with ADHD exhibit more heterogeneous neural response patterns across individuals compared to TD children. Our findings underscore the significance of modeling trial-wise neural variability in understanding cognitive control deficits in ADHD.

Childhood attention deficit/hyperactivity disorder (ADHD) is one of the most prevalent neurodevelopmental disorders, affecting 5–8% of children worldwide[1,2]. Hallmarks of ADHD include hyperactivity, impulsivity, and deficits in sustaining attention and behavioral control[3,4]. Inhibitory dysregulation, the impaired ability to suppress context-inappropriate responses, is hypothesized to be one of the core mechanisms underlying these behavioral phenotypes[3,5–9]. Theories posit that temporal fluctuations in inhibitory control processes are central to inhibitory dysregulation in ADHD, although a crucial yet overlooked aspect is the stability and variability of neural coding during cognitive control[10]. Distinguishing temporal fluctuations in brain responses and associated cognitive processes with underlying inhibitory control is critical for elucidating neural mechanisms and

pathways characterizing the disorder's core phenotypic features. Yet, most prior studies have largely focused on average neural responses, overlooking spatiotemporal dynamics that may provide more sensitive biological signatures of ADHD[11,12].

Inhibitory regulation involves two distinct processes–proactive and reactive[13,14]. Proactive control refers to the preparation of strategic responses in advance of actions, when increased control is anticipated; whereas reactive control involves inhibiting prepotent responses when interference occurs[13–16]. Elucidating the precise nature of dysfunction associated with proactive and reactive processes is crucial for models of cognitive control deficits in ADHD pathology. However, few studies have systematically examined the neural mechanisms underlying temporal fluctuations in proactive and reactive control and their

[1]Department of Psychiatry & Behavioral Sciences, Stanford University School of Medicine, Stanford, CA, USA. [2]Department of Psychology, University of Texas, Dallas, TX, USA. [3]Department of Psychology, University of Arizona, Tucson, AZ, USA. [4]Department of Psychology, University of California, Berkeley, CA, USA. [5]Department of Psychiatry & Behavioral Sciences, University of California, San Francisco, CA, USA. [6]Wu Tsai Neuroscience Institute, Stanford University, Stanford, CA, USA. [7]Department of Neurology & Neurological Sciences, Stanford University School of Medicine, Stanford, CA, USA. [8]Maternal & Child Health Research Institute, Stanford, CA, USA. ✉e-mail: zygao@stanford.edu; menon@stanford.edu; wdcai@stanford.edu

relation to behavioral fluctuations and clinical symptoms in children with ADHD. Understanding these distinct control mechanisms and their dynamic neural representations is necessary for tailoring interventions that address specific deficits, potentially improving treatment outcomes by targeting underlying cognitive processes more accurately.

Indeed, deficits in these separate but interacting processes may underlie distinct cognitive and behavioral symptoms of ADHD[6]. For example, impaired proactive control may contribute to problems with sustained attention, yet reactive control deficits may drive increased impulsivity[14,17]. Disentangling the independent contributions of proactive and reactive control dysfunction to the clinical profile of ADHD will enable an understanding of underlying neural mechanisms and the development of tailored interventions. Furthermore, exploring both processes should provide insight into whether ADHD involves an overall inhibitory control impairment or selective deficits in reactive stopping versus proactive preparation[6].

Reactive control has been extensively investigated using the Stop-Signal Task (SST), in which participants make responses to frequent go signals but must withhold their responses when infrequent stop signals occur[18]. The stop-signal reaction time (SSRT), which estimates how fast one can cancel a prepotent response, has been widely used as an index of reactive control[18,19], characterizing inhibitory control deficits in clinical populations[5,20]. Reactive control engages a widely distributed set of cortical regions, including the anterior insula (AI) and pre-supplementary motor cortex (preSMA) nodes of the salience network (SN), and the dorsolateral PFC (dlPFC) and posterior parietal cortex nodes of the frontoparietal network (FPN)[21–29]. Previous studies have shown that the AI is involved in the initiation of control processes and the preSMA in maintaining and adjusting these processes over time[30–33]. Disturbance or dysfunction in these regions has been linked to difficulties in maintaining sustained attention and regulating responses[34–38]. The dlPFC and PPC are key components of the frontoparietal network, which supports executive functions such as maintenance and manipulation of information in working memory and task switching[39–44]. These regions have been consistently implicated in studies of inhibitory control[21,22,30]. The dlPFC is particularly important for implementing control strategies, while the PPC is crucial for integrating sensory and motor information to guide behavior[39,45]. However, few studies have examined the temporal variability in neural coding and the spatial stability of neural representations, which limits our understanding of how ADHD impacts the brain's ability to consistently engage in reactive control.

These challenges are further compounded when considering proactive control, an area where even less is known about the underlying neural mechanisms and their disruption in ADHD. Proactive control typically involves scenarios in which participants receive prior cues about impending tasks requiring heightened cognitive control. This approach assesses the ability to strategize and prepare for control demands in advance[16,46–51]. Research reveals that individuals showing greater degrees of proactive control (or response slowing) also exhibit better reactive control (i.e., faster SSRT), and this is demonstrated in both adults[46] and children[6]. These findings align with prior investigations demonstrating that proactive and reactive inhibitory control engage overlapping brain areas in the SN and FPN[46–49,52–54]. Despite the recognized importance of distinguishing between proactive and reactive control in ADHD, no research has yet systematically investigated if these control processes share common neural substrates or how their potential neural coding disruptions manifest within ADHD.

As noted, a critical unexplored aspect of ADHD is neural variability and its relation to fluctuations in behavior. A prominent feature of ADHD is high intra-individual variability in behavior and symptom profiles over time[10]. Previous behavioral studies of ADHD reveal that high intra-individual response variability (IIRV) is a consistent behavioral phenotype of ADHD[5,12,55,56]. Attentional lapses are attributed to elevated IIRV[57,58], although empirical evidence demonstrating strong associations between IIRV and clinical measures of inattention is still lacking. Examining stability versus variability of task-evoked responses at the single-trial level can provide a window into the neural dynamics that may directly relate to behavioral regulation. For example, trial-level instability in activation patterns within prefrontal regions critical for cognitive control could manifest as intermittent attentional lapses or impulse control failures. Capturing moment-to-moment fluctuations in brain function has the potential to shed light on the neurobiological bases of the heterogeneity and inconsistency of ADHD symptoms.

Here, we address crucial knowledge gaps in our understanding of the neural mechanisms of proactive and reactive control through an innovative experimental design and single-trial analysis tailored for disentangling neural coding of proactive vs. reactive control. We used a cued stop-signal task (CSST) (Fig. 1a), which allows us to directly probe neural responses associated with proactive and reactive control (Fig. 1b). Leveraging ultra-fast fMRI, we model trial-evoked responses to assess variability over time and similarity of spatial patterns across trials using representational similarity analysis (RSA) (Fig. 1c–e). RSA captures fine-grained patterns, revealing whether brain regions encode information similarly across task events[59]. This approach goes beyond previous research that has relied on average activation levels, enabling us to capture temporal variations in neural representations across trials. Examining inter-trial variability and consistency of neural coding allows us to relate neural instability directly to the heterogeneity and fluctuation of behavioral regulation in children with ADHD. This experimental and analytic approach elucidates whether and how neural network dynamics differ in children with ADHD versus TD children.

Our study adopts a comprehensive approach to investigate neural instability in ADHD. We begin with a whole-brain analysis of temporal variability and spatial stability of trial-evoked responses during proactive and reactive control. This allowed us to identify regions across the entire brain that show atypical neural dynamics in ADHD. Following this, we narrow our focus to specifically examine neural instability within key regions of the cognitive control system. This targeted analysis enables us to investigate how neural instability in these critical areas relates to behavioral instability and core ADHD symptoms.

We hypothesized that children with ADHD would show increased temporal variability and reduced spatial stability of trial-evoked responses in the AI and preSMA, which are the key nodes of the SN, as well as dlPFC and PPC, which are the key nodes of the FPN; weakened association between neural coding of inhibitory control and fluctuating behavior; decreased within-group similarity of trial-evoked responses in the SN and FPN; and temporal and spatial variability of trial-evoked response in association with clinical symptoms. Our findings of neural instability advance models of inhibitory control dysfunction by providing insights into the neural mechanisms that instantiate the heterogeneous behavioral symptoms of ADHD. More generally, advanced neuroimaging techniques allowed us to move beyond traditional methods that average neural activity over trials, providing a more detailed and accurate picture of how neural variability contributes to cognitive control deficits in ADHD.

## Results

### Weak and unstable dual control in children with ADHD

We recruited 107 children (9–12 years old) from the local community, comprising 53 children with ADHD and 40 TD children, who all completed two runs of the CSST. The final fMRI sample included 26 children with ADHD (10.8 ± 1.2 years old, 10 F/16 M) and 35 TD children (10.5 ± 1.1 years old, 13 F/22 M) who met task performance and head motion criteria (see the Method section for details). There were no significant differences in age or gender between the two groups (all

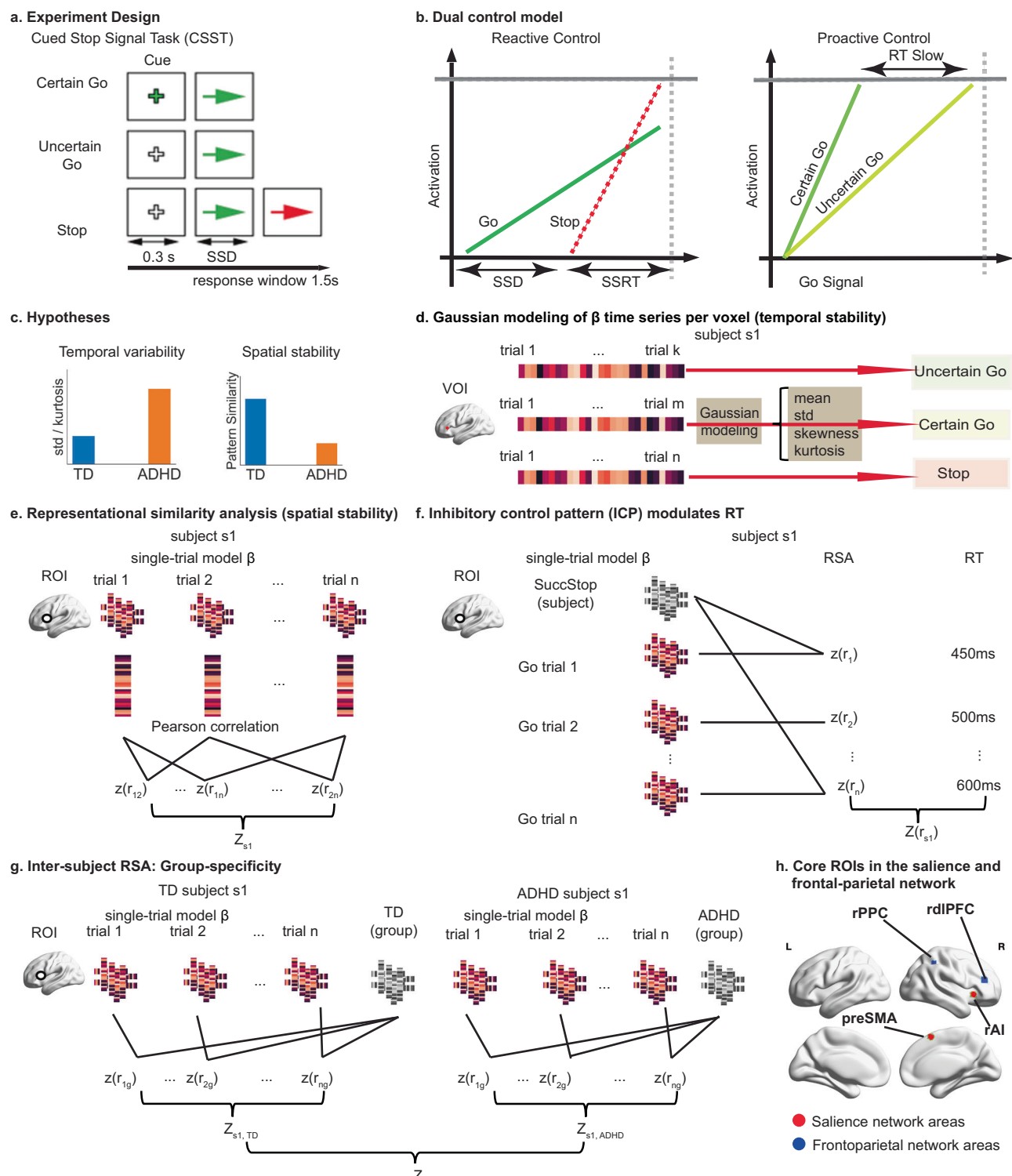

**Fig. 1 | Experimental design, analysis pipeline, and hypotheses. a** CSST paradigm involved Certain Go, Uncertain Go, and Stop trials. **b** Dual control model decomposed reactive control and proactive control processes in the CSST. **c** We hypothesized that children with ADHD will show higher temporal variability and lower spatial stability of trial-evoked brain responses than TD children. **d** The Gaussian model was applied to assess the temporal variability of trial-evoked neural responses. **e** Representational similarity analysis (RSA) was used to examine the spatial stability of trial-evoked activation patterns. **f** ICP index was computed to measure the extent to which proactive control was implemented in Certain and Uncertain Go trials, which modulates trial-wise behavioral fluctuation. **g** Inter-

subject pattern similarity analysis was used to investigate the group-specificity of neural coding during proactive and reactive control. **h** Core regions of interests (ROIs) in the salience (red color) and frontoparietal (blue color) network, defined from an independent meta-analysis study[30]. CSST stop-signal task, SSD stop-signal delay, TD typically developing, ADHD attention-deficit/hyperactivity disorder, SSRT stop-signal reaction time, ICP inhibitory control pattern, SuccStop Successful Stop trials, rAI right anterior insula, preSMA presupplementary motor cortex, rdlPFC right dorsal lateral prefrontal cortex, rPPC right posterior parietal cortex, RT reaction time, RSA representational similarity analysis.

**Table 1 | Demographics and behavioral results (fMRI sample)**

| Demographics | TD controls (*N* = 35) | ADHD (*N* = 26) | *p* value |
|---|---|---|---|
| Age (SD) | 10.457 (1.07) | 10.808 (1.20) | 0.234 |
| Gender (female/male) | 13/22 | 10/16 | 0.916 |
| IQ (SD) | 113.533 (13.65) | 103.680 (16.51) | 0.014 |
| Head motion (SD) | 0.059 (0.024) | 0.088 (0.031) | *p* < 0.001 |
| Inattention score | 49.41 | 82.71 | *p* < 0.001 |
| Hyperactivity score | 48.59 | 78.08 | *p* < 0.001 |
| Cued SST performance | | | |
| Certain Go accuracy (%) | 92.5 | 92.8 | 0.851 |
| Uncertain Go accuracy (%) | 94.5 | 93.3 | 0.440 |
| Stop accuracy (%) | 52.4 | 54.0 | 0.412 |
| SSRT (ms) | 290.2 | 303.8 | 0.364 |
| RT slowing (ms) | 20.4 | 23.8 | 0.833 |
| Certain Go RT (ms) | 483.7 | 515.5 | 0.124 |
| Uncertain Go RT (ms) | 504.1 | 539.1 | 0.064 |
| Ex-Gaussian modeling | | | |
| Certain Go RT (mu) | 398.0 | 416.9 | 0.113 |
| Certain Go RT (sigma) | 48.5 | 45.2 | 0.428 |
| Certain Go RT (tau) | 85.8 | 98.4 | 0.382 |
| Uncertain Go RT (mu) | 418.3 | 428.9 | 0.383 |
| Uncertain Go RT (sigma) | 45.8 | 44.3 | 0.778 |
| Uncertain Go RT (tau) | 85.8 | 110.1 | 0.051 |

A single-tailed $\chi^2$ test was performed to assess the presence of gender differences between children with ADHD and TD children. For all other between-group comparisons, two-tailed two-sample *t* tests were utilized. *TD* typically developing, *ADHD* attention deficit/hyperactivity disorder.

*p*s > 0.2, two-sample *t* test, Table 1). In comparison to TD children, children with ADHD had significantly higher inattention and hyperactivity/impulsivity scores (all *p*s < 0.001), greater head motion, and lower full-scale IQ (all *p*s < 0.05, two-sample *t* test, Table 1).

Because participants who have more severe clinical symptoms of ADHD, such as excessive head motion, are more likely to be excluded, we conducted a subsequent sensitivity analysis that included participants who have large head motions in the behavioral analysis. The larger behavior-only sample includes 50 children with ADHD and 37 TD children. There were no significant differences in age and gender between the two groups (all *p*s > 0.4, Supplementary Table S1).

Reactive control was estimated using the SSRT. We did not find a significant group difference in SSRT in the fMRI sample ($t_{59}$ = 1.13, *p* = 0.26, 95% CI = [−0.049, 0.013], Cohen's *d* = 0.293). Yet, in the behavior-only sample, we found that children with ADHD had significantly longer SSRT than TD children ($t_{85}$ = 2.46, *p* = 0.016, 95% CI = [0.008, 0.074], Cohen's *d* = 0.532, two-sample *t* tests).

Proactive control was estimated by the RT difference between Uncertain and Certain Go trials. Both children with ADHD and TD children exhibited significant proactive control, i.e., response slowing in Uncertain versus Certain Go trials (*p*s < 0.01, single-sample two-tailed *t* test). However, there was no significant group difference in the fMRI and behavior-only samples (*p*s > 0.7).

We also examined IIRV, quantified by the sigma and tau of the ex-Gaussian model[60], in the Certain and Uncertain Go trials separately. There were no significant group differences in sigma and tau in either trial type in the fMRI sample (all *p*s > 0.05). However, in the behavior-only sample, children with ADHD showed significantly higher tau than TD children in Uncertain ($t_{85}$ = 3.23, *p* = 0.002, 95% CI = [0.013, 0.052], Cohen's *d* = 0.70) and Certain Go trials ($t_{85}$ = 2.36, *p* = 0.02, 95% CI = [0.038, 0.451], Cohen's *d* = 0.51).

The behavioral results from the full sample indicate that children with ADHD exhibit poorer reactive control and less stable performance while maintaining proactive control compared to their TD peers. While

a similar trend was observed in the fMRI sample, these effects were less robust, likely due to the reduced sample size. This reduction primarily resulted from excessive head motion, which was particularly prevalent among children diagnosed with ADHD.

**Increased temporal variability of trial-evoked brain activation across trial types in children with ADHD**

Next, we assessed the temporal variability of trial-by-trial brain activation during the CSST. First, we conducted a single-trial GLM analysis to estimate the trial-evoked brain response for each single trial. Then, we fit the Gaussian model to the trial-wise activation per voxel per task condition, from which the four model parameters i.e., mean, standard deviation (STD), skewness, and kurtosis were calculated. The normal distribution of trial-evoked brain responses was assessed (Supplementary Methods and Supplementary Tables S2 and S4). The significance of the between-group comparison was determined using the threshold-free cluster enhancement method (TFCE, *p* < 0.05 corrected).

First, no significant between-group difference in mean was found in any task condition.

Then, we examined STD of trial-evoked neural responses. In comparison to TD children, children with ADHD exhibited significantly higher STD in the frontal, temporal, and cingulate regions, including the bilateral inferior frontal gyrus, preSMA, precuneus, and lateral and medial temporal lobe, in Uncertain and Certain Go and Stop trials (Fig. 2a and Supplementary Table S12).

Next, we measured kurtosis of trial-evoked neural responses. Kurtosis measures the degree of tailedness in a data distribution. Higher kurtosis is often associated with heavy tails or more outliers. In comparison to TD children, children with ADHD had significantly higher kurtosis in the lateral and ventromedial prefrontal, posterior insular, posterior parietal, and temporal regions in Uncertain and Certain Go trials (Fig. 2b and Supplementary Table S13).

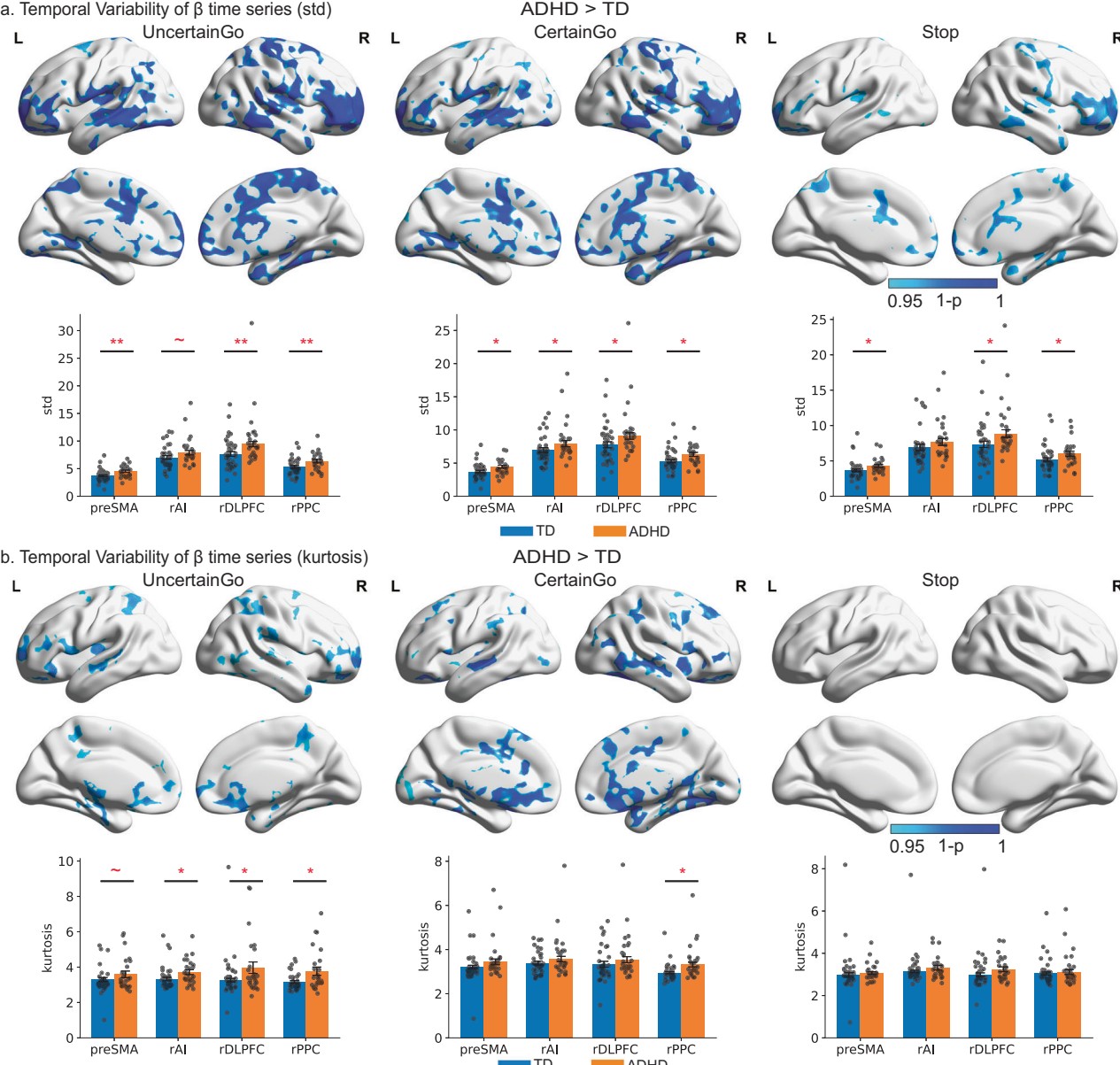

**Fig. 2 | Heightened temporal variability of neural responses in children with ADHD. a** Top: children with ADHD showed significantly greater standard deviation (std) of trial-evoked neural responses than TD controls (TFCE, $p < 0.05$, corrected). Bottom: compared to TD children, children with ADHD exhibited significantly higher standard deviations in all four ROIs during Certain Go trials (preSMA, $p = 0.012$; rAI, $p = 0.048$; rdlPFC, $p = 0.034$; rPPC, $p = 0.012$), and in the preSMA ($p = 0.006$), rdlPFC ($p = 0.007$), and rPPC ($p = 0.007$) during Uncertain Go and Stop trials (single-tailed two-sample $t$ tests with FDR correction, N(TD) = 35, N(ADHD) = 26). **b** Top: children with ADHD also showed significantly greater kurtosis of trial-evoked neural responses than TD children (TFCE, $p < 0.05$, corrected).

Bottom: children with ADHD had significantly higher kurtosis than TD children in the rAI ($p = 0.015$), rdlPFC ($p = 0.015$), and rPPC ($p = 0.014$) during Uncertain Go trials, and in the rPPC ($p = 0.001$) during Certain Go trials (single-tailed two-sample $t$ tests with FDR correction, N(TD) = 35, N(ADHD) = 26). Notes: data are presented as mean ± SEM. Error bars represent ±1 SEM. Each dot corresponds to an individual subject. TD is typically developing, ADHD attention-deficit/hyperactivity disorder, rAI right anterior insula, preSMA pre-supplementary motor cortex, rdlPFC right dorsal lateral prefrontal cortex, rPPC right posterior parietal cortex. *$p < 0.05$, **$p < 0.01$, ***$p < 0.001$. Source data are provided in the Source Data file.

Last, we examined skewness of trial-evoked neural responses. Skewness measures the degree of asymmetry in a data distribution. Positive skew indicates that data distribution has a long right tail (i.e., greater activation) whereas negative skew indicates that data distribution has a long left tail (i.e., weaker activation or deactivation). Children with ADHD showed significantly smaller skewness than TD children in the frontal, parietal, and cingulate regions, including bilateral inferior frontal gyrus, middle frontal gyrus, superior frontal gyrus, preSMA, posterior cingulate cortex/precuneus, and PPC, in Uncertain Go trials (Supplementary Fig. S1).

Taken together, children with ADHD demonstrated greater temporal variability of trial-evoked neural responses (larger STD) with more outlier activation (greater kurtosis) in the SN and FPN, suggesting unstable recruitment of core cognitive control systems during task performance.

Note that we also conducted control analyses using conventional GLM to examine averaged task-evoked brain activation but did not find significant between-group differences in any task conditions.

### Increased temporal variability of trial-evoked activation across trial types in the cognitive control system in children with ADHD and relation to ADHD symptoms

Next, we specifically tested whether the cognitive control system demonstrated significantly increased temporal variability in children with ADHD using pre-defined ROIs. These ROIs included the rAI and preSMA from the SN, as well as the rdlPFC and rPPC from the FPN (see Methods and Fig. 1h).

The ROI analysis focused on the standard deviation and kurtosis. We found that in comparison to TD children, children with ADHD exhibited significantly higher standard deviation in all four ROIs during Certain Go trials, in the preSMA, rdlPFC, and rPPC during Uncertain Go trials, and Stop trials (all $p$s < 0.05, FDR corrected, Fig. 2a). Moreover, children with ADHD had significantly higher kurtosis than TD children in the rAI, rDLPC, rPPC during Uncertain Go trials, the rPPC during Certain Go trials (all $p$s < 0.05, FDR corrected, Fig. 2b).

We also examined whether temporal variability of trial-evoked brain responses in the key regions of the cognitive control system is associated with the core clinical symptoms of ADHD. For brain measures, we used kurtosis because kurtosis is particularly influenced by extreme values. After controlling the age, gender, IQ, and head motion, inattention scores were significantly correlated with the kurtosis in the rPPC ($r_{partial} = 0.37$, $p = 0.042$, FDR corrected) during Certain Go trials, suggesting that higher kurtosis is associated with more severe inattention problems. This finding suggests that temporal variability of trial-evoked brain response has clinical implications.

### Weak spatial stability of trial-evoked brain response patterns across trial types in children with ADHD

Next, we examined whether the spatial stability of trial-evoked brain responses across different trial types during the CSST is weakened by the disorder. Spatial stability was quantified by the similarity (i.e., Pearson's correlation) of voxel-wise activation patterns between any pair of the same type of trials, and a searchlight approach was implemented to produce a whole-brain map of task-dependent spatial stability per subject[59,61]. The significance of spatial stability was determined using the threshold-free cluster enhancement method (TFCE, $p < 0.05$ corrected).

We found that children with ADHD demonstrated significantly weaker spatial stability in bilateral inferior frontal gyrus, preSMA, the lateral parietal cortex, and ventral/medial temporal lobe in both Certain and Uncertain Go trials than TD children ($p < 0.05$, corrected, Fig. 3a and Supplementary Table S14). Additionally, children with ADHD have lower spatial stability in the bilateral AI during Uncertain Go trials than TD children ($p < 0.05$, corrected, Fig. 3a and Supplementary Table S14). Children with ADHD also exhibited reduced spatial stability in the right inferior frontal gyrus, AI, PPC, posterior cingulate cortex, and preSMA during Stop trials compared to TD children ($p < 0.05$, corrected, Fig. 3a and Supplementary Table S14).

We did not find significantly increased spatial stability in any brain region in children with ADHD in comparison to TD children.

Together, these results suggest that children with ADHD recruit highly variable distributed brain regions from trial to trial during cognitive control performance in comparison to TD children.

### Weak spatial stability of trial-evoked brain response patterns during proactive and reactive control in children with ADHD

Because trials involving the same cognitive processes would exhibit greater representational similarity compared to trials that involve different processes, it allows us to identify representational brain systems uniquely associated with proactive and reactive control. Proactive control was measured by the RSA contrast between Uncertain and Certain Go trials, and reactive control was measured by the RSA contrast between Successful Stop and Uncertain Go trials. For proactive control, TD children showed significant spatial

stability in the right AI, inferior frontal gyrus, and bilateral visual cortex, whereas children with ADHD showed weak effects only in visual areas ($p < 0.05$, corrected, Fig. 3b and Supplementary Tables S15 and S16). For reactive control, TD children showed significant spatial stability in the bilateral AI, inferior frontal gyrus, PPC, anterior and posterior cingulate cortex, and visual areas, whereas children with ADHD showed significant spatial stability in the visual areas and right PPC ($p < 0.05$, corrected, Fig. 3b and Supplementary Tables S15 and S16).

### Reduced spatial stability of trial-evoked activation in the cognitive control system in children with ADHD

Next, we examined whether children with ADHD exhibited reduced spatial stability in the cognitive control system across different trial types. The ROI analyses revealed that in comparison to TD children, children with ADHD demonstrated significantly lower spatial stability in all four ROIs during Uncertain Go trials, in the rPPC during Certain Go trials, and in the preSMA, rdlPFC, and rPPC during Stop trials (all $p$s < 0.05, FDR corrected, Fig. 3a).

Furthermore, TD children showed significant spatial stability of trial-evoked neural responses in the rdlPFC and rPPC during proactive control and in all four ROIs during reactive control ($p < 0.05$, FDR corrected, Fig. 3b). Children with ADHD also exhibited significant spatial stability in the rAI, preSMA, and rPPC during reactive control but only in the rPPC during proactive control ($p < 0.05$, FDR corrected, Fig. 3b). Moreover, TD children demonstrated significantly greater spatial stability of trial-evoked neural responses during proactive control in the preSMA, rAI, and rdlPFC compared to children with ADHD ($p < 0.05$, FDR corrected, Fig. 3b), while no significant effect was observed during reactive control. Together, our findings suggest that children with ADHD have weakened spatial stability from trial to trial in the distributed brain systems and in the frontoparietal network during proactive control in comparison to TD children.

### Weak association between trial-evoked inhibition-alike brain response and RT in children with ADHD

Increased IIRV is a robust behavioral phenotype associated with ADHD, which is commonly regarded as an index of attention lapse or inattention[6,12,62]. However, behavioral fluctuation could also be driven by trial-by-trial response strategy adjustment, a dynamic proactive control process modulated by participants' time-varying anticipation[6,23,63,64]. Here we examined whether the degree of proactive control in trial-evoked brain response patterns can track RT in Certain and Uncertain Go trials. As proactive and reactive control may share similar neural underpinnings[49], we developed an "inhibitory control pattern" (ICP) index in each Certain and Uncertain Go trial by quantifying the similarity between trial-evoked brain response pattern on Go trial and a subject-specific "template" brain response pattern of inhibitory control (Fig. 1f). The subject-specific "template" brain response pattern of inhibitory control was obtained by averaging trial-evoked brain response pattern from all the Successful Stop trials (β-maps from the single-trial model). Then, a searchlight approach was implemented to produce a whole-brain map of the ICP index in Certain and Uncertain Go trials separately per subject (see Methods for more details). The larger the ICP value, the greater the similarity between the trial-evoked brain response pattern on Certain and Uncertain Go trials and the averaged brain response pattern on Successful Stop trials, indicating greater engagement of inhibitory control systems during Go trials. Last, we examined the correlation between voxel-wise ICP and RT across trials.

We found that trial-wise RT is significantly correlated with the ICP in the bilateral AI, right inferior frontal gyrus, superior frontal gyrus, preSMA in TD children in Certain and Uncertain Go trials

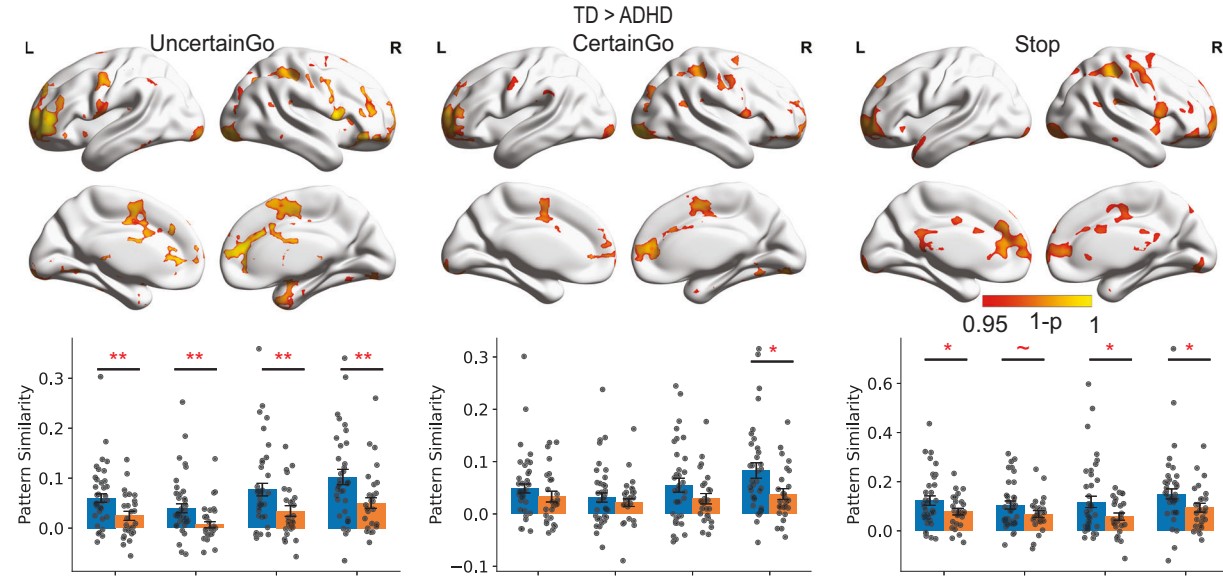

a. Weaker spatial stability of trial-evoked brain response patterns in children with ADHD than TD children

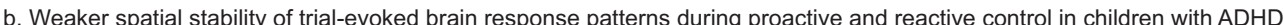

b. Weaker spatial stability of trial-evoked brain response patterns during proactive and reactive control in children with ADHD

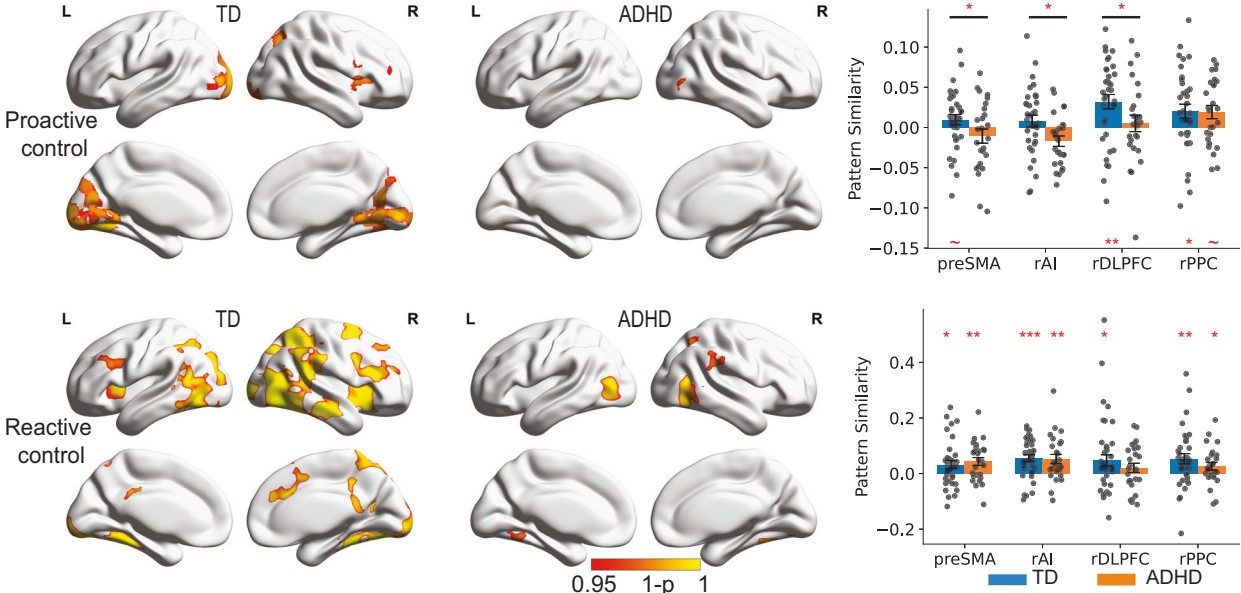

**Fig. 3 | Weaker spatial stability of trial-evoked neural responses in children with ADHD. a** Top: TD children showed greater spatial stability of trial-evoked neural responses than children with ADHD in salience and frontoparietal networks during Uncertain Go, Certain Go, and Stop trials (TFCE, $p < 0.05$, corrected). Bottom: children with ADHD demonstrated significantly lower spatial stability in all four ROIs during Uncertain Go trials (preSMA, $p = 0.006$; rAI, $p = 0.006$; rdlPFC, $p = 0.007$; rPPC, $p = 0.007$), in the rPPC ($p = 0.044$) during Certain Go trials, and in the preSMA ($p = 0.05$), rdlPFC ($p = 0.047$), and rPPC ($p = 0.047$) during Stop trials than TD children (single-tailed two-sample $t$ tests with FDR correction, $N$(TD) = 35, $N$(ADHD) = 26) **b** Top: Highly stable recruitment of salience and frontoparietal networks during proactive and reactive control was found in TD children (TFCE, $p < 0.05$, corrected) but not in children with ADHD. Bottom: TD children showed significant spatial stability of trial-evoked neural responses in the rdlPFC ($p = 0.002$) and rPPC ($p = 0.027$) during proactive control and in the preSMA ($p = 0.015$), rAI

($p < 0.001$); rdlPFC ($p = 0.015$) and rPPC ($p = 0.006$) during reactive control (single-tailed one sample $t$ tests with FDR correction, $N = 35$). Children with ADHD exhibited significant spatial stability in the rAI ($p = 0.004$), preSMA ($p = 0.004$), and rPPC ($p = 0.041$) during reactive control (single-tailed one sample $t$ tests with FDR correction, $N = 26$). TD children showed greater spatial stability during proactive control in the preSMA ($p = 0.036$), rAI ($p = 0.027$), and rdlPFC ($p = 0.036$) compared to children with ADHD (single-tailed two-sample $t$ tests with FDR correction, $N$(TD) = 35, $N$(ADHD) = 26). Notes: data are presented as mean ± SEM. Error bars represent ±1 SEM. Each dot corresponds to an individual subject. TD typically developing, ADHD attention-deficit/hyperactivity disorder, rAI right anterior insula, preSMA pre-supplementary motor cortex, rdlPFC right dorsal lateral prefrontal cortex, rPPC right posterior parietal cortex. *$p < 0.05$, **$p < 0.01$, ***$p < 0.001$. Source data are provided in the Source Data file.

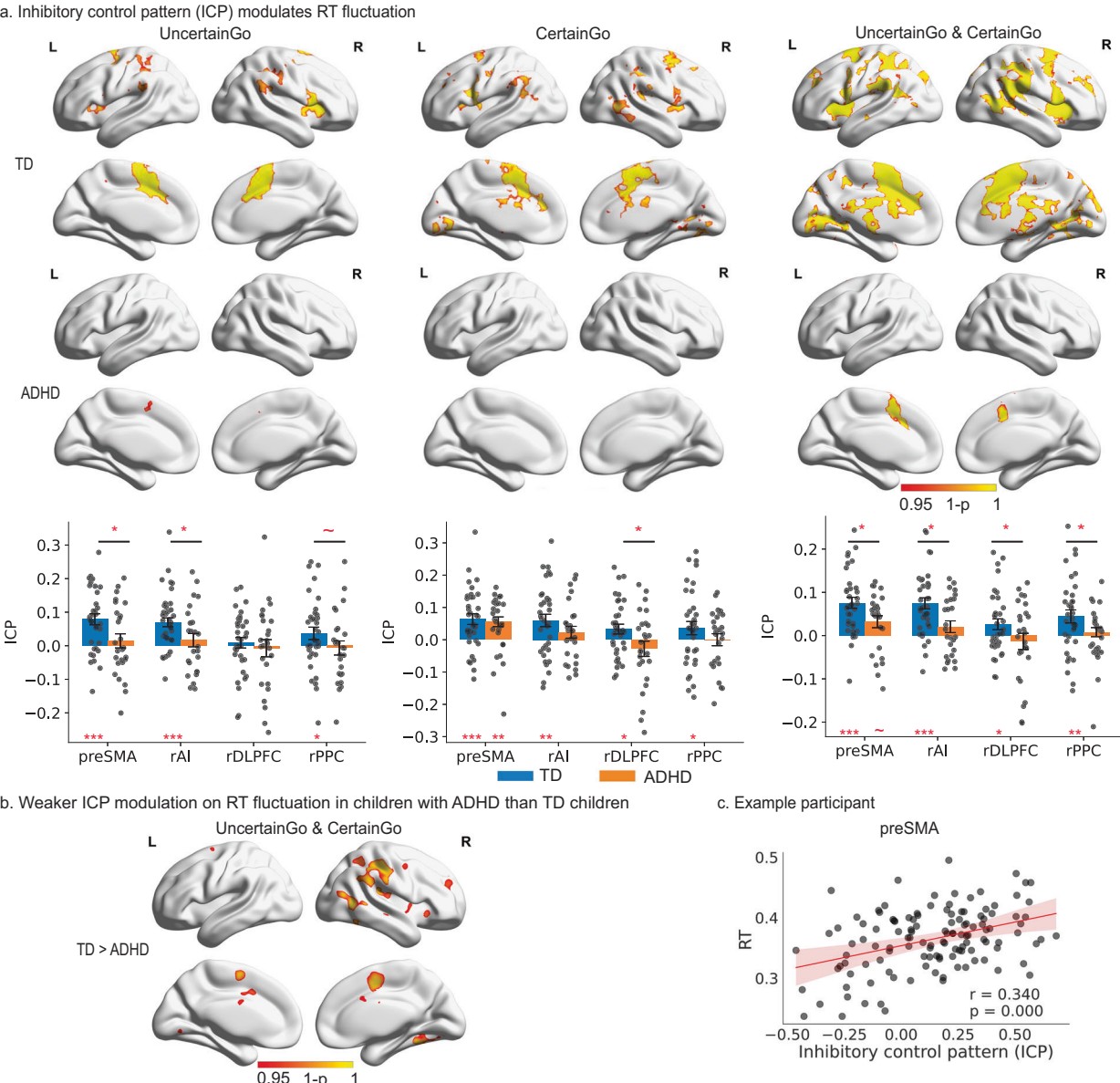

**Fig. 4 | Atypical neural dynamics underlie impaired behavioral regulation in children ADHD. a** Top: inhibitory control pattern (ICP) index of the salience and frontoparietal networks during Certain and Uncertain Go trials modulates trial-wise reaction time (RT) in TD children and children with ADHD (TFCE, $p < 0.05$, corrected). Bottom: TD children showed significant relationships between trial-wise RT fluctuations across trial types and ICP of the preSMA, rAI, and rPPC during Uncertain Go, all four ROIs during Certain Go, and when both Certain and Uncertain Go trials were considered (all $ps < 0.05$, single-tailed one sample $t$ tests with FDR correction, $N = 35$). Children with ADHD showed a significant relationship between trial-wise RT and ICP in the preSMA during Certain Go trials ($p = 0.002$, single-tailed one-sample $t$ tests with FDR correction, $N = 26$). TD children exhibited a significantly stronger association between trial-wise RT and the ICP in the preSMA ($p = 0.022$), rAI ($p = 0.011$); rdlPFC ($p = 0.034$), and rPPC ($p = 0.034$) during both Uncertain and Certain Go trials, in the rAI ($p = 0.021$) and preSMA ($p = 0.021$) during Uncertain trials, and in the rdlPFC ($p = 0.049$) in the Certain Go trials than children with ADHD (single-tailed two-sample $t$ tests with FDR correction, N(TD) = 35, N(ADHD) = 26). **b** TD children demonstrated significantly greater association between ICP index of regions from the SN and FPN and trial-wise RT than children with ADHD (TFCE, $p < 0.05$, corrected). **c** Data from an exemplary participant illustrated a positive association between ICP of the preSMA and RT. preSMA presupplementary motor area. Notes: data are presented as mean ± SEM. Error bars represent ±1 SEM. Each dot corresponds to an individual subject. TD typically developing, ADHD attention deficit/hyperactivity disorder, ICP inhibitory control pattern, rAI right anterior insula, preSMA presupplementary motor cortex, rdlPFC right dorsal lateral prefrontal cortex, rPPC right posterior parietal cortex. *$p < 0.05$, **$p < 0.01$, ***$p < 0.001$. Source data are provided in the Source Data file.

($p < 0.05$, corrected, Fig. 4a and Supplementary Table S17). Children with ADHD exhibited a similar effect in the supplementary motor area and anterior cingulate cortex during Certain and Uncertain Go trials ($p < 0.05$, corrected, Fig. 4a and Supplementary Table S18), though the spatial extent of the effect appears to be smaller in children with ADHD than TD children. The between-group comparison further revealed that children with ADHD have significantly weaker relationships between trial-wise RT and ICP in anterior cingulate cortex/supplementary motor cortex, right AI, right posterior middle temporal gyrus, and right supramarginal gyrus extending to the postcentral gyrus in Certain and Uncertain Go trials ($p < 0.05$, corrected, Fig. 4b and Supplementary Table S19). These findings suggest that TD children can flexibly recruit inhibitory control systems to implement proactive control and thereby modulate trial-wise response time, but such ability is much weakened in children with ADHD.

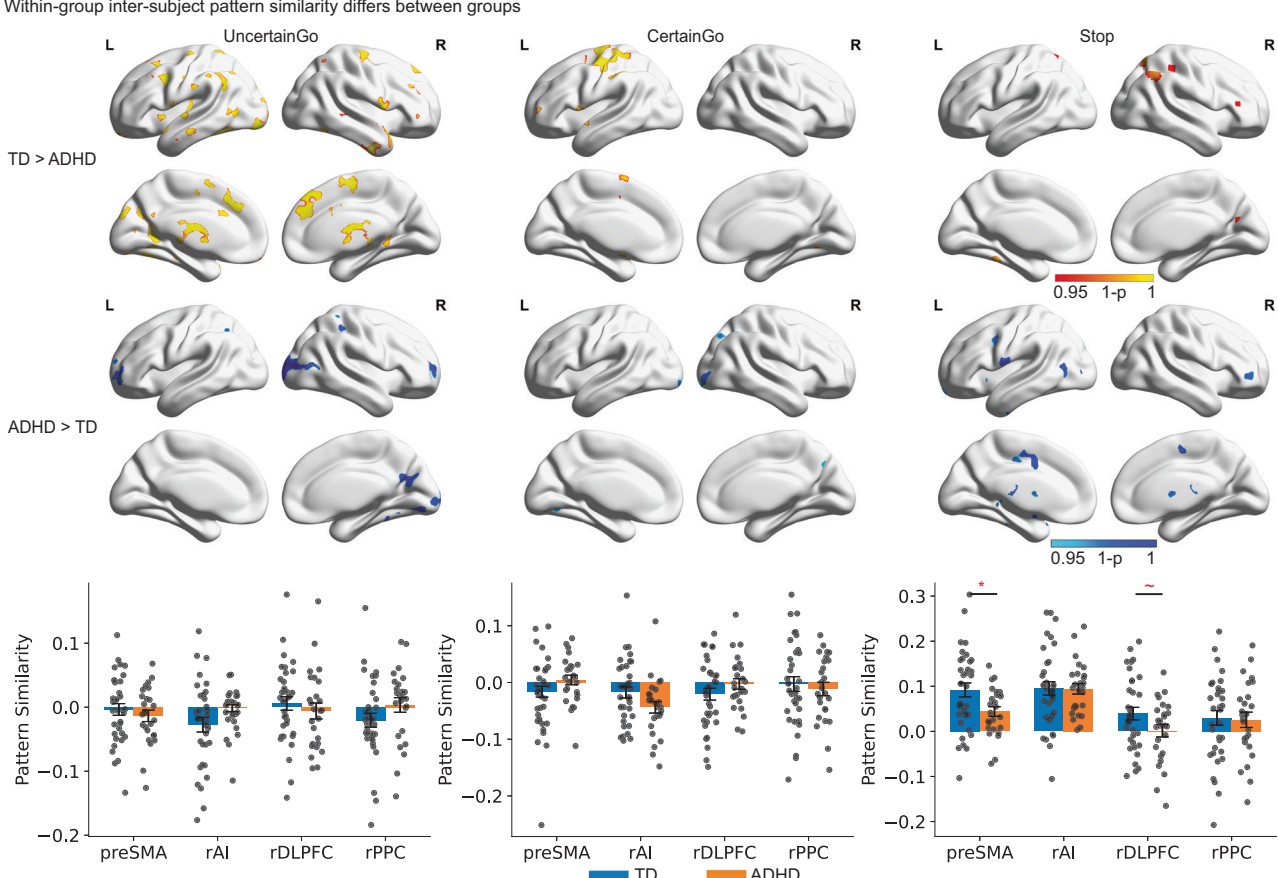

**Fig. 5 | Heterogeneity of trial-evoked neural responses in children with ADHD.**
Top: Within-group inter-subject spatial pattern stability analyses revealed between-group differences during Uncertain Go, Certain Go, and Stop trials (TFCE, $p < 0.05$, corrected). Bottom: TD children showed significantly higher within-group inter-subject spatial stability in the preSMA ($p = 0.039$) during Stop trials than children with ADHD (single-tailed two-sample $t$ tests with FDR correction, $N$(TD) = 35,

$N$(ADHD) = 26). Notes: data are presented as mean ± SEM. Error bars represent ±1 SEM. Each dot corresponds to an individual subject. TD typically developing, ADHD attention-deficit/hyperactivity disorder, rAI right anterior insula, preSMA pre-supplementary motor cortex, rdlPFC right dorsal lateral prefrontal cortex, rPPC right posterior parietal cortex. *$p < 0.05$, **$p < 0.01$, ***$p < 0.001$. Source data are provided in the Source Data file.

## Weak association between trial-evoked inhibition-alike brain response in the cognitive control system and RT in children with ADHD

Next, we examined whether children with ADHD exhibited a weak association between ICP in the cognitive control system and RT. TD children demonstrated a significant relation between trial-wise RT fluctuations across trial types and ICP of the preSMA, rAI, and rPPC during Uncertain Go, all four ROIs during Certain Go and when both Certain and Uncertain Go trials were considered (all $p$s < 0.05, FDR corrected, Fig. 4a). Children with ADHD also demonstrated significant relationships between trial-wise RT and ICP in the preSMA during Certain ($p < 0.05$, FDR corrected, Fig. 4a), but not in the other ROIs (all $p$s > 0.05). Moreover, in comparison to children with ADHD, TD children exhibited a significantly stronger association between trial-wise RT and the ICP of all four ROIs during both Uncertain and Certain Go trials, in the rAI and preSMA during Uncertain trials, and in the rdlPFC in the Certain Go trials (all $p$s < 0.05, FDR corrected).

We also examined whether the association strength between ICP in the cognitive control system and RT is associated with severity of clinical systems. After accounting for the age, gender, IQ, and head motion, the association strength in the rdlPFC during the Certain Go trial was significantly correlated with inattention score ($r_{partial} = -0.381$, $p = 0.028$, FDR corrected); and the association strength in the rPPC was also significantly correlated with inattention score when both Uncertain and Certain Go trials were considered ($r_{partial} = -0.378$, $p = 0.03$,

FDR corrected). Our findings suggest that children who exhibit severe clinical symptoms are less likely to effectively modulate their cognitive control systems in order to adapt their response strategies.

## Group-specific, inter-subject similarity patterns underlying proactive and reactive control

Next, we examined whether children with ADHD demonstrated weakened spatial stability of trial-evoked brain responses across different trial types within their respective groups. Notably, for a participant's within-group spatial similarity, the participant's own brain response pattern was excluded when computing the averaged spatial pattern of brain response of the group that the participant belongs to, which is to avoid overestimation of the within-group spatial similarity. The searchlight algorithm was used to estimate within-group inter-subject similarity across the whole brain, and its statistical significance was determined using the threshold-free cluster enhancement method (TFCE, $p < 0.05$ corrected).

We found that, in comparison to TD children, children with ADHD demonstrated significantly weakened inter-subject spatial stability in distributed areas in bilateral inferior frontal gyrus, left AI, PPC, pre-central gyrus, right middle frontal gyrus, superior frontal gyrus, pre-SMA/supplementary motor cortex, bilateral middle and inferior temporal lobe, visual cortex, and subcortical regions including caudate and thalamus in Uncertain Go trials; in precentral/postcentral gyrus and left inferior frontal gyrus in Certain Go trials; and rPPC, and

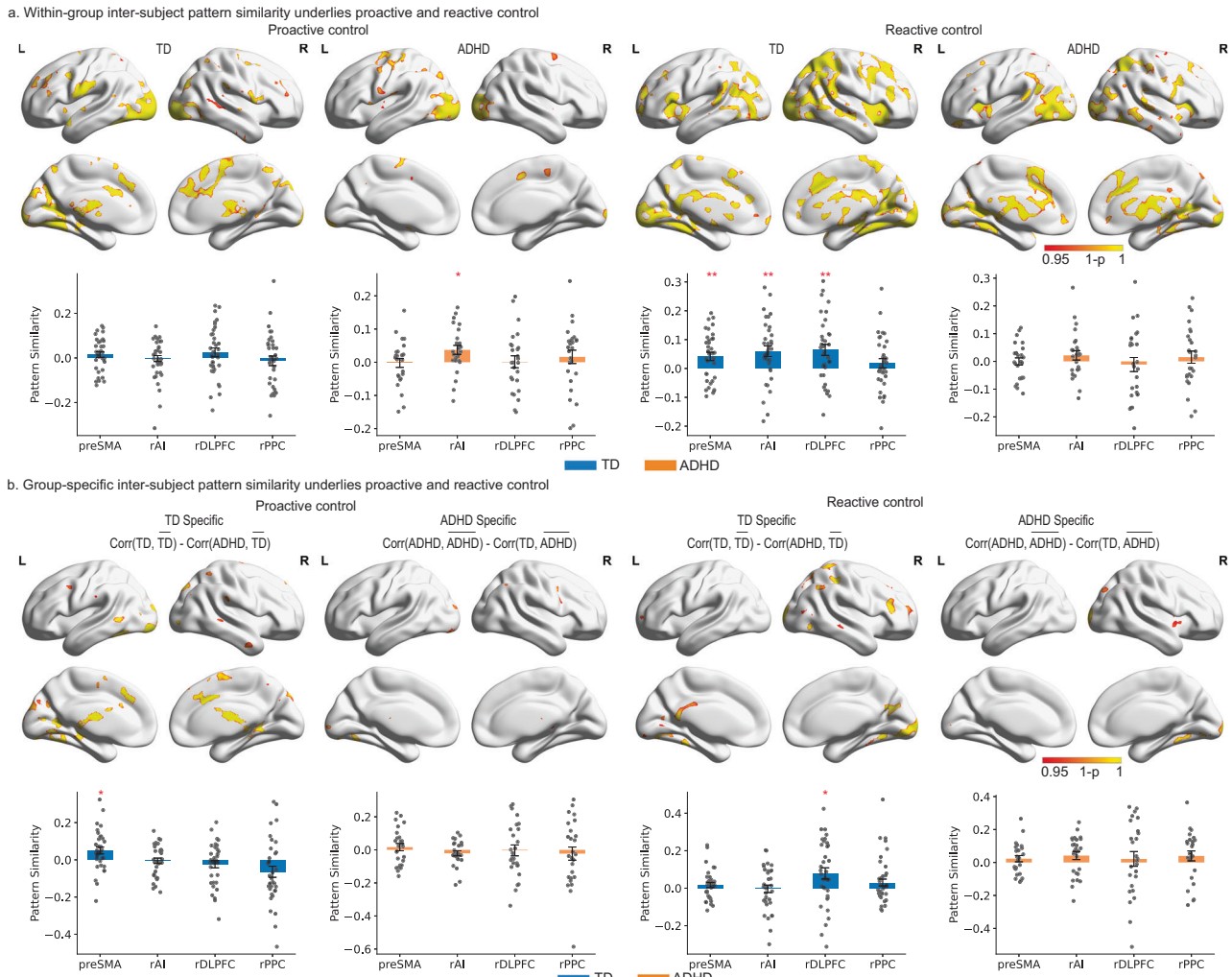

**Fig. 6 | Group-specific inter-subject pattern similarity during proactive and reactive control. a** Top: TD children and children with ADHD showed distinct within-group inter-subject spatial pattern stability during proactive and reactive control (TFCE, $p < 0.05$, corrected). Bottom: TD children showed significant within-group inter-subject spatial similarity of brain responses in the rAI ($p = 0.002$), preSMA ($p = 0.004$), and rdlPFC ($p = 0.002$) during reactive control (single-tailed one sample $t$ tests with FDR correction, $N = 35$). Children with ADHD exhibited significant within-group inter-subject spatial similarity of brain responses in the rAI during proactive control ($p = 0.02$, single-tailed one-sample $t$ tests with FDR correction, $N = 26$). **b** Top: Group-specificity of inter-subject spatial pattern similarity analyses revealed more heterogeneous trial-evoked neural response during

proactive and reactive control in children with ADHD than TD children (TFCE, $p < 0.05$, corrected). Bottom: TD children showed significant group specificity of spatial stability in the preSMA during proactive control ($p = 0.02$, single-tailed one sample $t$ tests with FDR correction, $N = 35$), and in the rdlPFC during reactive control ($p = 0.02$, single-tailed one sample $t$ tests with FDR correction, $N = 35$). Notes: Data are presented as mean ± SEM. Error bars represent ±1 SEM. Each dot corresponds to an individual subject. TD typically developing, ADHD attention deficit/hyperactivity disorder, rAI right anterior insula, preSMA, presupplementary motor cortex, rdlPFC right dorsal lateral prefrontal cortex, rPPC right posterior parietal cortex. *$p < 0.05$, **$p < 0.01$, ***$p < 0.001$. Source data are provided in the Source Data file.

bilateral visual cortex in Stop trials ($p < 0.05$, corrected, Fig. 5 and Supplementary Table S20). Children with ADHD also exhibited strengthened spatial stability in visual cortex across conditions; in bilateral frontal pole, and supramarginal gyrus in Uncertain Go trials; and in right inferior frontal gyrus, bilateral precentral gyrus, and thalamus in Stop trials ($p < 0.05$, corrected, Fig. 5 and Supplementary Table S21).

Furthermore, we asked whether children with ADHD not only have weakened inter-subject spatial stability in each type of trial but also show more variable inter-subject spatial patterns of brain response during proactive and reactive control than TD children. So, we examined the within-group inter-subject spatial similarity of brain response during proactive and reactive control per group. The preSMA, bilateral inferior frontal gyrus, left precentral/postcentral area, left lateral and medial temporal lobe, and visual cortex showed significant inter-subject pattern similarity during proactive control in TD children ($p < 0.05$, corrected,

Fig. 6a and Supplementary Tables S22 and S23), whereas children with ADHD showed a similar effect in the preSMA, visual and motor cortex. The bilateral AI, preSMA, PCC, lateral and medial temporal lobe, and basal ganglia showed significant inter-subject pattern similarity during reactive control for both TD and ADHD groups ($p < 0.05$, corrected, Fig. 6a and Supplementary Tables S22 and S23).

Next, we examined the group specificity of inter-subject spatial similarity of brain response during proactive and reactive control per group. For each participant, we computed within-group inter-subject spatial similarity, i.e., the spatial similarity between a participant and the group of the participant, and between-group inter-subject spatial similarity, i.e., the spatial similarity between a participant and the group that the participant does not belong to (Fig. 1g). The group specificity was defined by the difference between within-group inter-subject spatial similarity and between-group inter-subject spatial similarity. Positive values indicate that a participant's regional

activation pattern is more similar to the rest of the participants from the same group than the participants from the other group, whereas negative values indicate vice versa. During proactive control, we observed greater spatial similarity among TD children than between TD children and children with ADHD in the supplementary motor cortex, preSMA, bilateral posterior middle temporal gyrus, bilateral medial temporal lobe, as well as subcortical clusters, including the thalamus and caudate ($p < 0.05$, corrected, Fig. 6b and Supplementary Tables S24 and S25). During reactive control, the right inferior frontal gyrus, PCC, middle temporal gyrus, and lateral parietal cortex show greater spatial similarity among TD children than between TD children and children with ADHD during reactive control, whereas the right AI showed greater similarity among children with ADHD than TD children ($p < 0.05$, corrected, Fig. 6b and Supplementary Tables S24 and S25).

### Group-specific, inter-subject pattern similarity during proactive and reactive control in the cognitive control system

We first conducted an ROI-based analysis to examine whether TD children and/or children with ADHD exhibited consistent within-group inter-subject similarity across different trial types. Our results revealed that, in comparison to children with ADHD, TD children demonstrated significantly higher within-group inter-subject spatial stability in the preSMA during Stop trials ($p < 0.05$, FDR corrected, Fig. 5). We did not observe significant between-group differences in the ROIs during UnCertain and Certain Go trials ($p > 0.05$).

Next, we examined inter-subject pattern similarity during proactive and reactive control in the cognitive control system. Our ROI analyses found significant within-group inter-subject spatial similarity of brain responses for TD children in the rAI, preSMA, and rdlPFC during reactive control ($ps < 0.05$, FDR corrected, Fig. 6a), but not in the rPPC ($p = 0.13$). Children with ADHD only exhibited significant within-group inter-subject spatial similarity of brain responses in the rAI during proactive control ($p = 0.02$, FDR corrected, Fig. 6a).

Then, we examined the group specificity of inter-subject spatial similarity during proactive and reactive control in key regions of the SN and the FPN within each group. Only TD children demonstrated significant group specificity of spatial stability in the preSMA during proactive control ($p = 0.02$, FDR corrected, Fig. 6b), and in the rdlPFC during reactive control ($p = 0.02$, FDR corrected, Fig. 6b). We did not find a significant group specificity during proactive and reactive control for children with ADHD (all $ps > 0.1$, FDR corrected).

Last, we investigated whether similarity to the averaged spatial patterns of the ADHD group during proactive and reactive control is associated with individual differences in core symptoms of ADHD. We found a significant positive correlation between the pattern similarity to the ADHD group averaged spatial pattern during proactive control in rAI and inattention score ($r = 0.354$, $p = 0.037$, FDR corrected). And this effect remained significant after accounting for age, gender, IQ, and head motion ($r_{partial} = 0.355$, $p = 0.009$, Fig. 7). Our results indicated that a more ADHD-like neural activity pattern in the rAI during proactive control was associated with more severe clinical symptoms.

### Robustness with respect to potential confounding factors

While we have meticulously excluded trials with large head motion from the single-trial model (see details in Methods), head motion may remain a potential confounding variable and bias the temporal variability and spatial stability analyses. Although there were no significant group differences in age and gender, these factors could also influence our findings. To assess the robustness of our results with respect to head motion, age, and gender, we conducted additional analyses and replicated the main findings (see Supplementary Method, Supplementary Results Figs. S2 and S4 and Tables S5-S8).

## Discussion

We used cutting-edge analytic methodology to provide a new window into the temporal and spatial stability of neural dynamics elicited by proactive and reactive control processes in children with ADHD. We modeled single-trial brain responses and utilized representational similarity analysis to relate neural coding stability to heterogeneity, temporal fluctuation in behavior, and clinical symptoms. Children with ADHD showed heightened temporal variability in task-evoked responses within key regions of the SN and FPN that are associated with behavioral variability across trials. They also demonstrated reduced spatial stability of activation patterns supporting both reactive stopping and proactive response preparation. Additionally, we observed decreased homogeneity of within-group response patterns in ADHD, indicating more variable neural recruitment strategies between children with ADHD. Importantly, indices of increased variability were further associated with more severe attention deficits and hyperactive/impulsive symptoms.

Our findings underscore the fundamentally disrupted stability of neural coding across trials in ADHD, leading to inefficient implementation of cognitive control mechanisms, providing a potential mechanism for inconsistent behavioral regulation. Our analytical approach advances knowledge of functional brain abnormalities in ADHD, moving beyond gross activation deficits toward a more precise mechanistic understanding. Our findings may help catalyze more targeted models of inhibitory control dysfunction and neurobiological heterogeneity in ADHD.

Temporal instability of cognitive and behavioral processes is increasingly recognized as a core feature of ADHD[5,12,55,56,65]. Neural correlates of this intra-individual variability could manifest as increased fluctuation of task-evoked responses that standard averaging approaches can miss. To address this, we sought to elucidate moment-to-moment changes in inhibitory control processes by modeling single-trial variability in neural response. We expected that this approach would reveal instability in neural dynamics obscured in traditional group comparisons, providing a more precise understanding of cognitive control deficits in ADHD.

Temporal variability in neural responses has garnered increasing attention, particularly in resting-state studies, and more recently, in task-based paradigms[66,67]. This metric encompasses measures such as standard deviation and kurtosis, which capture not only the dispersion of neural activity but also the presence of frequent extreme values. These measures differ significantly from traditional approaches which rely on time-averaged responses, and have emerged as important and useful neural indicators related to task performance and aging[68,69]. In the present study, we extended this approach to task-evoked responses in ADHD, estimating trial-specific β time series and fitting them to a Gaussian model to derive metrics of neural variability, including standard deviation, skewness, and kurtosis. This method allowed us to characterize the dynamic nature of neural responses in ADHD beyond what conventional analyses typically reveal.

We found that children with ADHD exhibited elevated temporal variability compared to TD children. This was evidenced by increased standard deviation and kurtosis, as well as decreased skewness, of trial-wise brain responses in multiple prefrontal, parietal, and temporal cortex regions across all CSST conditions (Fig. 2). These findings suggest that children with ADHD have more variable and extreme activation levels from trial to trial than TD children.

Our results align with previous work showing more variable and noisier intrinsic brain activity in ADHD. For example, Cai et al. found more variable brain states during resting-state fMRI in children with ADHD compared to controls[11]. Similarly, several resting-state EEG and fMRI studies have reported increased neural variability and noise in ADHD across different brain regions and frequency bands[67,68,70–72]. Our findings extend these results by demonstrating heightened temporal variability of task-evoked neural responses in ADHD. While previous

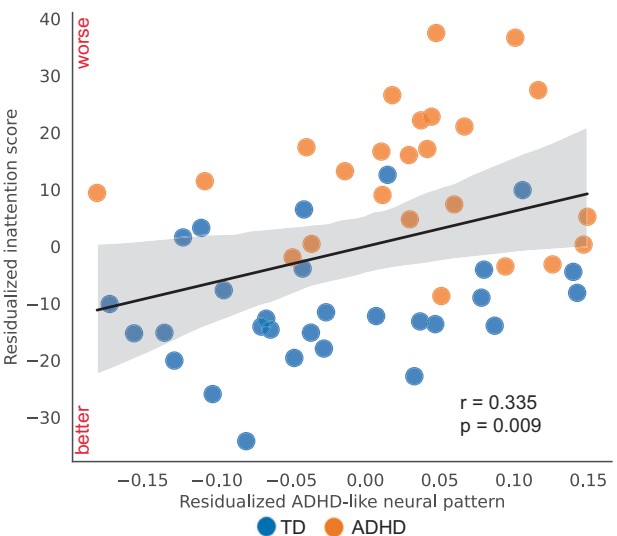

**Fig. 7 | Neural variability of the rAI predicts ADHD symptom severity.** The residualized ADHD-like activation pattern in the right anterior insula (rAI) exhibited a significant correlation with the residualized inattention scores after accounting for age, gender, IQ, and head motion ($r = 0.355$, $p = 0.009$; N(TD) = 35, N(ADHD) = 26). Notes: each dot corresponds to an individual subject. TD (blue) typically developing, ADHD (orange) attention-deficit/hyperactivity disorder, rAI right anterior insula. *$p < 0.05$, **$p < 0.01$, ***$p < 0.001$. Source data are provided in the Source Data file.

studies focused on intrinsic or resting-state brain activity, we show that the increased variability persists during cognitive demands and affects task-relevant brain regions. This task-based neural variability may have important implications for cognitive and behavioral flexibility.

Taken together, our findings provide clear evidence of heightened trial-wise neural variability in ADHD. The more temporally unstable and extreme task-related brain responses observed in ADHD likely contribute to the overall lack of consistency in performance. The noisier neural dynamics may reflect inefficiencies in cognitive control processes necessary for maintaining stable attention and motor planning over time. These results build upon a growing literature recognizing temporal variability as a core feature of ADHD and provide evidence linking this instability to dysfunctional inhibitory control networks. Increased neural variability may thus be a key mechanism underlying the hallmark behavioral inconsistency in ADHD, offering insights into the neurobiology of this disorder.

In addition to temporal variability, we examined the spatial stability of trial-evoked brain responses during task performance. Spatial stability, defined as the similarity in activation patterns between pairs of trials within a given condition, was quantified using RSA[59]. This approach provides a powerful tool for investigating the distributed coding properties of cognitive control networks[73,74]. By quantifying pattern similarity both between and within individuals across conditions, spatial stability offers a distinct perspective from mean activation, offering insights beyond those provided by traditional mean activation analyses. Previous studies, for instance, have demonstrated that spatial stability can predict memory performance, even when univariate mean activation was controlled[75–77]. However, to our knowledge, no study has used trial-based RSA to examine inhibitory control in ADHD.

We computed the spatial stability of each voxel by measuring pattern similarity among neighboring voxels and used a searchlight algorithm to generate whole-brain maps of task-dependent spatial stability. Compared to controls, children with ADHD exhibited less stable activation patterns (i.e., greater spatial variability) within and across all task conditions (Uncertain Go, Certain Go, and Stop, Fig. 3a).

Importantly, this reduced spatial stability was observed in key regions of the salience and frontoparietal networks, including AI, inferior frontal gyrus, middle frontal gyrus, preSMA. These findings suggest that the salience and frontoparietal networks, critical for cognitive control, are recruited more variably from trial to trial in children with ADHD compared to TD children.

Next, we examined the spatial stability of neural representations associated with reactive and proactive control. Reactive control was assessed by contrasting Successful Stop and Uncertain Go trials: participants deployed proactive control in both these types of trials, but the reactive control process was only triggered in Stop trials. Consistent with previous studies[21,22,30], we found that reactive control recruited key regions in the salience and frontoparietal networks (Fig. 3b). Crucially, TD children showed high spatial stability in widely distributed regions overlapping with the salience and frontoparietal networks during reactive control, while children with ADHD showed high spatial stability in more restricted regions, including bilateral visual cortex and right supramarginal gyrus.

Uniquely, our experimental design involved both Uncertain Go and Certain Go trials, which allowed us to examine proactive control, an aspect that has eluded previous studies and has required complex computational modeling to uncover[6,63,78]. While some studies have suggested that proactive control recruits a highly similar brain system as reactive control[46,48,49,52–54], the spatial stability of neural representations associated with proactive control in children with ADHD remained unknown.

In the present study, proactive control was assessed by contrasting Uncertain Go and Certain Go trials, as participants proactively slowed their responses in Uncertain Go trials. We found that TD children showed high spatial stability in the AI and the inferior frontal gyrus during proactive control, but this pattern was not observed in children with ADHD. The reduced spatial stability observed in children with ADHD during proactive control tasks provides insights into the neural mechanisms underlying their difficulties in implementing and maintaining preparatory control strategies. This finding may explain the inconsistent behavioral performance often observed in this population when anticipatory control is required[6,79].

Our findings suggest that while proactive and reactive control engage similar brain networks, particularly in the salience and frontoparietal regions, the consistency of recruitment within these networks differs between TD children and children with ADHD. This difference in spatial stability may contribute to a broad range of cognitive control deficits observed in ADHD. Specifically, our results indicate that the inhibitory control system was not recruited in a consistent manner from trial to trial in children with ADHD during both proactive and reactive control. The overlap in the salience and frontoparietal networks suggests that the instability of neural representations in these systems may be a common mechanism underlying deficits in both aspects of cognitive control.

In summary, our findings highlight the importance of considering the stability of neural representations associated with both proactive and reactive control in understanding the neurocognitive deficits in ADHD. The reduced consistency of recruitment within the salience and frontoparietal networks may be a key factor contributing to the cognitive control challenges faced by children with ADHD, and this may inform the development of targeted interventions aimed at improving the stability of these neural representations.

Given the similarity in brain systems recruited by proactive and reactive control[46,48,49,52–54], we hypothesized that if a Go trial elicits an activation pattern resembling the typical activation pattern of reactive control, the Go trial likely involves strong proactive control, leading to elongated reaction times. This reflects the parallel neural activation patterns involved in maintaining/anticipating and executing task demands during uncertain Go and Success Stop trials. To test this hypothesis, we first created a typical brain activation pattern of

reactive control by averaging trial-evoked brain responses across all Successful Stop trials for each participant. We then developed an ICP index by computing the similarity between the trial-evoked response in each Go trial and the typical activation pattern of reactive control.

We examined the correlation between the trial-wise ICP index and RT, with a high correlation indicating that greater spatial similarity between the Go trial-evoked response and the ICP is associated with longer RTs. Crucially, we found that the ICP index in key nodes of the salience and frontoparietal networks strongly predicted trial-wise reaction time in TD children. In contrast, this brain-behavior association was significantly attenuated in children with ADHD, suggesting a fundamental disruption in their dynamic cognitive control processes (Fig. 4).

Our findings have several important implications. First, the results align with our hypothesis that proactive control recruits a similar inhibitory control system as reactive control. Second, it suggests that spatial variability in Go trials, specifically the degree to which the trial-evoked pattern resembles activation elicited by reactive inhibitory control, encodes the amount of proactive control implemented by TD children. Third, spatial variability of trial-evoked brain responses in children with ADHD is less associated with trial-by-trial adjustment of proactive control compared to TD children.

These findings provide insights into the neural mechanisms underlying the behavioral regulation deficits observed in ADHD. While children with ADHD may have equivalent response slowing in Uncertain Go than Certain Go trials as TD children, children with ADHD demonstrated significant deficits in proactively adapting their response strategy[6]. This impairment in adaptive proactive control may contribute to the increased behavioral variability and inconsistency often observed in ADHD.

ADHD is characterized by substantial heterogeneity in symptom profiles and cognitive deficits across individuals[10,80,81]. Despite this clinical variability, little is known about the heterogeneity in task-dependent neural responses across individuals with ADHD. This gap in knowledge may contribute to the inconsistent findings in the ADHD neuroimaging literature, as highlighted by recent meta-analyses[81,82]. To address this issue, we examined within-group spatial similarity during proactive and reactive control for each participant. We defined within-group spatial similarity as the extent to which an individual's brain activation pattern resembles the averaged activation pattern of their respective group (ADHD or TD). We hypothesized that a more homogeneous group would show greater within-group spatial similarity compared to a less homogeneous group.

Consistent with our hypothesis, TD children demonstrated greater within-group spatial similarity across a widely distributed brain system during both reactive and proactive control. In contrast, children with ADHD exhibited within-group similarity in more restricted brain regions, primarily those implicated in sensorimotor processing (Fig. 6a). This finding suggests that children with ADHD recruit more heterogeneous and idiosyncratic brain systems during cognitive control tasks. Such neural heterogeneity may reflect the diverse cognitive and behavioral profiles observed in ADHD and could explain why group-level analyses often fail to capture consistent neural markers of the disorder.

To further investigate the group specificity of neural response patterns, we developed a measure of group-specific spatial similarity by subtracting between-group spatial similarity from within-group spatial similarity. The between-group spatial similarity was defined as the extent to which a participant's brain activation pattern resembles the average activation pattern of the opposite group. Thus, the spatial similarity for a TD child measured the degree to which the child's activation pattern is more similar to the averaged activation pattern among TD children than the averaged activation pattern among children with ADHD. Using this approach, we found that TD children showed high group specificity of spatial similarity in the inferior frontal

gyrus and PPC during reactive control, and in the preSMA and subcortical regions during proactive control (Fig. 6b). These findings suggest that TD children recruit these regions more consistently during cognitive control tasks, whereas children with ADHD show more variable recruitment of these areas.

The heterogeneity in neural response patterns observed in children with ADHD may stem from several factors. First, the variability in symptom profiles and cognitive deficits across individuals with ADHD may be associated with distinct neural mechanisms[10,81,83]. Second, children with ADHD may employ more diverse cognitive strategies during task performance, leading to greater variability in brain activation patterns[10,84]. Finally, the inherent instability of neural dynamics in ADHD, as demonstrated by our findings of increased temporal variability and reduced spatial stability, may also contribute to the heterogeneity of neural response patterns.

Taken together, our findings reveal that children with ADHD exhibit more heterogeneous neural response patterns during cognitive control tasks compared to TD children. This neural heterogeneity may reflect the diverse clinical presentations and cognitive profiles observed in ADHD. Our approach of examining within-group and between-group spatial similarity provides a framework for investigating the neural basis of symptom variability in ADHD and other neuropsychiatric disorders.

Having established that children with ADHD exhibit increased temporal variability and reduced spatial stability of neural responses during cognitive control tasks, we next investigated whether these neural measures could predict individual differences in ADHD clinical symptoms. We focused on two core symptom domains: inattention and hyperactivity/impulsivity.

We found that higher temporal variability of trial-evoked neural responses in one key region of the salience and frontoparietal networks, the right PPC, was associated with more severe inattention symptoms. This finding aligns with previous research suggesting a potential association between stimulation of the right PPC region and improvements in attentional control during task performance[85]. It suggests that the increased neural variability observed in children with ADHD is not merely a reflection of general neurodevelopmental differences but is directly related to the severity of clinical impairments. The link between temporal variability and symptom severity may stem from the role of these networks in maintaining stable goal-directed behavior and suppressing irrelevant distractors[30,31,74,86–91]. Increased variability in these networks may lead to more frequent lapses of attention and difficulty inhibiting impulsive responses, thereby exacerbating ADHD symptoms.

Furthermore, we found that children who exhibited more ADHD-like spatial activation patterns during proactive control had more severe inattention symptoms (Fig. 7). This finding aligns with our previous work showing that children with less adult-like activation patterns during inhibitory control tasks had weaker overall inhibitory control function[92]. However, our previous study focused on averaged activation patterns across trials, potentially obscuring trial-by-trial variability in neural responses[92]. The current findings extend our previous work by demonstrating that the spatial stability of trial-evoked brain responses, rather than just the average activation pattern, is predictive of ADHD symptom severity. This metric of neural stability may provide a more robust measure of the neural mechanisms underlying ADHD symptoms and may serve as a potential biomarker for the disorder.

The association between less stable spatial activation patterns in the rAI and more severe inattention symptoms suggests that inattention in ADHD is linked to inconsistent neural representations, particularly within the SN. This finding highlights the importance of considering not only the magnitude of neural activation but also the consistency and typicality of neural response patterns when investigating the neural basis of ADHD symptoms.

Our findings have important implications for developing interventions targeting the neural mechanisms underlying ADHD symptoms. Specifically, interventions aimed at enhancing the stability of neural responses in the salience and frontoparietal networks could potentially reduce symptom severity in children with ADHD. For instance, neurofeedback training[93–97] could be tailored to improve the consistency of activation patterns in these key networks. Moreover, the spatial stability of trial-evoked neural responses could serve as a valuable biomarker for tracking treatment response and predicting long-term outcomes in ADHD.

Importantly, our analytical framework has broad applicability beyond ADHD, extending to other psychiatric disorders characterized by cognitive control deficits. Our recent work has demonstrated a generalized neural dynamic mechanism underlying cognitive control across various tasks, developmental stages, and clinical populations[42].

Our study has several limitations to consider. First, while we observed significant between-group differences in reactive control in the full behavioral sample, these differences were not significant in the fMRI sample. This discrepancy likely stems from the necessary exclusion of participants with excessive head motion during scanning, potentially biasing our neuroimaging sample towards less hyperactive individuals. This highlights a persistent challenge in ADHD neuroimaging research: generalizing findings to more severely affected or hyperactive participants. Second, although we focused on proactive control, other factors, such as fatigue or motivation, could contribute to slower response times. However, these factors are typically associated with decreased, rather than increased, neural stability, making them unlikely explanations for our observed patterns. Third, our whole-brain analysis revealed significant effects in regions not typically associated with cognitive control, such as the visual cortex and thalamus. While these findings may reflect important bottom-up or feedback-driven processes interacting with cognitive control, they were not the primary focus of our study and require further investigation. Fourth, ADHD's heterogeneity and high comorbidity rates with conditions like Oppositional Defiant Disorder and Conduct Disorder present challenges for interpretation. Our limited sample size prevented the exploration of neural underpinnings related to this heterogeneity and comorbidities. Larger-scale studies are needed to address these important questions and to increase statistical power for detecting subtle group differences.

We employed cutting-edge neuroimaging techniques and innovative analytical approaches to investigate the neural mechanisms underlying dynamic, proactive, and reactive control processes in children with ADHD. Our findings provide insights into the neural underpinnings of cognitive control deficits in ADHD, which have been poorly understood due to the limitations of traditional neuroimaging methods. This study advances the neuroscientific understanding of ADHD by focusing on inter-trial variability rather than mean activation, highlighting neural variability as a key mechanism underlying the hallmark behavioral inconsistency observed in ADHD.

We demonstrate that children with ADHD exhibited increased temporal variability and weakened spatial stability of trial-evoked brain responses in the salience and frontoparietal networks compared to typically developing (TD) children. Furthermore, TD children showed highly similar spatial patterns of neural activity, whereas children with ADHD demonstrated more heterogeneous and idiosyncratic patterns. Crucially, the temporal variability and spatial pattern similarity of neural responses were functionally and clinically relevant. In TD children, spatial stability in the salience and frontoparietal networks tracked trial-by-trial fluctuations in response time, but this effect was much weaker in children with ADHD. Moreover, neural variability measures were related to the severity of core ADHD symptoms, such as inattention and hyperactivity/impulsivity.

Our findings provide insights into the dynamic and heterogeneous neural abnormalities underlying cognitive control deficits and symptom variability in childhood ADHD. These results highlight the potential of neural variability measures as biomarkers for ADHD and as targets for innovative interventions aimed at enhancing the stability and consistency of neural activity patterns. Our study demonstrates the utility of advanced neuroimaging approaches in uncovering subtle and dynamic neural mechanisms in ADHD and other neurodevelopmental disorders, paving the way for personalized interventions and improved outcomes.

## Methods

### Participants
One hundred and seven children (9–12 years old) were recruited from the local community. Informed consent was obtained from the legal guardians of the children, and the study was approved by the Institutional Review Board of Stanford University. Ninety-eight children completed two runs of CSST in the scanner. Thirty-seven children were excluded from the analysis because of missing data, image artifacts, excessive head motion (criteria: mean frame displacement (FD) was larger than 0.25 mm, max FD was larger than 5 mm), and behavioral performance violations of the Race model (criteria: mean Go trials accuracy was <75%, stop accuracy was lower than 25% or higher than 75%)[98]. The final dataset included 26 children with ADHD (10 female, 16 male) and 35 TD children (13 female, 22 male).

### Clinical and neuropsychological assessments
Children and their guardians completed a clinical and neuropsychological assessment session. ADHD diagnosis was informed by the children's guardians and further confirmed by a clinical questionnaire administered by clinical assessors under the supervision of a clinical psychologist. ADHD with conduct disorder and oppositional defiant disorder were not excluded because of their high comorbidity rates[99]. Additional enrollment criteria for both children with ADHD and TD children included no history of claustrophobia, head injury, serious neurological or medical illness, autism, psychosis, mania/bipolar, major depression, learning disability, substance abuse, sensory impairment such as vision or hearing loss, birth weight < 2000 g and/ or gestational ages of <34 weeks. All children were right-handed with an IQ > 80. Inattention and hyperactivity/impulsivity symptoms were assessed using The Conners' Rating Scale (Parent). Participants who were under stimulant treatment had gone through a washout period of at least five half-lives of the medicine before testing.

### Cued stop-signal task
The CSST was modified from the standard SST in order to dissociate proactive control from reactive control processes (Fig. 1a)[46,49]. Each trial began with a white or green cross (Cue) in the center of the screen for 300 milliseconds, followed by a green arrow (go signal). Participants were demanded to make an accurate and speedy button press in response to the pointing direction of the green arrow within 1.5 s after the onset of the Go stimuli. Occasionally, the green arrow turned to red (stop signal), and participants had to withhold their responses when the color changed. The color of the cue indicated the probability of the stop signal in the coming trial. A green cross indicates that no stop signal would occur after the coming go signal, which is defined as a Certain Go trial. A white cross indicates that a stop signal may occur after the coming go signal (33% chance), which is defined as an Uncertain Go trial. If a stop signal is presented, the trial is defined as Stop trial. The stop-signal delay (SSD) was initialized at 200 ms and adjusted in a staircase fashion. If the participant successfully canceled a prepotent response, the SSD increased by 50 ms in the next Stop trial. If the participant failed to stop, the SSD decreased by 50 ms in the next Stop trial. Participants completed two runs of the CSST in the scanner, and each run included 32 Certain Go trials, 32 Uncertain Go trials, and 16 Stop trials with jittered inter-trial intervals (ITIs) between 1 and 4 seconds.

## Behavioral measures

Reactive control was measured by SSRT. First, we confirmed that behavioral data did not violate the main assumption of the Race Model, that the mean RT in Unsuccessful Stop (US) trials should be shorter than the mean RT in Go trials. Then, SSRT was computed using the integration method based on the Race model[18,98]: SSRT = T−mean SSD, where T is the cutoff point where the integral of the observed distribution of Go RT in the SST (or Uncertain Go RT in the CSST) equals the probability of unsuccessful stopping. Proactive control was measured by response slowing modulated by task cues, i.e. Uncertain Go RT minus Certain Go RT.

## Neuroimaging data acquisition

Imaging data were acquired on a 3.0 T GE Signa scanner using a 32-channel head coil at the Richard M Lucas Center for Imaging at Stanford University. Functional images of 42 axial slices were acquired using the multiband gradient-echo planar imaging with the following parameters: TR = 490 ms; TE = 30 ms; flip angle = 45°, FOV = 22.2 cm, matrix = 74 × 74 and in-plane resolution = 3 mm. A high-order shimming method was used prior to data acquisition to reduce blurring and signal loss arising from field inhomogeneity. High-resolution T1-weighted images were acquired using a spoiled-gradient-recalled inversion recovery three-dimensional (3D) MRI sequence with the following parameters: TR = 8.4 ms, TE = 1.8 ms, flip angle = 15°, FOV = 22 cm, matrix = 256 × 192.

## fMRI data preprocessing analysis

Image preprocessing and statistical analysis were performed using SPM12. The first 12 volumes before the task were discarded to allow for T1 equilibrium. The remaining images were then realigned to correct for head movements and underwent slice-timing correction. EPI images were registered to the MNI standard space. Data were spatially smoothed using a 2 mm FWHM Gaussian kernel as recent studies suggested that minimal spatial smoothing could enhance signal-to-noise ratio while preserving distributed pattern information[100].

## General linear model (GLM)

Conventional GLM was conducted to estimate trial-averaged neural responses elicited by task conditions in the CSST, including Certain Go, Uncertain Go, Successful Stop (SuccStop), Unsuccessful Stop (UnsuccStop), and Go error. The trial onset was locked to the Go stimulus. Six motion parameters were entered as covariates of no interest.

## Single-trial GLM

The GLMs were performed separately to estimate the activation pattern for each trial using the Least Square–Separate approach[101,102], in which the trial of interest was modeled as one regressor, with all other trials modeled as separate regressors. Specifically, each single-trial GLM included five regressors: (1) the single trial of interest from one condition, for instance, one Successful Stop trial (SS); (2) all other SS trials; (3) all the Uncertain Go trials; (4) all the Certain Go trials; (5) all the US trials. Six head motion parameters were included as regressors of no interest. Each event was modeled by defining the time of stimulus onset as a stick function and convolving this stick function with a canonical hemodynamic response function. Temporal autocorrelations were modeled with the FAST model[103]. This voxel-wise GLM was used to compute the activation associated with each trial. The β-map for each trial was used for the main analyses, including temporal variability and spatial stability analyses. Crucially, in order to minimize the impact of head movements, we removed trials in which head motion exceeded 0.5 mm framewise displacement (FD) within 4–8 seconds after stimulus onset, a method widely used in previous studies[104–106]. These exclusions accounted for 0.28% and 1.52% of the trials for TD and ADHD groups, respectively (see Supplementary Table S9 for the number of trials/pairs included in the analysis across different trial types).

## Temporal variability analysis

We first assessed the normality of the distribution of trial-evoked brain responses (β time series) (Supplementary Methods and Tables S2 and S4). Then, the β time series were fitted with the Gaussian Model per voxel and per condition, and four parameters were derived, including mean, standard deviation, skewness, and kurtosis. Given the limited number of SS and US trials, we merged these two types of trials into one 'Stop' condition in this analysis. FSL's Randomize procedure was implemented to determine statistical significance with 5000 permutations, and results were thresholded at $p < 0.05$ TFCE corrected (Threshold-Free Cluster Enhancement)[107–109]. The Randomize tool utilizes the Freedman-Lane method, which is robust against outliers and provides a more conservative framework for group comparisons[109]. Of note, the significance map generated from FSL's Randomize represents values of 1 minus p.

## Spatial stability analysis

We utilized RSA employing both searchlight and regions of interest (ROI) methods[59,61] to examine spatial stability during task performance. For searchlight analysis, β-maps were obtained from a cubic region of interest containing 125 surrounding voxels across each participant's whole brain (as in Zheng et al.[76], Viganò and Piazza[110], Xu et al.[111], and Gao et al.[112]). To calculate pattern similarity, we correlated activity vectors of any given pair of trials using Pearson correlation. Subsequently, we transformed these similarity scores into Fisher's z-scores and compared them between conditions to quantify proactive and reactive control. ROI-based analyses were conducted similarly, except that the Pearson correlation was computed using voxels specifically selected from the chosen ROI. Of note, we excluded any pairs presented in the same run from the calculation of pattern similarity to avoid any autocorrelation issues. There were no significant differences in task performance across runs between TD children and children with ADHD (Supplementary Tables S10 and S11).

Spatial stability of trial-evoked brain responses was calculated in Uncertain Go, Certain Go, and Stop trials for each participant, and then between-group comparisons were conducted to examine whether the spatial stability was influenced by the disorder. Proactive control was quantified by the contrast of within-condition pattern similarity between Uncertain Go and Certain Go trials, as the sole distinction between these two conditions lies in the inclusion of a proactive control component in Uncertain Go but not in Certain Go trials. Similarly, reactive control was quantified by contrasting Successful Stop trials with Uncertain Go trials, given that only the former incorporates a reactive component (stopping). The contrast maps were used for both within-group analysis and between-group comparisons. Statistical significance was assessed using FSL's 'randomize,' as previously described. Specifically, for the within-group analysis, contrast maps were tested against zero using FSL's randomize, where the sign of the map was randomly permuted during each iteration. In the between-group comparison, group labels were randomly swapped in FSL's randomize to test for differences between the groups.

## ICP and its relation to trial-wise RT

We developed an "ICP" index in each Certain and Uncertain Go trial by quantifying the similarity between trial-evoked brain response pattern on Go trial and a subject-specific "template" brain response pattern of inhibitory control (Fig. 1f). First, we created an activation pattern template of inhibitory control by averaging trial-evoked activation patterns (i.e., single trial β maps) across all successful stop trials for each participant. Then, we computed the similarity of activation patterns between the inhibitory control template and each go trial, resulting in a time series of correlation coefficients along all the go trials. Next, we correlated the resulting time series of pattern similarity with the RT across go trials, using the searchlight approach. The correlation coefficient maps were z transformed and subsequently used

for group-level comparison. Statistical significance was tested using FSL's 'randomize' like previously mentioned.

## Group-specific inter-subject similarity analysis

We developed a group-specific inter-subject similarity index to characterize the extent to which a participant's brain activation pattern is similar to the rest of the participants from the same group relative to the participants from the other group[113]. For each participant, we averaged all other participants' β-maps from the same group (ADHD or TD) for each condition. Each participant's β map was then correlated with the average maps either from their own group or from the other group, resulting within-group inter-subject similarity map and between-group inter-subject similarity map, respectively. The group-specificity inter-subject similarity index was calculated by subtracting the between-group inter-subject similarity map from the within-group inter-subject similarity map. We compared these pattern similarity measures between conditions, similar to a within-participants analysis, to investigate whether children with and without ADHD exhibited group-specific neural activity patterns related to proactive and reactive control. This analysis was conducted in a whole-brain searchlight approach. The significance test was conducted using a permutation test implemented in FSL randomize, as previously mentioned.

## Temporal variability and spatial pattern similarity in association with clinical symptoms

To investigate whether temporal variability and spatial pattern similarity could explain individual differences in core symptoms of ADHD, we conducted an ROI-based correlation analysis. Our focus was on the core regions implicated in cognitive control (see the below ROI in the Method part). Temporal variability of brain responses was extracted from independent defined ROIs, and clinical symptoms, including hyperactivity and inattention scores, were assessed using Conners' Rating Scale. Spearman's correlation was employed to evaluate their relationship. Regarding spatial pattern similarity and its link to clinical symptoms, our interest lies in examining the presence of ADHD-like neural activity patterns supporting proactive and reactive control. Thus, we conducted an ROI-based analysis to investigate whether the similarity to the averaged spatial patterns of the TD group was associated with clinical symptoms of ADHD.

## ROI analysis

Key ROIs from the core cognitive control system were used to test the hypotheses. ROIs were made of spheres with a radius of 6 mm, whose centers are defined by the peak coordinates from a meta-analysis of fMRI studies on inhibitory control[30]. The ROIs included the right anterior insula (rAI, MNI: $x = 38$, $y = 20$, $z = -4$) and presupplementary motor cortex (preSMA, MNI: $x = 10$, $y = 14$, $z = 58$) from the SN, and the right dorsolateral prefrontal cortex (rdlPFC, MNI: $x = 42$, $y = 38$, $z = 20$) and right posterior parietal cortex (rPPC, MNI: $x = 40$, $y = -46$, $z = 46$) from the FPN.

## Reporting summary

Further information on research design is available in the Nature Portfolio Reporting Summary linked to this article.

## Data availability

Original data reported in this study is available at https://openneuro.org/datasets/ds005899. Source data are provided with this paper.

## Code availability

Functional MRI data preprocessing and statistical analyses were performed on the SPM12 and FSL 6, and Matlab 2020. Code to analyze the data is available at https://osf.io/5apc8.

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

## Acknowledgements

This research was supported by National Institutes of Health MH105625 (W.C.), MH124816 (W.C.) MH121069 (V.M.), EB022907(V.M.), and NS086085 (V.M.), NARSAD Young Investigator Award (W.C.), Stanford Maternal and Child Health Research Institute Grant (W.C.), Stanford University Department of Psychiatry Innovator Grant (W.C.), and Arizona Alzheimer's Consortium & State of Arizona DHS Pilot Grant (L.Z.). We thank Rachel Rehert and Ahmad Belai AI-Zughoul for their assistance with data collection.

## Author contributions

Study design: W.C., V.M.; data collection: W.C., K.D., S.W.; data analysis: Z.G., W.C., L.Z.; manuscript drafting: Z.G., W.C.; manuscript editing: Z.G., S.W., S.H., V.M., W.C.

## Competing interests

The authors declare no competing interests.
