## [Transparent Peer Review file · Nature Communications]

Reduced temporal and spatial stability of neural activity patterns predict cognitive control deficits in children with ADHD

Corresponding Author: Dr Zhiyao Gao

Version 0:

Reviewer comments:

Reviewer #1

(Remarks to the Author)

This is a very interesting study into the neural correlation of dual cognitive control- proactive and reactive control mechanisms- among those with parent reported ADHD symptoms. By modeling single-trial variability in neural response, the authors reveal more variability and more extreme values amongst those with ADHD in both. The temporal and spatial correlates of cognitive control. This stands in contrast to a more conventional approach – a GLM examining group means, with no consideration of other distributional parameters- which showed no diagnostic differences. This finding of more variable neural activity in ADHD sits well with the current literature which reports quicker shifts between different brain states in ADHD, along with more variability and ‘noise’ in some electrophysiological indices. The broad concept of more variability in brain regions recruited to accomplish cognitive tasks has also been reported amongst those with other neurodevelopmental conditions.

The paper’s strengths are obvious. It is a very sophisticated analytic approach that shows how much can be missed when simply assuming in GLM that the normal distributions being examined are identical. The work extends the examination of proactive control by this group, by parsing its neural correlates. The paper is clearly and concisely written and makes complex methods accessible.

Despite the many strengths I have several questions.

1) Why did the authors focus on a form of proactive control that doesn’t differ behaviorally between those with ADHD and those without the diagnosis? This group have provided some very elegant demonstrations of how ADHD is associated with challenges in proactive control that is related to prior performance (eg post-error monitoring and changes in behavior). Yet, here they examine context driven proactive control- the only form that is intact in ADHD. This is perhaps puzzling as it seems that they also have imaging data on the SSRT that could be used to parse neural correlates of performance and anticipation-driven proactive control that are impaired in ADHD. This leads to specific questions.

A) Why focus on a form of proactive control that is intact in ADHD? If context-driven proactive control is intact in ADHD, could the authors explain why it is important that its neural correlates are more variable?

B) Do they have imaging data on performance and anticipation proactive control from the SSRT that they could use?

C) If not, is there some way to extract a measure of performance related proactive control from the CCST?

D) The authors should also probably temper down some statements that they are probing the neural basis of impaired proactive control. It seems incorrect to state < The reduced spatial stability in children with ADHD during proactive control provides novel insights into the neural mechanisms underlying the difficulties in maintaining and implementing preparatory control strategies often observed in this population.>. The form of proactive control being examined is intact.

2) Issues around the analyses of the distribution. The paper relies upon contrasting the properties of normal distributions (standard deviations, kurtosis, skew) in both those with and without ADHD. Thus it is critical that trial-evoked neural

responses should be normally distributed in each group, considered separately. Currently only data for the normality of the two groups combined is given in the supplement. However, what is needed is data for the ADHD and unaffected groups separately. This is particularly important as the group sizes are quite different with only 25 in the ADHD group. In short:-

- A) What percentage of trial-evoked neural response are normally distribution for each group, considered separately?
- B) What was done with the non-normally distributed responses? Were they omitted? This is important as up to 23% were non-normally distributed.
- C) The ADHD distribution has only 25 datapoints. It is easy to imagine that many of the findings could be driven by one or two outliers. Is this the case? For example--are data from one or two individuals having an undue influence of the distributional parameters? Could this be looked at (eg Cook's leverage?)
- D) How would the authors respond to those who would say that it would be better to drop parametric analyses entirely, and focus instead on non-parametric approaches if the neural responses differ so much in the properties of the their normal distributions?

3) While the authors have considered in-scanner motion through using thresholds and covarying for residual motion, there is still a concern that in scanner-motion could explain much of the pattern of findings, which or similar to what you'd expect if in-scanner motion showed subtle group differences. I suggest one further robustness analysis. What are the results if they do a matched analysis 1:1 – focusing on in-scanner motion. Demonstrating differences in the neural responses in groups of the same size, and matched on perhaps the most critical confound would greatly add to confidence in the results.

Minor points

- 4) What is the measure of spatial stability on the graphs? If it's a measure spearman's r- the values seem pretty low (albeit significant).
- 5) Some of the analyses seem different expressions of the same underlying finding. For example, if the ADHD neural responses show more variability and more extreme values, then is it almost inevitable that pairwise similarity for this group will be less than pairwise similarity for unaffected individuals drawn from a less variable (more perfectly normal) distribution? For example for the hypothesis: < We hypothesized that a more homogeneous group would 639 show greater within-group spatial similarity compared to a less homogeneous group.>. To what extent is this inevitable?
- 6) It seems that the SS and US were combined in some analyses but not others? For example, in the temporal variability analyses they were combined- but completely omitted in the spatial, but yet considered in the single trial modeling?
- 7) There is so much data removed in the ADHD group, that it raises the question of how this group differ from those excluded. How did the groups passing vs failing QC differ- and how were they similar?

Reviewer #2

(Remarks to the Author)

This study examined the temporal and spatial stability/variability of neural activity patterns during a cued stop-signal task (CSST) in relation to cognitive control deficits in children with ADHD in contrast to TD children. For cognitive control, reactive control and proactive control are conceptually defined as the stop-signal reaction time (SSRT) and response slowing (the difference, in reaction time, between cued certain go and cued uncertain go conditions) with a dual control model, respectively. The stability and variability of task evoked neural activity were examined at the single-trial level using representational similarity analysis (RSA) which is an ideal solution. The sample size (9-12 years old children with ADHD and TD children, 107 in total) is relatively large. The authors are cautious in making a causal conclusion from the patterns of neural response to behavior by examining the relationship between brain response and clinical symptoms. However, the major question about the causal relationship between the brain and the behavior is still unanswered by the association methods adopted here. In addition, is there possible a third latent variable(s) driving both neural and behavioral responses? Overall, this is a solid study with a clear hypothesis stated. However, some methodological questions need to be clarified. See below for major and minor critiques line by line.

1. The novel metrics of temporal variability and spatial stability are intriguing, but their validity and underlying mechanisms require further substantiation. Critical questions include: What specific neural activities do these metrics reflect? How do they relate to traditional measures like activity and connectivity? Are there correlations among these metrics? Addressing these points would strengthen the validity and interpretability of these measures as potential ADHD biomarkers. For example, Line 243-245. To ask further, what is the underlying behavior or neural mechanisms of the "unstable recruitment of core cognitive control systems during task performance"? Line 247-249. It is unusual to find no significant group difference in terms of brain activation during in any task conditions.

2. Lacking of behavioral effects between ADHD and TD groups of the fMRI samples (e.g., in Line 190-205) needs to be carefully discussed. It is hard to argue that the neural results, without the behavioral evidence, are sufficient to provide the support to the hypothesis. Why there was no significant group difference in temporal variability of trial-by-trial brain activation during the CSST (i.e., in Line 213-222)? This is again an issue as the last critique.

3. The whole-brain analysis reveals inconsistencies in analytical approaches across different metrics, potentially compromising the coherence and interpretability of results. Some analyses directly compare ADHD and TD groups across conditions, while others examine proactive and reactive control contrasts separately within each group, or present results for each condition separately for ADHD and TD groups. This lack of uniformity in analytical logic makes it challenging to draw comprehensive conclusions. Moreover, the analysis of stop trials is inconsistent, sometimes including all stop trials and other times focusing only on successful stops. This inconsistency further complicates interpretation.

Abstract:

4. The clarity of abstract needs to be improved. For instance, the key concepts of proactive and reactive cognitive control

need to be explained. The statements that not reflect the results accurately need to be justified, i.e., "Typically developing (TD) children exhibited stable neural response patterns for efficient proactive and reactive dual control mechanisms."

Introduction:

5. Line 83-85 and Line 115-117. Reference(s) should be provided for this statement, i.e., for the "few studies".

6. Line 116. The functional role of the frontoparietal network (FPN) and the salience network (SN) in cognitive control has been examined by Wu et al., (2020) by conducting an activation-likelihood-estimation-based large-scale meta-analysis, with the fMRI studies using stop-signal task included.

Wu et al, (2020). The functional anatomy of cognitive control: A domain-general brain network for uncertainty processing. DOI: 10.1002/cne.24804

Methods:

7. The participant sample reporting raises several concerns. Of 107 recruited children, only 93 (53 ADHD + 40 TD) are accounted for initially. The final sample of 26 ADHD and 53 TD indicates substantial exclusions, particularly in the ADHD group. A clear, detailed report of exclusions is crucial, specifying the number excluded for each reason in each group. The high exclusion rate in the ADHD group raises concerns about potential bias. Does excluding behavioral outliers inadvertently remove participants with more severe ADHD symptoms? This could significantly impact the study's representativeness and generalizability. Additionally, the rationale for excluding participants with stop accuracy higher than 75% is unclear and needs justification. This criterion might potentially remove high-performing ADHD participants, further skewing the sample.

8. The sample size of 26 ADHD and 35 TD children raises concerns about statistical power. While acceptable for fMRI studies, it's relatively small and could limit the robustness and generalizability of findings. It's unclear whether a priori power analysis was conducted to determine the ideal sample size for detecting the expected effects.

9. Clear details about the CSST paradigm, particularly the trial timeline, need to be reported. This is crucial for single trial extraction accuracy and potential signal contamination between trials. Providing estimated hemodynamic response curves for each condition could help address this concern (see "Wu, T., Wang, X., Wu, Q., Spagna, A., Yang, J., Yuan, C., Wu, Y., Gao, Z., Hof, P. R., & Fan, J. (2019). Anterior insular cortex is a bottleneck of cognitive control. *NeuroImage*, 195, 490–504" as an example). Additionally, reporting the number of trials included in the final analysis for each condition is necessary.

10. Details of GLM onset extraction needs to be clarified. With 2-3 stimuli events per trial (cue, green arrow, red arrow), it's unclear which event the trial onset is locked to. Is modeling these multi-event trials with a single stick function appropriate? Additionally, the treatment of go trials without responses in GLM construction needs explanation. In addition, Line 831-834. For the (3) Uncertain Go trials, (4) Certain Go trials, and (5) US trials, they all involve a motor response in common. Can you justify that there is no need to model (extract out) the motor response related activation?

11. The study includes numerous univariate ROI analyses, raising concerns about multiple comparisons. It's crucial to report whether any correction for multiple comparisons was applied to these analyses.

12. It's unclear whether the group-level whole-brain analyses controlled for important covariates such as gender and age. This information is crucial for understanding the robustness of the findings. Additionally, the study doesn't specify whether ADHD comorbidities were considered or if analyses were conducted for different ADHD subtypes. Given the heterogeneity of ADHD, these factors could significantly impact the results and their interpretation.

Results:

13. The presentation of results could be optimized for clarity and comprehension. For Figures 2a&b, the group difference in STD (ADHD > TD) is very different from the hypothesized regions/networks. It's challenging to discern whether activations fall within ROIs. Overlaying ROI contours on whole-brain activation results would be beneficial. The inconsistency in brain surface inflation between Figure 1h (ROI presentation) and activation results hinders comparison. In addition, lists of activation foci/clusters are missing, which would aid in precise interpretation. In addition, can you also make the plot of TD > ADHD to see the regions with more temporal variability in the TD group?

14. Several figures require improvements: Figure 4a lacks a color bar; Figure 4c wasn't mentioned in the main text; Figure 1 panels were referenced inconsistently, with some mentioned even after Figure 3; Figure 1g is confusing, as both TD and ADHD example subjects are labeled "s1". Line 346. Should it be Figure 1f? Line 431. Figure 1g instead of 1f? Line 455-457. The scatter plot (Figure 6) does not indicate a correlation of .436. It is misleading since you have a regression model with other variables included as covariates. Omit this figure?

15. Line 303-304. This sentence is not clear: i.e., the "ADHD recruit highly variable distributed brain regions." Do you imply the regions shown in Figure 3's top panel? For Figure 3, three words of "weaker", "weak", and "weakened" are used. Do they express different meanings?

16. Line 312-314. The operationally defined contrasts for proactive control (Uncertain vs. Certain Go trials) and reactive control (Successful Stop vs. Uncertain Go trials) need to be justified/explained here. The two conditions for the contrast of reactive control are different in terms of motor response. Also, they involve a common condition, the uncertain go trials. Therefore, the two contrasts/measures are not totally independent.

17. Line 318-320. This sentence may not be accurate: "whereas children with ADHD showed significant spatial stability in less distributed brain regions, including visual areas and right PPC ...". In addition, the results/inferences of Figure 3b for the

ADHD and TD groups are not supported by the testing of group difference. Therefore, it should be cautious in make such inferences/conclusions solely based on visual inspection.

18. Line 328-332. If you treat this study as hypothesis driven, ROI can be considered to be reported first with the whole brain results as data-driven further examination or for the visualization of the brain responses.

Discussion:

19. The subtitles of the Discussion are very informative. However, this section has some redundancy. Some reviews and materials that have been covered in the Introduction and Results are repeated extensively here, e.g., lines 494-510, 433-551, 632-640. In addition, the discussion section would benefit from clearer distinctions between experimental results, interpretations, and speculations. More explicit language should be used to differentiate these elements. The potential clinical applications of the findings for ADHD diagnosis and treatment could be discussed in more detail, while clearly stating the need for further validation. The discussion should more explicitly articulate how this study advances neuroscientific understanding of ADHD and how it challenges or complements existing ADHD models.

20. The discussion lacks a comprehensive examination of the activity mechanisms and functional modes of brain networks and key regions. There's insufficient unified discussion on brain mechanisms related to temporal and spatial variability, leading to a lack of a coherent framework for understanding the roles of CON and FPN in ADHD. In addition, the analysis revealed activations in areas beyond the predefined ROIs, such as the visual cortex and thalamus. The potential roles of these regions in ADHD should be addressed.

21. The discussion mentions three potential sources of inter-subject similarity: variability in symptom profiles and cognitive deficits, diversity of cognitive strategies during task performance, and inherent instability of neural dynamics. Interestingly, these hypotheses seem testable using the existing dataset. These additional analyses could provide valuable insights, strengthening the study's conclusions and offering a more comprehensive understanding of inter-subject variability in ADHD. Consider incorporating these analyses or discussing why they weren't pursued.

22. The study's focus on cognitive control deficits in ADHD raises important questions about the specificity and generalizability of the observed changes in temporal and spatial variability. Cognitive control deficits are common across various psychiatric disorders, not just ADHD. We recommend discussing: Whether the observed patterns of temporal and spatial variability are specific to ADHD or might be seen in other disorders with cognitive control deficits? The task-specificity of these new measures. How might they generalize across different cognitive control paradigms? The potential for these measures to differentiate ADHD from other disorders with similar cognitive deficits, and the broader applicability of this analytical approach to studying neural dynamics in other psychiatric conditions.

Reviewer #3

(Remarks to the Author)

The present manuscript provides novel insights into trial-based variability in ADHD during inhibitory control. In a comprehensive set of analyses, the authors show for example:

- Greater temporal variability of trial-based brain activation in ADHD
- Greater spatial variability in ADHD in various inhibition-related brain regions
- Weaker association of higher RT with increased "similarity" between go trials and successful stop trials in ADHD in inhibition and motor-response related brain regions
- Greater inter-subject spatial variability of brain activation within the ADHD group

The manuscript is generally well written, the study is well executed and the statistics appear to be sound. While generally excited for the analysis approach the authors present, my enthusiasm is somewhat reduced by two aspects: 1) reporting marginally significant findings and reporting uncorrected fMRI results in a study of this quality and rigor seems ill placed 2) a more nuanced discussion of the findings seems warranted, there were instances of too bold or broad statements that do not entirely reflect the reality of the presented data.

My comments are detailed below.

Overall:

- The manuscript would benefit from thorough proof reading, there were a number of typos

Results:

- Line 207 ff: stating that children with ADHD have poor reactive control and less stable performance but preserved proactive control seems a bit bold given that the only significant findings were observed in the full sample and were not replicated in the fMRI subsample
- Line 270 ff: Please refrain from reporting "trend level" statistics. While I understand this is tempting, a non-significant result at a given threshold remains non-significant. Further, given the vast number of tests run on this dataset, setting critical alpha to .05 is already very liberal (also see my question regarding correction for multiple testing).
- Line 320 ff: Similarly, please do not interpret a "weakened spatial stability in the distributed brain systems during proactive and reactive control in comparison to TD children" based on $p < .01$ uncorrected results. I suggest to omit this completely. At the very least, specify this as an explorative analysis and discuss the findings accordingly and carefully.

Discussion:

- Line 610 ff: "Notably, we found that the ICP index in key regions of the salience and frontoparietal network nodes tracked trial-wise RT fluctuations in TD children." This could be misunderstood. The authors did not correlate with a measure of RT fluctuation, but with RT in general. This is used a few times across the manuscript. I do understand what the authors are trying to say, but think this could be misinterpreted.
- Line 615 ff: The interpretation of the association between ICP index and RT is cut short to an extent. While the authors state

carefully that they measure proactive control for instance by RT slowing in uncertain go vs. certain go trials, they argue in the discussion that based on the found association of higher RT with increased “similarity” between go trials and successful stop trials, this higher similarity then encodes the amount of proactive control implemented. However, the stated correlation was found for certain and uncertain go trials alike. Thus, generally suggesting that higher RT reflect overall increased proactive control. This is not warranted, as other processes and factors may contribute to RT slowing.

- Line 696 ff: “Furthermore, we found that children who exhibited more ADHD-like spatial activation patterns during proactive control had more severe inattention symptoms” this statement seems a lot too broad given that this finding was based on an ROI analysis and significant for the rAI only. Please also adjust the following discussion accordingly.

- A brief discussion on the fact that the authors did not find group differences in the mean activation in standard GLM analysis of this task, along with implications of this, would be appreciated. This would be an important point for discussion given the partly heterogenous evidence of standard fMRI studies in ADHD and further highlight the relevance of this new approach.

Methods:

- Could the authors comment on their decision to exclude participants with “outlier” behavioral data and how the cutoffs for that were selected?

- How long was the task in total for each run?

- The authors excluded any pairs presented in the same run from the calculation of pattern similarity to avoid any autocorrelation issues. How could this have affected results differentially for TD and ADHD groups given that ADHD participants second run may have been impacted by impairments in sustaining attention?

- It is not clear how precisely randomize was used for the different types of analyses. It seems, sometimes comparisons were run between groups, sometimes between conditions or tested against zero.

- The description of ROI definition should be revised. The first sentence is not very clear.

- More importantly, ROI definition based on resting state data in a separate dataset seems suboptimal, if the authors cannot give a clearer rationale for this. ROI selection based on published meta-analytic results or at least task-based effects in previous studies seems more warranted.

- Is there a rationale for the 125 voxel searchlight size?

- The authors seem to correct for multiple testing within each analysis (FDR corrected p values are reported). I am wondering if they also correct globally for the large amount of tests they performed overall on this dataset?

Figure 4: The figure title and captions include ICP modulation of RT, I suggest to phrase this more carefully. Correlational analyses cannot speak to causality and we do not know if the recruitment of brain regions in a fashion that is similar to successful go trials actually modulates RT, or of both the spatial pattern in brain activation and the RT are modulated by another, underlying process or factor.

Reviewer #4

(Remarks to the Author)

Version 1:

Reviewer comments:

Reviewer #3

(Remarks to the Author)

Overall, the authors have addressed my concerns thoroughly and thoughtfully. Only few minor points remain.

General:

- Please check language again. For instance, “significant” instead of “significantly” is used in various instances across the manuscript.

Results:

- Line 218ff: if the authors made any changes to their statement that children with ADHD have poor reactive control and less stable performance but preserved proactive control, this is not tracked/highlighted. I still think the statement is too bold as it is currently, given that the findings were only significant in the full but not the fmri sample.

- In terms of structure, adding the ROI analyses directly after the respective whole brain level analyses might be helpful and more in line with the figures.

- Line 371 ff: “Children with ADHD showed less distributed effect in the supplementary motor area and anterior cingulate cortex in Certain and Uncertain Go trial” please clarify that this is by visual inspection/comparison, not statistical comparison.

Reviewer #4

(Remarks to the Author)

I co-reviewed this manuscript with one of the reviewers who provided the listed reports. This is part of the Nature

Communications initiative to facilitate training in peer review and to provide appropriate recognition for Early Career Researchers who co-review manuscripts.

For this rebuttal review, as my senior co-reviewer declined to participate in the review process, I have discussed with the editor and was advised to provide my independent assessment of the revised manuscript. I have reviewed the revised manuscript and the authors' responses to the previous comments.

Overall, I appreciate the authors' extensive efforts in addressing the previous concerns and improving the manuscript. The proposed metrics based on single-trial BOLD signals (temporal variability and spatial stability) offer novel perspectives for understanding neural dynamics in ADHD and similar disorders, showing promising potential for future research. The additional control analyses have enhanced the study's rigor and reliability. While the manuscript has been substantially improved, there are several remaining issues that require attention:

(1) There is a notable disconnect between the network-level framework emphasized in the Introduction and Discussion (focusing on SN and FPN), and the region-based reporting in the Results section. This inconsistency may pose challenges for readers less familiar with these networks. A clearer mapping between individual regions and their corresponding networks would enhance the manuscript's accessibility and coherence.

(2) Several aspects of figure presentation require improvement. While Figures 2-5 now include colorbars as requested, they lack proper labeling of the metrics being displayed. The correspondence between whole brain results and ROI-based results remains difficult to interpret due to inconsistent scaling between whole brain colorbars and ROI-based y-axes, and unclear mapping of ROI locations to whole-brain results. Additionally, the inconsistent use of colormaps across figures lacks clear justification, and Figure 6 needs proper legend explaining the color-coding of dots. These presentation issues significantly impact the interpretability of the results.

(3) While the Discussion section has been revised, it still contains substantial overlap with the Results and Methods sections, and has actually increased in length. This makes it more challenging to follow the logical flow of arguments. The discussion of within-group spatial similarity and its relationship to heterogeneity in ADHD symptoms and cognitive deficits requires more careful consideration, as these interpretations currently lack direct supporting evidence from the results.

(4) The specification of event onsets in the fMRI GLM analysis, while included in the response letter, should be explicitly detailed in the main text. The current description is too brief for replication purposes. More detailed information about event onset definitions, timing parameters, and relevant processing steps would enhance the reproducibility of the study. These modifications would further strengthen the manuscript and improve its utility for the research community. Despite these concerns, I believe this study makes a valuable contribution to our understanding of neural dynamics in ADHD, particularly through its innovative methodological approach

Reviewer #5

(Remarks to the Author)

This is a novel and well executed study examining proactive and reactive control in children with ADHD compared to neurotypical children. The authors utilised a novel computational method to explore neural temporal and spatial stability/instability in ADHD of trial-evoked activity in a stop signal reaction time task. Via this technique authors could explore temporal stability by examining measures such as standard deviation and kurtosis in neural activity and spatial stability using representational similarity analysis (RSA). Overall this study provides new exciting insights into cognitive control in ADHD. My questions regarding the analysis and meaning of the variables such as the standard deviation and kurtosis were addressed by responses to previous reviewers. I have a number of minor remaining questions but overall the manuscript is an important, well executed one.

I am not sure how accurate it is to state that inhibitory dysregulation is a core feature of ADHD – this is highlighted by the relatively old references provided to support that statement. In fact many children with ADHD have no issues with inhibitory control.

It is somewhat misleading to state that participants were actually age and gender matched when the case is that there were no significant differences in groups between age and gender

What was the criterion for ADHD diagnosis? Were clinical interviews conducted and what cut-off was used for the Conners'?

Reviewer #1 (Remarks to the Author):

This is a very interesting study into the neural correlation of dual cognitive control- proactive and reactive control mechanisms- among those with parent reported ADHD symptoms. By modeling single-trial variability in neural response, the authors reveal more variability and more extreme values amongst those with ADHD in both. The temporal and spatial correlates of cognitive control. This stands in contrast to a more conventional approach – a GLM examining group means, with no consideration of other distributional parameters- which showed no diagnostic differences. This finding of more variable neural activity in ADHD sits well with the current literature which reports quicker shifts between different brain states in ADHD, along with more variability and ‘noise’ in some electrophysiological indices. The broad concept of more variability in brain regions recruited to accomplish cognitive tasks has also been reported amongst those with other neurodevelopmental conditions.

The paper’s strengths are obvious. It is a very sophisticated analytic approach that shows how much can be missed when simply assuming in GLM that the normal distributions being examined are identical. The work extends the examination of proactive control by this group, by parsing its neural correlates. The paper is clearly and concisely written and makes complex methods accessible.

Despite the many strengths I have several questions.

1.1 Why did the authors focus on a form of proactive control that doesn’t differ behaviorally between those with ADHD and those without the diagnosis? This group have provided some very elegant demonstrations of how ADHD is associated with challenges in proactive control that is related to prior performance (eg post-error monitoring and changes in behavior). Yet, here they examine context driven proactive control- the only form that is intact in ADHD. This is perhaps puzzling as it seems that they also have imaging data on the SSRT that could be used to parse neural correlates on performance and anticipation-driven proactive control that are impaired in ADHD. This leads to specific questions.

A) Why focus on a form of proactive control that is intact in ADHD? If context-driven proactive control is intact in ADHD, could the authors explain why it is important that its neural correlates are more variable?

B) Do they have imaging data on performance and anticipation proactive control from the SSRT that they could use?

C) If not, is there some way to extract a measure of performance related proactive control from the CCST?

D) The authors should also probably temper down some statements that they are probing the neural basis of impaired proactive control. It seems incorrect to state < The reduced spatial stability in children with ADHD during proactive control provides novel insights into the neural mechanisms underlying the difficulties in maintaining and implementing preparatory control strategies often observed in this population.>. The form of proactive control being examined is intact.

Response: We thank the reviewer for the positive feedback on our current and previous work. The reviewer is correct that, in our previous study (Cai et al., 2023), we did not find significant group difference (ADHD vs. TD) in context-driven proactive control (driven by explicit context cue). However, different underlying mechanisms may contribute to response slowing, for instance effortful proactive control, inattention and fatigue. It is unknown whether similar context-driven proactive control, measured by RT difference between Uncertain and Certain Go trials in the CSST, shares the same or different underlying neural mechanisms. More importantly, Uncertain and Certain Go trials in the CSST allow us to dissociate proactive control (Uncertain – Certain Go) from reactive control (Stop – Uncertain Go) processes, so the single trial model and representational similarity analysis can probe temporal and spatial stability of neural representation associated with proactive and reactive control processes separately. The classic SST has only Go and Stop trials, both involving proactive control, thus limiting our ability to dissociate neural underpinning of the two control processes. In our previous study, we found significant deficit in proactive control associated with performance monitoring and anticipation (driven by trial history) in children with ADHD. Anticipation-driven proactive control was estimated using the Dynamic Belief Model, which can only be applied in the classic SST but not the CSST. We are pursuing a separate study to investigate neural mechanisms underlying anticipation-driven proactive control using the SST.

- A. Behavioral measures, such as accuracy and RT, are the output from a sequence of cognitive and motor processes. Different neurocognitive mechanisms may underlie similar behavioral performance. Many previous studies in clinical neuroscience have shown significant between-group difference in neural measures while behavioral performance were not significantly different (Bergeron et al., 2020; Iravani et al., 2021; Musella & Weyandt, 2023; Suskauer et al., 2008; Yap et al., 2021). In most clinical behavioral and functional neuroimaging studies, proactive control is quantified using response slowing. However, response slowing could be driven by factors other than effortful proactive control, such as attention or motivation. If effortful proactive control underlies response slowing in children with ADHD and TD children, we will expect to see similar temporal and spatial stability of neural representation and similar inter-trial associations with RT fluctuation in both groups. However, if other internal processes, such as inattention or poor motivation, contributes to response slowing in different scales between children with ADHD and TD children, we will expect to see between-group differences in temporal and spatial stability of neural representation and inter-trial association with RT. Indeed, that is what we found.*
- B. In our previous study (Cai et al., 2023), anticipation-driven proactive control was estimated using the Dynamic Belief Model, which can only be applied in the classic SST but not the CSST. Nevertheless, we are pursuing a different study to investigate neural mechanisms underlying anticipation-driven proactive control using the SST.*

C. *The CSST was specifically designed to dissociate context-driven proactive control (Uncertain – Certain Go) from reactive control (Stop – Uncertain Go) processes. While the Dynamic Belief Model cannot be directly applied in the CSST to study anticipation-driven proactive control, we can potentially probe proactive control associated with performance monitoring. However, the context cue also modulates proactive control process and thus can interfere with proactive control induced by negative feedback from last unsuccessful stop trials. Therefore, we decided not to investigate proactive control associated with performance monitoring in the CSST. However, we are pursuing a separate study to examine neural mechanisms underlying proactive control associated with performance monitoring in the classic SST in children with ADHD using a different dataset.*

D. *In line with the reviewer's suggestion, we have tempered our statements regarding deficits in proactive control and revised the manuscript accordingly. Please see below:*

“The reduced spatial stability in children with ADHD during proactive control provides novel insights into neural mechanisms underlying implementation of preparatory effort in this population.”

1.2 Issues around the analyses of the distribution. The paper relies upon contrasting the properties of normal distributions (standard deviations, kurtosis, skew) in both those with and without ADHD. Thus it is critical that trial-evoked neural responses should be normally distributed in each group, considered separately. Currently only data for the normality of the two groups combined is given in the supplement. However, what is needed is data for the ADHD and unaffected groups separately. This is particularly important as the group sizes are quite different with only 25 in the ADHD group. In short:-

- A) What percentage of trial-evoked neural response are normally distribution for each group, considered separately?

- B) What was done with the non-normally distributed responses? Were they omitted? This is important as up to 23% were non-normally distributed.

- C) The ADHD distribution has only 25 datapoints. It is easy to imagine that many of the findings could be driven by one or two outliers. Is this the case? For example---are data from one or two individuals having an undue influence of the distributional parameters? Could this be looked at (eg Cook's leverage?)

- D) How would the authors respond to those who would say that it would be better to drop parametric analyses entirely, and focus instead on non-parametric approaches if the neural responses differ so much in the properties of the their normal distributions?

*Response: We appreciate the reviewer's suggestion and have now provided the percentage of trial-evoked neural responses that are normally distributed for each group separately (ADHD and unaffected groups). This information is included in the revised Supplementary Materials **Table S2~S4.***

Regarding the handling of non-normally distributed responses, excluding participants based on normality checks alone is challenging because the identification of outliers was inconsistent across voxels and trial types, making it unfeasible to exclude participants systematically in a whole-brain analysis. Instead, we focus on a set of pre-defined hypothesis-driven ROIs, with a focus on the key nodes of the inhibitory control network.

To address the concern that group differences might be driven by these low-percentage potential outliers, we conducted non-parametric permutation tests using FSL's *randomize* for group comparisons instead of traditional parametric analysis. *Fsl's randomize* utilizing the Freedman-lane method, this approach is robust against outliers and provides a more conservative framework for group comparisons (Winkler et al., 2014). We conducted non-parametric permutation tests for both the within group group-level analyses and between-group comparisons. This methodology is now clarified in the revised manuscript.

Additionally, following the reviewer and other reviewer's suggestions, we created 4 spherical ROIs based on a meta-analysis of inhibitory control fMRI studies within the frontoparietal and salience networks. We conducted linear regression analyses, treating diagnosis as the independent variable and brain measures (e.g., temporal variability) as the dependent variable, while controlling for age, gender, and head motion as covariates of no interest. Cook's distance was employed to mitigate the influence of potential outliers. The new analyses replicated our findings and suggested our findings were not driven by potential outliers. Please see results in **Supplementary Results and Table S5~S8**.

“Robustness with respect to age, gender, and head motion

To assess the robustness of our findings against potential confounding factors, including age, gender, and head motion, we conducted additional ROI-based analyses for between-group comparisons while controlling for these variables. We selected ROIs implicated in cognitive control from salience (SN), encompassing the right anterior insula (rAI) and presupplementary motor cortex (preSMA), and frontal-parietal regions (FPN), encompassing right dorsal lateral prefrontal cortex (rdLPFC), and right posterior parietal cortex (rPPC) (see Methods and Figure 1h).

Increased temporal variability of trial-evoked brain responses in children with ADHD

In comparison to TD children, children with ADHD exhibited significantly higher standard deviation in the preSMA, rDLPFC, and rPPC during Uncertain and Certain Go trials, in all four ROIs during Stop trials after accounting for age, gender, and head motion (all $ps < 0.002$, FDR corrected). Children only showed lower standard deviation in the rAI during Uncertain and Certain Go trials ($ps < 0.001$, FDR corrected), see **Table S5**. Moreover, children with ADHD had significantly higher kurtosis than TD children in the

rAI during Uncertain Go trials, and the rPPC during Certain Go trials (all $p < 0.006$, FDR corrected) after controlling for age, gender, and head motion. FDR correction was applied to four ROIs, see **Table S6**.

Weaker spatial stability of trial-evoked brain responses in children with ADHD

In comparison to TD children, children with ADHD exhibited significant lower spatial stability in the preSMA and rPPC across all three trial types ($p < 0.05$, FDR corrected) after controlling the effect of age, gender, and head motion, see **Table S7**.

Weakened association between proactive control system and behavioral regulation in ADHD

In comparison to TD children, children with ADHD exhibited a significantly weaker association between the inhibitory control pattern (ICP) and reaction time in the preSMA, rAI, and rdLPFC during Uncertain Go trials after accounting for age, gender, and head motion ($p \leq 0.015$, FDR corrected). This weaker association was also observed in the preSMA when both Uncertain and Certain Go trials were combined ($p = 0.004$, FDR corrected), after controlling for age, gender, and head motion, see **Table S8**.

Table S5. Increased temporal variability (standard deviation) of trial-evoked brain responses in children with ADHD (FDR corrected).

Brain measures	Conditions	Slope	p-value
Uncertain Go	preSMA	0.248	$p < 0.001$
	rAI	-0.390	$p < 0.001$
	rdLPFC	0.601	$p = 0.001$
	rPPC	0.373	$p < 0.001$
Certain Go	preSMA	0.214	$p < 0.001$
	rAI	-0.382	$p < 0.001$
	rdLPFC	0.579	$p < 0.001$
	rPPC	0.434	$p < 0.001$
Stop	preSMA	0.104	$p < 0.001$
	rAI	0.121	$p < 0.001$
	rdLPFC	0.127	$p < 0.001$
	rPPC	0.086	$p < 0.002$

Table S6. Increased temporal variability (kurtosis) of trial-evoked brain responses in children with ADHD (FDR corrected).

Brain measures	Conditions	Slope	p-value
	preSMA	0.210	0.445
	rAI	0.408	0.026

Uncertain Go	rdIPFC	0.104	0.445
	rPPC	0.188	0.445
Certain Go	preSMA	0.077	0.402
	rAI	0.180	0.402
	rdIPFC	0.077	0.402
	rPPC	0.318	0.015
Stop	preSMA	0.070	0.708
	rAI	0.134	0.708
	rdIPFC	0.243	0.558
	rPPC	0.141	0.374

Table S7. Weakened spatial stability of trial-evoked brain responses in children with ADHD (FDR corrected).

Brain measures	Conditions	Slope	p-value
Uncertain Go	preSMA	-0.034	0.004
	rAI	-0.025	0.119
	rdIPFC	-0.032	0.119
	rPPC	-0.028	0.011
Certain Go	preSMA	-0.011	0.039
	rAI	0.008	0.695
	rdIPFC	-0.032	0.108
	rPPC	-0.038	0.010
Stop	preSMA	-0.024	0.023
	rAI	-0.005	0.061
	rdIPFC	-0.045	0.339
	rPPC	-0.061	0.023

Table S8. Weakened association between proactive control system and behavioral regulation in ADHD (FDR corrected).

Brain measures	Conditions	Slope	p-value
Uncertain Go	preSMA	-0.069	0.009
	rAI	-0.042	0.015
	rdIPFC	0.027	0.004
	rPPC	-0.053	0.134
Certain Go	preSMA	-0.014	0.121
	rAI	-0.066	0.121
	rdIPFC	-0.040	0.121
	rPPC	-0.023	0.818

Uncertain + Certain Go	preSMA	-0.050	0.004
	rAI	-0.048	0.054
	rdIPFC	0.008	0.054
	rPPC	-0.029	0.184

1.3 While the authors have considered in-scanner motion through using thresholds and covarying for residual motion, there is still a concern that in-scanner motion could explain much of the pattern of findings, which is similar to what you'd expect if in-scanner motion showed subtle group differences. I suggest one further robustness analysis. What are the results if they do a matched analysis 1:1 – focusing on in-scanner motion. Demonstrating differences in the neural responses in groups of the same size, and matched on perhaps the most critical confound would greatly add to confidence in the results.

Response: We appreciate the reviewer's thoughtful suggestion. However, creating a matched group based on head motion while also controlling for other critical confounds, such as age and gender, proved infeasible due to limited sample size. For instance, in our attempts to match head motion, we selected ADHD participants with smaller head motion and TD children with larger head motion (could not achieve statically equivalent groups), which inadvertently resulted in the ADHD participants being older than the TD children. Nevertheless, we addressed in-scanner motion by applying stringent thresholds, excluding trials with large head motion in the first-level analysis, and including residual motion as a covariate of no interest in our between-group comparisons in both whole-brain searchlight and ROI-based analyses. We would also like to point out that we did not find significant between-group difference using conventional GLM methods, suggesting the low likelihood that the between-group difference in the single-trial model was driven by head motion. Nevertheless, we acknowledged this limitation in the discussion section.

Minor points

1.4 What is the measure of spatial stability on the graphs? If it's a measure spearman's r- the values seem pretty low (albeit significant).

Response: We apologize for the confusion. The measure of spatial stability was assessed using the Pearson's correlation of neural activity patterns between pairs of trials. We have revised the schematic figure for clarity. While the correlation values are indeed relatively low, they are consistent with those reported in previous studies employing similar approaches. Similar effect sizes have been reported in studies of episodic and working memory involving higher-order regions (Deuker et al., 2016; Favila et al., 2018; Hu et al., 2024; Miller et al., 2022; Zheng et al., 2018).

1.5 Some of the analyses seem different expressions of the same underlying finding. For example, if the ADHD neural responses show more variability and more extreme values, then is it almost inevitable that pairwise similarity for this group will be less than pairwise similarity for unaffected individuals drawn from a less variable (more perfectly normal) distribution? For example for the hypothesis: < We hypothesized that a more

homogeneous group would show greater within-group spatial similarity compared to a less homogeneous group.>. To what extent is this inevitable?

Response: We thank the reviewer for raising this point.

First, it is essential to differentiate between the measures we employed. Temporal variability assesses the standard deviation and frequency of extreme values of neural activity within each voxel, which is inherently a univariate measure. In contrast, spatial stability evaluates the consistency of neural activity patterns across multiple voxels between pairs of trials, making it a multivariate measure. While similar temporal patterns may be observed in multiple voxels, this does not necessarily result in a variation in spatial stability. For instance, voxels may have high temporal variability in task-induced BOLD signals but they may consistently engage from trial to trial, showing a stable spatial pattern.

Second, within-subject variability of neural representation is a different metric from within-group heterogeneity of neural representation. The former measures stability of brain representation across trials, whereas the latter examines whether participants use similar or different neural coding strategies. It is possible that TD children show high within-subject spatial stability of neural representation, but they may have used distinct neural coding strategy (i.e. different neural representations). Similar inquiries have been explored in other domains, such as episodic memory, movie watching, and story comprehension (Chen et al., 2017; Mochalski et al., 2024; Nguyen et al., 2019); these questions remain open in the context of cognitive control. Such investigation can provide us insights into heterogeneity of neurocognitive functions in children with ADHD.

1.6 It seems that the SS and US were combined in some analyses but not others? For example, in the temporal variability analyses they were combined- but completed omitted in the spatial, but yet considered in the single trial modeling?

Response: We apologize for the inconsistency of analysis description.

To clarify, we chose to model SS and US trials separately in the single-trial modeling in order to capture variability specifically associated with each type of trials. However, using only SS or US trials would not provide enough trials to reliably estimate a normal distribution, therefore we combined them in the analysis. While we acknowledge the limitations of this approach, it allows us to offer valuable insights and a more comprehensive understanding of temporal variability.

Following the reviewer and other reviewer's suggestion, and to improve consistency, we have now reported both temporal and spatial stability for the combined 'Stop' trials in both within-individual and cross-subject analyses in our revised manuscript, and obtained similar results. Please see below for Figure 2, Figure 3a, and Figure 5a:

a. Temporal Variability of β time series (std)

b. Temporal Variability of β time series (kurtosis)

a. Weaker spatial stability of trial-evoked brain response patterns in children with ADHD than TD children

b. Weak spatial stability of trial-evoked brain response patterns during proactive and reactive control in children with ADHD

a. Within-group inter-subject pattern similarity differs between groups

b. Within-group inter-subject pattern similarity underlies proactive and reactive control

c. Group-specific inter-subject pattern similarity underlies proactive and reactive control

1.7 There is so much data removed in the ADHD group, that it raises the question of how this group differ from those excluded. How did the groups passing vs failing QC differ- and how were they similar?

Response: We thank the reviewer for raising this important question. Most of excluded data are due to poor head motion and unwillingness to continue the experiment. While we agree that the ratio of excluded data is high, this is not uncommon in the field of clinical neuroimaging, especially for children with ADHD. Previous studies have shown similar exclusion rate (Albajara Sáenz et al., 2020; Duffy et al., 2021; van Hulst et al., 2017).

The exclusion criteria we applied are based on standard procedures widely used in the field, covering both behavioral and fMRI data. From a behavioral perspective, participants were excluded if their stop accuracy fell outside the range of 0.25 to 0.75, or if their Go accuracy was lower than 0.75. These criteria are fundamental to the assumptions of the Race model (Logan & Cowan, 1984), which is crucial for calculating SSRT and ensuring reliable measures of cognitive control. Without these thresholds, participants might employ alternative strategies, such as 'waiting' to respond, or consistently pressing the button for go stimuli, leading to unreliable accuracy and inhibition measures. These criteria are commonly used in stop-signal studies to ensure valid results. Regarding head motion criteria, participants were excluded if their mean frame displacement (FD) was larger than 0.25mm, or max frame displacement was larger than 5mm (Cai et al., 2019; Cai et al., 2021), these criteria are also commonly used in the clinical neuroimaging studies.

Reviewer #2 (Remarks to the Author):

This study examined the temporal and spatial stability/variability of neural activity patterns during a cued stop-signal task (CSST) in relation to cognitive control deficits in children with ADHD in contrast to TD children. For cognitive control, reactive control and proactive control are conceptually defined as the stop-signal reaction time (SSRT) and response slowing (the difference, in reaction time, between cued certain go and cued uncertain go conditions) with a dual control model, respectively. The stability and variability of task evoked neural activity were examined at the single-trial level using representational similarity analysis (RSA) which is an ideal solution. The sample size (9-12 years old children with ADHD and TD children, 107 in total) is relatively large. The authors are cautious in making a causal conclusion from the patterns of neural response to behavior by examining the relationship between brain response and clinical symptoms. However, the major question about the causal relationship between the brain and the behavior is still unanswered by the association methods adopted here. In addition, is there possible a third latent variable(s) driving both neural and behavioral responses? Overall, this is a solid study with a clear hypothesis stated. However, some methodological questions need to be clarified. See below for major and minor critiques line by line.

2.1. The novel metrics of temporal variability and spatial stability are intriguing, but their validity and underlying mechanisms require further substantiation. Critical questions include: What specific neural activities do these metrics reflect? How do they relate to traditional measures like activity and connectivity? Are there correlations among these metrics? Addressing these points would strengthen the validity and interpretability of these measures as potential ADHD biomarkers. For example, Line 243-245. To ask further, what is the underlying behavior or neural mechanisms of the “unstable recruitment of core cognitive control systems during task performance”? Line 247-249. It is unusual to find no significant group difference in terms of brain activation during in any task conditions.

Response: We thank the reviewer's question. Temporal variability in neural responses has garnered increasing attention in recent decades, particularly in resting-state studies (Hong & Hwang, 2022; Nomi et al., 2018). This metric encompasses measures such as standard deviation and kurtosis, which capture not only the dispersion of neural activity but also the presence of frequent extreme values. These measures differ significantly from traditional mean-based approaches, but have been revealed as an important and useful neural measure in relation to task performance and aging (Dinstein et al., 2015; Grady & Garrett, 2014). Some researchers argue that higher variability reflects neural noise, while others suggest it represents a greater capacity for neural flexibility, enabling more dynamic responses in complex and changing environments (Arazi et al., 2017; Grady & Garrett, 2014; He, 2013; He & Zempel, 2013; Nomi et al., 2017; Wehrheim et al., 2024). Our study extended previous work by investigating temporal variability in task-evoked brain responses, rather than in resting-state data. Spatial stability reflects the consistency of distributed neural activation patterns across trials, as measured using representational similarity analysis (RSA). RAS has been widely used in studying memory, perception, and working memory (Blank & Davis, 2016; Favila et al., 2018; Gao et al., 2023; Gao et al., 2022; Miller et al., 2022; Rissman & Wagner, 2012; Zheng et al., 2018). Spatial stability offers a distinct perspective from mean activation, as it focuses on the trial-by-trial consistency of neural patterns. Previous studies, for instance, have demonstrated that spatial stability in the visual areas and hippocampus can predict memory, even when mean activation is controlled for or when highly selective voxels are excluded (Xue et al., 2010; Zheng et al., 2023; Zheng et al., 2018). Our study is among the first to, and to apply spatial stability measures to cognitive control processes in clinical populations. We have expanded the discussion in the main text to further elucidate what these two metrics reflect in terms of neural processes.

“unstable recruitment of core cognitive control systems” refer to our hypothesis that cognitive control is implemented through distributed neural coding and consistent recruitment of the cognitive control systems from trial to trial, measured by high spatial stability in neural activity patterns, is crucial to main high cognitive control function. Difficulty in recruiting the cognitive control systems in a reliable and consistent manner could lead poor task performance.

Null finding is actually not uncommon in clinical neuroimaging studies (Dennis-Tiway et al., 2019; Heather, 2014; Kennedy et al., 2015; Müller et al., 2017). Despite of

publication bias, non-significant between-group differences could be driven by compensatory mechanisms and heterogeneity of clinical samples. Indeed, recent meta-analytic studies have shown that no significant convergent structural or functional alteration in children with ADHD from previous neuroimaging studies (Cortese et al., 2021; Samea et al., 2019). Our results highlight the limitation of conventional analytic approach with overemphasis of mean activation and raise the possibility of a more powerful and sensitive method for clinical neuroimaging research.

2.2. Lacking of behavioral effects between ADHD and TD groups of the fMRI samples (e.g., in Line 190-205) needs to be carefully discussed. It is hard to argue that the neural results, without the behavioral evidence, are sufficient to provide the support to the hypothesis. Why there was no significant group difference in temporal variability of trial-by-trial brain activation during the CSST (i.e., in Line 213-222)? This is again an issue as the last critique.

Response: Behavioral measures, such as accuracy and RT, are the output from a sequence of cognitive and motor processes. Different neurocognitive mechanisms could lead to similar behavioral performance. Many previous studies in clinical neuroscience have shown significant between-group difference in neural measures while behavioral performance were not significantly different (Bergeron et al., 2020; Iravani et al., 2021; Musella & Weyandt, 2023; Suskauer et al., 2008; Yap et al., 2021). In most clinical behavioral and functional neuroimaging studies, proactive control is quantified using response slowing. However, response slowing could be driven by factors other than effortful proactive control, such as attention or motivation. If effortful proactive control underlies response slowing in children with ADHD and TD children, we will expect to see similar temporal and spatial stability of neural representation and similar inter-trial associations with RT fluctuation in both groups. However, if other internal processes, such as inattention or poor motivation, contributes to response slowing in different scales between children with ADHD and TD children, we will expect to see between-group differences in temporal and spatial stability of neural representation and inter-trial association with RT. Indeed, that is what we found.

We found significant greater temporal variability in children with ADHD compared to TD children.

2.3. The whole-brain analysis reveals inconsistencies in analytical approaches across different metrics, potentially compromising the coherence and interpretability of results. Some analyses directly compare ADHD and TD groups across conditions, while others examine proactive and reactive control contrasts separately within each group, or present results for each condition separately for ADHD and TD groups. This lack of uniformity in analytical logic makes it challenging to draw comprehensive conclusions. Moreover, the analysis of stop trials is inconsistent, sometimes including all stop trials and other times focusing only on successful stops. This inconsistency further complicates interpretation.

Response: We thank the reviewer for the suggestion. Our intent was to extend beyond the traditional focus on mean activation group differences, which are often examined using univariate GLM approaches. These approaches can overlook important aspects of neural activity, such as the temporal and spatial stability of trial-evoked brain responses, which may be critical for understanding inconsistencies in previous findings. To address this gap, we conducted direct between-group comparisons focused on these stability measures.

Regarding proactive and reactive control, we hypothesized that these cognitive processes are implemented through distributed neural coding strategies, which can be detected via contrasts of spatial stability between different trial types. Specifically, we first examined contrasts between SuccStop and Uncertain Go trials, and between Uncertain Go and Certain Go trials within each group, to elucidate the neural mechanisms underlying reactive and proactive control. We then compared the ADHD and TD groups based on these measures to assess group differences.

Following the reviewer and other reviewer's suggestions, we have now included additional text in the introduction part explaining the logic of our analyses. Please see below:

"Our study adopts a comprehensive approach to investigate neural instability in ADHD. We begin with a whole-brain analysis of temporal variability and spatial stability of trial-evoked responses during proactive and reactive control. This allowed us to identify regions across the entire brain that show atypical neural dynamics in ADHD. Following this, we narrow our focus to specifically examine neural instability within key regions of the cognitive control system. This targeted analysis enables us to investigate how neural instability in these critical areas relates to behavioral instability and core ADHD symptoms."

We have also reported the results using combined 'Stop' trial throughout our revised manuscript to improve the consistency. Please see below for Figure 2, Figure 3a, and Figure 5a:

a. Temporal Variability of β time series (std)

b. Temporal Variability of β time series (kurtosis)

a. Weaker spatial stability of trial-evoked brain response patterns in children with ADHD than TD children

b. Weak spatial stability of trial-evoked brain response patterns during proactive and reactive control in children with ADHD

a. Within-group inter-subject pattern similarity differs between groups

b. Within-group inter-subject pattern similarity underlies proactive and reactive control

c. Group-specific inter-subject pattern similarity underlies proactive and reactive control

Abstract:

2.4. The clarity of abstract needs to be improved. For instance, the key concepts of proactive and reactive cognitive control need to be explained. The statements that not reflect the results accurately need to be justified, i.e., “Typically developing (TD) children exhibited stable neural response patterns for efficient proactive and reactive dual control mechanisms.”

Response: We thank the reviewer for the suggestions. We have revised the abstract to include an explanation of proactive and reactive cognitive control. Additionally, the statement regarding TD children’s neural response patterns has been corrected for accuracy. Please see the updated abstract below.

“This study investigates the neural underpinnings of cognitive control deficits in ADHD, focusing on trial-level variability of neural coding. We employed a novel computational approach to single-trial neural decoding on a cued stop-signal task which allowed us to probe both proactive and reactive cognitive control under the dual cognitive control model. The model posits two distinct modes of inhibitory control: reactive control, which involves suppressing or overriding an automatic response when interference is detected, and proactive control, which involves implementing preparatory strategies to manage potential interference based on prior information. In contrast to typically developing children, children with ADHD showed disrupted neural coding during both proactive and reactive control, characterized by increased temporal variability and diminished spatial stability in neural responses in salience and frontal-parietal network regions. This variability correlated with fluctuating task performance and ADHD symptoms. Additionally, children with ADHD exhibited more heterogeneous neural response patterns across individuals compared to TD children. Our findings underscore the significance of modeling single-trial variability and representational similarity in understanding distinct components of cognitive control deficits in ADHD, offering new perspectives on neurocognitive dysfunction in psychiatric disorders.”

Introduction:

2. 5. Line 83-85 and Line 115-117. Reference(s) should be provided for this statement, i.e., for the “few studies”.

Response: Thanks for raising this point out. Though there are few studies examining temporal variability during resting state in children with ADHD, no previous studies have examined temporal and spatial variability in neural coding during inhibitory control. Therefore, we have revised the text as below:

‘However, no previous studies have examined the temporal variability in neural coding and the spatial stability of neural representations in children with ADHD, which limits our understanding of how ADHD impacts the brain’s ability to consistently engage in reactive control.’

2.6. Line 116. The functional role of the frontoparietal network (FPN) and the salience network (SN) in cognitive control has been examined by Wu et al., (2020) by conducting an activation-likelihood-estimation-based large-scale meta-analysis, with the fMRI studies using stop-signal task included.

Wu et al, (2020). The functional anatomy of cognitive control: A domain-general brain network for uncertainty processing. DOI: 10.1002/cne.24804

Response: Thank you for highlighting this relevant study. We have now included the reference.

Methods:

2. 7. The participant sample reporting raises several concerns. Of 107 recruited children, only 93 (53 ADHD + 40 TD) are accounted for initially. The final sample of 26 ADHD and 53 TD indicates substantial exclusions, particularly in the ADHD group. A clear, detailed report of exclusions is crucial, specifying the number excluded for each reason in each group. The high exclusion rate in the ADHD group raises concerns about potential bias. Does excluding behavioral outliers inadvertently remove participants with more severe ADHD symptoms? This could significantly impact the study's representativeness and generalizability. Additionally, the rationale for excluding participants with stop accuracy higher than 75% is unclear and needs justification. This criterion might potentially remove high-performing ADHD participants, further skewing the sample.

Response: As we replied to the other reviewer, participants were excluded from our formal analysis if they did not meet specific behavioral criteria: Go accuracy below 75%, or Stop accuracy below 25% or above 75%. These exclusions are grounded in the Race model, which posits independent "Go" and "Stop" processes competing against each other. The task employed an adaptive procedure, meaning that if participants successfully (or fail to) withhold their response in one Stop trial, the next Stop trial becomes more challenging (or easier), ideally leading participants to achieve Stop accuracy around 50%. Participants with Stop accuracy above 75% are not high-performing ones but did not follow instructions and used a waiting strategy. These criteria are widely used in stop signal paradigms and are standard practice in the field (Congdon et al., 2012; Verbruggen et al., 2019; White et al., 2014).

2.8. The sample size of 26 ADHD and 35 TD children raises concerns about statistical power. While acceptable for fMRI studies, it's relatively small and could limit the robustness and generalizability of findings. It's unclear whether a priori power analysis was conducted to determine the ideal sample size for detecting the expected effects.

Response: We appreciate the reviewer's concern regarding the sample size. Since no previous studies have specifically examined the temporal and spatial variability of neural activity patterns during inhibitory control in children with and without ADHD, we conducted power analysis based on data from our prior study (Cai et al., 2021, Molecular Psychiatry). The power analysis indicated that a total of 60 participants (29 ADHD, 31 TD) is sufficient for 80% power at an alpha of 0.05. Details of power analysis

is included in the Supplementary Material results. The sample size also aligns with fMRI research standards and reflects the practical limitations of recruiting from this specific population.

2.9. Clear details about the CSST paradigm, particularly the trial timeline, need to be reported. This is crucial for single trial extraction accuracy and potential signal contamination between trials. Providing estimated hemodynamic response curves for each condition could help address this concern (see “Wu, T., Wang, X., Wu, Q., Spagna, A., Yang, J., Yuan, C., Wu, Y., Gao, Z., Hof, P. R., & Fan, J. (2019). Anterior insular cortex is a bottleneck of cognitive control. *NeuroImage*, 195, 490–504” as an example). Additionally, reporting the number of trials included in the final analysis for each condition is necessary.

Responses: We apologize for the lack of clarity regarding the CSST paradigm in our initial submission. We have revised the experiment figure and added more details to the Methods section. Additionally, we have included Table S9, which reports the average number of trials/pairs per condition and group included in the final fMRI analyses.

*We understand the reviewer’s concern about potential signal contamination between trials. However, single-trial model has been widely used for over a decade for studying perception, memory, working memory, and decision-making (Abdulrahman & Henson, 2016; Bein et al., 2020; Feng et al., 2021; Mumford et al., 2014; Zeithamova et al., 2017; Zheng et al., 2021; Zheng et al., 2018; Zhou et al., 2022). Among many single-trial estimation approaches, the Least Squares - Single (LSS) method, which we used in this study, has become a standard approach for addressing trial-to-trial contamination caused by temporal autocorrelation in fast-event design (Mumford et al., 2014; Mumford et al., 2012; Prince et al., 2022). Since its introduction, the LSS method has been refined. One key improvement is modeling the targeted single trial separately from all other trials, which are categorized into separate EVs (Zeithamova et al., 2017). Experts in the field suggest that an average trial duration of approximately 3 seconds (stimulus duration + inter-trial interval) can yield reasonable results (see GLM Single Documentation: <https://glmsingle.readthedocs.io/en/latest/wiki.html>). Our study not only adopted this updated LSS approach and an optimized stop signal experiment with an average trial duration of 3.96 seconds but also benefited from the use of an ultra-fast fMRI sequence with a TR of 0.49 seconds, making our approach even more robust compared to previous studies. We have provided the number of trials included in the analysis in each condition in the updated Supplementary Material **Table S9**.*

Table S9. Averaged number of trials/pairs included in fMRI analysis by condition and group

	TD				ADHD			
	Uncertain Go	Certain Go	SuccStop	Stop	Uncertain Go	Certain Go	SuccStop	Stop
Trial Number	59.714	58.2	16.486	31.514	56.423	56.192	16.385	29.846
Pair Number	893.143	849.4	68.571	248.257	800.346	793.692	67.692	223.385

SuccStop: Successful stop trials.

We hope this addresses the reviewer's concerns, and we remain open to conducting additional analyses if deemed necessary.

2.10. Details of GLM onset extraction needs to be clarified. With 2-3 stimuli events per trial (cue, green arrow, red arrow), it's unclear which event the trial onset is locked to. Is modeling these multi-event trials with a single stick function appropriate? Additionally, the treatment of go trials without responses in GLM construction needs explanation. In addition, Line 831-834. For the (3) Uncertain Go trials, (4) Certain Go trials, and (5) US trials, they all involve a motor response in common. Can you justify that there is no need to model (extract out) the motor response related activation?

Response: Trial onset was locked to the green arrow, which represents the Go stimulus. Go trials without responses were included as nuisance regressors in the GLM. It is important to clarify that our experiment design, the CSST should not be treated like a typical multi-component paradigm, such as visual short-term memory tasks that have distinct encoding, maintenance, probe, and response phases. In the CSST, events (cue, Go stimulus, Stop signal) are temporally adjacent and brief in duration, with the cue phase having a fixed duration of 300 ms.

Given the temporal proximity between events, there is a significant degree of collinearity. Attempting to model the cue and red arrow alongside the Go event would introduce collinearity issues, complicating the model. Therefore, we chose to model the Go stimulus onset with a single stick function, which is a standard approach used in stop-signal task (Fauth-Bühler et al., 2012; Padmala & Pessoa, 2010; Sebastian et al., 2013; Sebastian et al., 2017; Smittenaar et al., 2013).

For reactive control, a key contrast involves comparing SuccStop and Uncertain Go trials. Modeling motor responses separately would complicate this comparison because no motor response occurs in SuccStop trials, and the differences in stimuli (i.e., the "Stop" and "Go" arrows) could introduce confounding factors. While both Uncertain Go and Certain Go trials involve motor responses, proactive control mechanisms are engaged before motor execution. Thus, separating motor response-related activation would not affect our ability to identify the neural mechanisms of proactive control.

In conclusion, we believe that the modeling approach we adopted not only aligns with standard practices in the field for identifying neural substrates of proactive and reactive

control, but also improves single-trial estimation by addressing potential collinearity issues.

2.11. The study includes numerous univariate ROI analyses, raising concerns about multiple comparisons. It's crucial to report whether any correction for multiple comparisons was applied to these analyses.

Response: We apology for not making this clear. FDR correction was applied for every ROI-based analysis, and we have claimed this in our revised manuscript.

2.12. It's unclear whether the group-level whole-brain analyses controlled for important covariates such as gender and age. This information is crucial for understanding the robustness of the findings. Additionally, the study doesn't specify whether ADHD comorbidities were considered or if analyses were conducted for different ADHD subtypes. Given the heterogeneity of ADHD, these factors could significantly impact the results and their interpretation.

Response: We appreciate the reviewer's constructive suggestion. To enhance the robustness of our findings, we conducted additional control analyses to account for the potential influences of age, gender, and head motion. Specifically, we performed a series of linear regression analyses where the neural measures served as dependent variables, the group was treated as the independent variable, and age, gender, and head motion were included as covariates of no interest. Additionally, we utilized Cook's distance to assess and mitigate the influence of potential outliers. Importantly, these analyses yielded results that replicated our original findings. Please see the results in Supplementary Materials results and **Table S5-S8**. Unfortunately, our sample size prevent us from investigation of comorbidity and subtypes. Future studies are needed to address those important questions.

"Robustness with respect to age, gender, and head motion"

To assess the robustness of our findings against potential confounding factors, including age, gender, and head motion, we conducted additional ROI-based analyses for between-group comparisons while controlling for these variables. We selected ROIs implicated in cognitive control from salience (SN), encompassing the right anterior insula (rAI) and presupplementary motor cortex (preSMA), and frontal-parietal regions (FPN), encompassing right dorsal lateral prefrontal cortex (rdLPFC), and right posterior parietal cortex (rPPC) (see Methods and Figure 1h).

Increased temporal variability of trial-evoked brain responses in children with ADHD

In comparison to TD children, children with ADHD exhibited significantly higher standard deviation in the preSMA, rDLPFC, and rPPC during Uncertain and Certain Go trials, in all four ROIs during Stop trials after accounting for age, gender, and head motion (all $p < 0.002$, FDR corrected). Children only showed lower standard deviation in the rAI during Uncertain and Certain Go trials ($p < 0.001$, FDR corrected), see **Table S5**. Moreover, children with ADHD had significantly higher kurtosis than TD children in the

rAI during Uncertain Go trials, and the rPPC during Certain Go trials (all $p < 0.006$, FDR corrected) after controlling for age, gender, and head motion. FDR correction was applied to four ROIs, see **Table S6**.

Weaker spatial stability of trial-evoked brain responses in children with ADHD

In comparison to TD children, children with ADHD exhibited significant lower spatial stability in the preSMA and rPPC across all three trial types ($p < 0.05$, FDR corrected) after controlling the effect of age, gender, and head motion, see **Table S7**.

Weakened association between proactive control system and behavioral regulation in ADHD

In comparison to TD children, children with ADHD exhibited a significantly weaker association between the inhibitory control pattern (ICP) and reaction time in the preSMA, rAI, and rdLPFC during Uncertain Go trials after accounting for age, gender, and head motion ($p \leq 0.015$, FDR corrected). This weaker association was also observed in the preSMA when both Uncertain and Certain Go trials were combined ($p = 0.004$, FDR corrected), after controlling for age, gender, and head motion, see **Table S8**.

Table S5. Increased temporal variability (standard deviation) of trial-evoked brain responses in children with ADHD (FDR corrected).

Brain measures	Conditions	Slope	p-value
Uncertain Go	preSMA	0.248	$p < 0.001$
	rAI	-0.390	$p < 0.001$
	rdLPFC	0.601	$p = 0.001$
	rPPC	0.373	$p < 0.001$
Certain Go	preSMA	0.214	$p < 0.001$
	rAI	-0.382	$p < 0.001$
	rdLPFC	0.579	$p < 0.001$
	rPPC	0.434	$p < 0.001$
Stop	preSMA	0.104	$p < 0.001$
	rAI	0.121	$p < 0.001$
	rdLPFC	0.127	$p < 0.001$
	rPPC	0.086	$p < 0.002$

Table S6. Increased temporal variability (kurtosis) of trial-evoked brain responses in children with ADHD (FDR corrected).

Brain measures	Conditions	Slope	p-value
	preSMA	0.210	0.445

Uncertain Go	rAI	0.408	0.026
	rdIPFC	0.104	0.445
	rPPC	0.188	0.445
Certain Go	preSMA	0.077	0.402
	rAI	0.180	0.402
	rdIPFC	0.077	0.402
	rPPC	0.318	0.015
Stop	preSMA	0.070	0.708
	rAI	0.134	0.708
	rdIPFC	0.243	0.558
	rPPC	0.141	0.374

Table S7. Weakened spatial stability of trial-evoked brain responses in children with ADHD (FDR corrected).

Brain measures	Conditions	Slope	p-value
Uncertain Go	preSMA	-0.034	0.004
	rAI	-0.025	0.119
	rdIPFC	-0.032	0.119
	rPPC	-0.028	0.011
Certain Go	preSMA	-0.011	0.039
	rAI	0.008	0.695
	rdIPFC	-0.032	0.108
	rPPC	-0.038	0.010
Stop	preSMA	-0.024	0.023
	rAI	-0.005	0.061
	rdIPFC	-0.045	0.339
	rPPC	-0.061	0.023

Table S8. Weakened association between proactive control system and behavioral regulation in ADHD (FDR corrected).

Brain measures	Conditions	Slope	p-value
Uncertain Go	preSMA	-0.069	0.009
	rAI	-0.042	0.015
	rdIPFC	0.027	0.004
	rPPC	-0.053	0.134
Certain Go	preSMA	-0.014	0.121
	rAI	-0.066	0.121
	rdIPFC	-0.040	0.121

	rPPC	-0.023	0.818
	preSMA	-0.050	0.004
Uncertain + Certain Go	rAI	-0.048	0.054
	rdIPFC	0.008	0.054
	rPPC	-0.029	0.184

Results:

2.13. The presentation of results could be optimized for clarity and comprehension. For Figures 2a&b, the group difference in STD (ADHD > TD) is very different from the hypothesized regions/networks. It's challenging to discern whether activations fall within ROIs. Overlaying ROI contours on whole-brain activation results would be beneficial. The inconsistency in brain surface inflation between Figure 1h (ROI presentation) and activation results hinders comparison. In addition, lists of activation foci/clusters are missing, which would aid in precise interpretation. In addition, can you also make the plot of TD > ADHD to see the regions with more temporal variability in the TD group?

Response: We have generated a new ROI figure. Additionally, we have included tables of activation foci and clusters in the Supplementary Materials **Table S12-S25** to facilitate precise interpretation.

Regarding the comparison between the TD and ADHD groups, our analyses did not reveal any clusters showing greater temporal variability in the TD children compared to the ADHD children.

2.14. Several figures require improvements: Figure 4a lacks a color bar; Figure 4c wasn't mentioned in the main text; Figure 1 panels were referenced inconsistently, with some mentioned even after Figure 3; Figure 1g is confusing, as both TD and ADHD example subjects are labeled "s1". Line 346. Should it be Figure 1f? Line 431. Figure 1g instead of 1f? Line 455-457. The scatter plot (Figure 6) does not indicate a correlation of .436. It is misleading since you have a regression model with other variables included as covariates. Omit this figure?

Response: Thank you for your feedback. We have modified the figures based on the suggestions.

In Figure 6, the correlation coefficient was based on a partial correlation between the ADHD-like neural pattern and clinical symptoms, with confounding factors accounted for. The residuals represented the output of neural and clinical measures after regressing out these confounding factors. We have now updated the figure caption for clarity. Please see the updated figure and caption below.:

"Last, we conducted ROI-based analysis to investigate whether similarity to the averaged spatial patterns of the ADHD group during proactive and reactive control is associated with individual differences in core symptoms of ADHD. We found a significant positive correlation between the pattern similarity to ADHD group averaged

*spatial pattern during proactive control in rAI and inattention score ($r = 0.354$, $p = 0.037$, FDR corrected, **Figure 6**). And this effect remained significant after accounting for age, gender, IQ, and head motion ($r_{\text{partial}}=0.355$, $p=0.009$). Our results indicated that a more ADHD-like neural activity pattern during proactive control was associated with more severe clinical symptoms.”*

Please see below for updated figures:

a. Experiment Design

Cued Stop Signal Task (CSST)

b. Dual control model

c. Hypotheses

d. Gaussian modeling of β time series per voxel (temporal stability)

e. Representational similarity analysis (spatial stability)

f. Inhibitory control pattern (ICP) modulates RT

g. Inter-subject RSA: Group-specificity

h. Core ROIs in the salience and frontal-parietal network

Figure 6. Neural variability of the rAI predicts ADHD symptom severity. The residualized ADHD-like activation pattern in the right anterior insula (rAI) exhibited a significant correlation with the residualized inattention scores after accounting for age, gender, IQ, and head motion ($r=0.355$, $p=0.009$).

2.15. Line 303-304. This sentence is not clear: i.e., the “ADHD recruit highly variable distributed brain regions.” Do you imply the regions shown in Figure 3’s top panel? For Figure 3, three words of “weaker”, “weak”, and “weakened” are used. Do they express different meanings?

Response: We have updated the wording accordingly. Please see below:

a. Weaker spatial stability of trial-evoked brain response patterns in children with ADHD than TD children

b. Weaker spatial stability of trial-evoked brain response patterns during proactive and reactive control in children with ADHD

Figure 3. Weaker spatial stability of trial-evoked neural responses in children with ADHD. *a.* Typically developing (TD) children showed greater spatial stability than children with ADHD in salience and frontoparietal networks during Uncertain Go, Certain Go and SuccStop trials. *b.* Highly stable recruitment of salience and frontoparietal networks during proactive and reactive control was found in TD children ($p < 0.05$, corrected) but not in children with ADHD.

2.16. Line 312-314. The operationally defined contrasts for proactive control (Uncertain vs. Certain Go trials) and reactive control (Successful Stop vs. Uncertain Go trials) need

to be justified/explained here. The two conditions for the contrast of reactive control are different in terms of motor response. Also, they involve a common condition, the uncertain go trials. Therefore, the two contrasts/measures are not totally independent.

Response: We acknowledge that the conditions for the contrasts used to define reactive control are different in terms of motor response, which is indeed necessary for studying reactive control. Specifically, successful inhibition requires no motor response, which differentiates it from the Uncertain Go trials. This approach aligns with standard practices in the field of inhibitory control research. Previous studies have used similar contrasts to examine proactive and reactive control processes (Hu et al., 2016; Messel et al., 2019; van Belle et al., 2014; van Rooij et al., 2014)

2.17. Line 318-320. This sentence may not be accurate: “whereas children with ADHD showed significant spatial stability in less distributed brain regions, including visual areas and right PPC ...”. In addition, the results/inferences of Figure 3b for the ADHD and TD groups are not supported by the testing of group difference. Therefore, it should be cautious in make such inferences/conclusions solely based on visual inspection.

Response: This particular sentence refers to Figure 3b. Following the reviewer’s suggestion, we have revised the statement as the following: “Further ROI analyses revealed that TD children showed great spatial stability of trial-evoked neural responses in the rAI, preSMA and rdIPFC during proactive control and in all four ROIs during reactive control ($p < 0.05$, FDR corrected, Figure 3b). Children with ADHD also exhibited great spatial stability in the rAI, preSMA, and rPPC during reactive control but only in the rPPC during proactive control ($p < 0.05$, FDR corrected, Figure 3b). Moreover, TD children demonstrated significant greater spatial stability of trial-evoked neural responses during proactive control in the preSMA, rAI, and rDLPFC compared to children with ADHD ($p < 0.05$, FDR corrected, Figure 3b), while no significant effect was observed during reactive control. Together, our findings suggest that children with ADHD have weakened spatial stability from trial to trial in the distributed brain systems and in the frontoparietal network during proactive control in comparison to TD children.”

a. Weaker spatial stability of trial-evoked brain response patterns in children with ADHD than TD children

b. Weaker spatial stability of trial-evoked brain response patterns during proactive and reactive control in children with ADHD

2.18. Line 328-332. If you treat this study as hypothesis driven, ROI can be considered to be reported first with the whole brain results as data-driven further examination or for the visualization of the brain responses.

Response: We thank the reviewer's suggestion. While it is a hypothesis driven study, we prefer to first provide readers a whole brain view of neural encoding pattern and followed by hypothesis-defined ROIs. Nevertheless, we are open to change the order of presentation if necessary.

Discussion:

2.19. The subtitles of the Discussion are very informative. However, this section has some redundancy. Some reviews and materials that have been covered in the Introduction and Results are repeated extensively here, e.g., lines 494-510, 433-551, 632-640. In addition, the discussion section would benefit from clearer distinctions between experimental results, interpretations, and speculations. More explicit language should be used to differentiate these elements. The potential clinical applications of the findings for ADHD diagnosis and treatment could be discussed in more detail, while clearly stating the need for further validation. The discussion should more explicitly articulate how this study advances neuroscientific understanding of ADHD and how it challenges or complements existing ADHD models.

Response: We thank the reviewer's suggestion and have revised the discussion based on these comments.

2.20. The discussion lacks a comprehensive examination of the activity mechanisms and functional modes of brain networks and key regions. There's insufficient unified discussion on brain mechanisms related to temporal and spatial variability, leading to a lack of a coherent framework for understanding the roles of CON and FPN in ADHD. In addition, the analysis revealed activations in areas beyond the predefined ROIs, such as the visual cortex and thalamus. The potential roles of these regions in ADHD should be addressed.

Response: We thank the reviewer's suggestion and have added extra discussion points in the revised manuscript.

2.21. The discussion mentions three potential sources of inter-subject similarity: variability in symptom profiles and cognitive deficits, diversity of cognitive strategies during task performance, and inherent instability of neural dynamics. Interestingly, these hypotheses seem testable using the existing dataset. These additional analyses could provide valuable insights, strengthening the study's conclusions and offering a more comprehensive understanding of inter-subject variability in ADHD. Consider incorporating these analyses or discussing why they weren't pursued.

Response: The reviewer has made an excellent point. However, a large-scale dataset is required to probe heterogeneity in the clinical profiles and cognitive strategies. We are exploring these opportunities using existing large-scale datasets, such as the ABCD study, in separate studies, although the task fMRI data in the ABCD study does not allow us to dissociate proactive versus reactive control.

2.22. The study's focus on cognitive control deficits in ADHD raises important questions about the specificity and generalizability of the observed changes in temporal and spatial variability. Cognitive control deficits are common across various psychiatric disorders, not just ADHD. We recommend discussing: Whether the observed patterns of temporal and spatial variability are specific to ADHD or might be seen in other disorders with cognitive control deficits? The task-specificity of these new measures. How might they generalize across different cognitive control paradigms? The potential for these

measures to differentiate ADHD from other disorders with similar cognitive deficits, and the broader applicability of this analytical approach to studying neural dynamics in other psychiatric conditions.

Response: We thank the reviewer for raising this great point. Indeed, our recent study has demonstrated a generalized neural dynamic mechanism underlying cognitive control across different cognitive tasks and populations (Cai et al., 2024, Nature Communications). We have now discussed these points in the revised manuscript.

Reviewer #3 (Remarks to the Author):

The present manuscript provides novel insights into trial-based variability in ADHD during inhibitory control. In a comprehensive set of analyses, the authors show for example:

- Greater temporal variability of trial-based brain activation in ADHD
- Greater spatial variability in ADHD in various inhibition-related brain regions
- Weaker association of higher RT with increased “similarity” between go trials and successful stop trials in ADHD in inhibition and motor-response related brain regions
- Greater inter-subject spatial variability of brain activation within the ADHD group

The manuscript is generally well written, the study is well executed and the statistics appear to be sound. While generally excited for the analysis approach the authors present, my enthusiasm is somewhat reduced by two aspects: 1) reporting marginally significant findings and reporting uncorrected fMRI results in a study of this quality and rigor seems ill placed 2) a more nuanced discussion of the findings seems warranted, there were instances of too bold or broad statements that do not entirely reflect the reality of the presented data.

My comments are detailed below.

Overall:

3.1- The manuscript would benefit from thorough proof reading, there were a number of typos

Response: We apologize for the typos in the initial submission. We have carefully proofread the revised manuscript and corrected typos.

Results:

3.2- Line 207 ff: stating that children with ADHD have poor reactive control and less stable performance but preserved proactive control seems a bit bold given that the only significant findings were observed in the full sample and were not replicated in the fMRI subsample.

Response: Thank you for this comment. We acknowledge that our statement may have been too strong. We have revised the text to more accurately reflect the findings, noting that the significant results were observed in the full sample and not replicated in the fMRI subsample.

3.3- Line 270 ff: Please refrain from reporting “trend level” statistics. While I understand this is tempting, a non-significant result at a given threshold remains non-significant. Further, given the vast number of tests run on this dataset, setting critical alpha to .05 is already very liberal (also see my question regarding correction for multiple testing).

Response: Thank you for your suggestions. Following the reviewer’s suggestion, we have removed all trend-level uncorrected results in the main text.

3.4- Line 320 ff: Similarly, please do not interpret a “weakened spatial stability in the distributed brain systems during proactive and reactive control in comparison to TD children” based on $p < .01$ uncorrected results. I suggest to omit this completely. At the very least, specify this as an explorative analysis and discuss the findings accordingly and carefully.

Response: Thank you for your suggestions. Following the reviewer’s suggestion, we have removed all trend-level uncorrected results in the main text. While our ROI-based analysis did show significant greater spatial stability in neural coding during proactive control in TD children compared to children with ADHD. We have now updated the figure to represent this result.

a. Weaker spatial stability of trial-evoked brain response patterns in children with ADHD than TD children

b. Weaker spatial stability of trial-evoked brain response patterns during proactive and reactive control in children with ADHD

Figure 3. Weaker spatial stability of trial-evoked neural responses in children with ADHD.

a. Typically developing (TD) children showed greater spatial stability than children with ADHD in salience and frontoparietal networks during Uncertain Go, Certain Go and SuccStop trials. **b.** Highly stable recruitment of salience and frontoparietal networks during proactive and reactive control was found in TD children ($p < 0.05$, corrected) but not in children with ADHD.

Discussion:

3.5- Line 610 ff: “Notably, we found that the ICP index in key regions of the salience and frontoparietal network nodes tracked trial-wise RT fluctuations in TD children.” This

could be misunderstood. The authors did not correlate with a measure of RT fluctuation, but with RT in general. This is used a few times across the manuscript. I do understand what the authors are trying to say, but think this could be misinterpreted.

Response: Thank you for your valuable feedback. We have rephrased the statement to clarify that the ICP index tracked general RT rather than fluctuations in RT. This revision has been made throughout the manuscript to prevent any potential misinterpretation. Please see below for relevant modifications:

“Weak association between trial-evoked inhibition-alike brain response and RT in children with ADHD

*Increased IIRV is a robust behavioral phenotype associated with ADHD, which is commonly regarded as an index of attention lapse or inattention (Cai et al., 2021; Cai et al., 2023; Mizuno et al., 2023). However, behavioral fluctuation could also be driven by trial-by-trial response strategy adjustment, a dynamic proactive control process modulated by participants’ time-varying anticipation (Cai et al., 2017; Cai et al., 2023; Ide et al., 2013; Shenoy & Yu, 2011). Here we examined whether the degree of proactive control in trial-evoked brain response patterns can track RT in Certain and Uncertain Go trials. As proactive and reactive control may share similar neural underpinnings (Swann et al., 2012), we developed an “inhibitory control pattern” (ICP) index in each Certain and Uncertain Go trial by quantifying the similarity between trial-evoked brain response pattern on Go trial and a subject-specific “template” brain response pattern of inhibitory control (**Figure 1f**). The subject-specific “template” brain response pattern of inhibitory control was obtained by averaging trial-evoked brain response pattern from all the Successful Stop trials (β -maps from the single-trial model). Then a searchlight approach was implemented to produce a whole brain map of ICP index in Certain and Uncertain Go trials separately per subject (see Methods for more details). The larger ICP value, the greater similarity between trial-evoked brain response pattern on Certain and Uncertain Go trials and the averaged brain response pattern on Successful Stop trials, indicating greater engagement of inhibitory control systems during Go trials. Last we examined correlation between voxel-wise ICP and RT across trials.*

*We found that trial-wise RT is significantly correlated with the ICP in the bilateral AI, right inferior frontal gyrus, superior frontal gyrus, preSMA in TD children in Certain and Uncertain Go trials ($p < 0.05$, corrected, **Figure 4a and Supplementary Table S17**). Children with ADHD showed less distributed effect in the supplementary motor area and anterior cingulate cortex in Certain and Uncertain Go trial ($p < 0.05$, corrected, **Figure 4a and Supplementary Table S18**). Between group comparison further revealed that children with ADHD have significantly weaker relationships between trial-wise RT and ICP in anterior cingulate cortex/supplementary motor cortex, right AI, right posterior middle temporal gyrus, and right supramarginal gyrus extending to the postcentral gyrus in Certain and Uncertain Go trials ($p < 0.05$, corrected, **Figure 4b and Supplementary Table S19**). These findings suggest that TD children can flexibly recruit inhibitory control*

systems to implement proactive control and thereby modulate trial-wise response time, but such ability is much weakened in children with ADHD.

Weak association between trial-evoked inhibition-alike brain response in the cognitive control system and RT in children with ADHD

Next, we examined whether children with ADHD exhibited weak association between ICP in the cognitive control system and RT. TD children demonstrated a significant relation between trial-wise RT fluctuations across trial types and ICP of the preSMA, rAI, and rPPC during Uncertain Go, all four ROIs during Certain Go and when both Certain and Uncertain Go trials were considered (all $p_s < 0.05$, FDR corrected, **Figure 4a**). Children with ADHD also demonstrated significant relationships between trial-wise RT and ICP in the preSMA during Certain ($p < 0.05$, FDR corrected, **Figure 4a**), but not in the other ROIs (all $p_s > 0.05$). Moreover, in comparison to children with ADHD, TD children exhibited significant stronger association between trial-wise RT and the ICP of all four ROIs during both Uncertain and Certain Go trials, in the rAI and preSMA during Uncertain trials, and in the rDLPFC in the Certain Go trials (all $p_s < 0.05$, FDR corrected).

We also examined whether the association strength between ICP in the cognitive control system and RT is associated with severity of clinical systems. After accounting for the age, gender, IQ, and head motion, the association strength in the rDLPFC during Certain Go trial was significantly correlated with inattention score ($r_{\text{partial}} = -0.381$, $p = 0.028$, FDR corrected); and the association strength in the rPPC was also significantly correlated with inattention score when both Uncertain and Certain Go trials were considered ($r_{\text{partial}} = -0.378$, $p = 0.03$, FDR corrected). Our findings suggest that the children who exhibit severer clinical symptoms are less likely to effectively modulate cognitive control system in order to adapt their response strategies.”

Also in the discussion: “Notably, we found that the ICP index in key regions of the salience and frontoparietal network nodes tracked trial-wise RT in TD children. However, this brain-behavior association was much weaker in children with ADHD compared to TD children (**Figure 4**).”

3.6- Line 615 ff: The interpretation of the association between ICP index and RT is cut short to an extent. While the authors state carefully that they measure proactive control for instance by RT slowing in uncertain go vs. certain go trials, they argue in the discussion that based on the found association of higher RT with increased “similarity” between go trials and successful stop trials, this higher similarity then encodes the amount of proactive control implemented. However, the stated correlation was found for certain and uncertain go trials alike. Thus, generally suggesting that higher RT reflect overall increased proactive control. This is not warranted, as other processes and factors may contribute to RT slowing.

Response: We appreciate the reviewer’s insightful comment. We acknowledge that our interpretation of the association between the ICP index and reaction time (RT) could

have been more clearly articulated. We agree with the reviewer that other factors such as inattention, fatigue, and reduced motivation are also critical in influencing proactive control. However, we doubt that those factors are the major contributor for the association between ICP and RT. ICP is measured by the similarity between Go and Stop trials. If response slowing is driven by low attention/cognitive control state, one would expect negative correlation between ICP and RT. Furthermore, in our previous study (Cai et al., 2023, Translational Psychiatry), we found negative correlation between response slowing and SSRT, suggesting that increased RT benefit inhibitory control. If response slowing is driven by low attention and motivation state, it will hinder rather than benefit inhibitory control processes. We have added these discussion points in the revised manuscript.

3.7- Line 696 ff: “Furthermore, we found that children who exhibited more ADHD-like spatial activation patterns during proactive control had more severe inattention symptoms” this statement seems a lot too broad given that this finding was based on an ROI analysis and significant for the rAI only. Please also adjust the following discussion accordingly.

Response: We acknowledge that our statement regarding the relationship between ADHD-like spatial activation patterns and inattention symptoms was too broad. We have revised the manuscript to reflect this and adjust the discussion accordingly.

3.8- A brief discussion on the fact that the authors did not find group differences in the mean activation in standard GLM analysis of this task, along with implications of this, would be appreciated. This would be an important point for discussion given the partly heterogenous evidence of standard fMRI studies in ADHD and further highlight the relevance of this new approach.

Response: We appreciate the reviewer’s insightful comment. As mentioned in response to another reviewer, it is not uncommon to observe null results in neuroimaging studies involving children with ADHD. A growing body of literature, including recent meta-analyses, has highlighted the inconsistency of findings in traditional fMRI studies of ADHD, especially in task-based GLM analyses. Our results align with these broader findings, as the heterogeneity inherent in ADHD often makes it difficult to identify consistent group differences using standard univariate methods. The absence of group differences in mean activation in our study underscores the limitations of traditional approaches in capturing the full complexity of neural dynamics in ADHD. In contrast, our use of novel measures, such as temporal variability, spatial stability, and ICP offers a more sensitive approach, capable of detecting subtle neural differences that conventional GLM analyses might overlook. This highlights the value of advanced neuroimaging techniques in better understanding ADHD-related neural alterations. We have expanded the discussion in the revised manuscript to further elaborate on the implications of these findings and to emphasize the relevance and potential of the new approach we adopted.

Methods:

3.9- Could the authors comment on their decision to exclude participants with “outlier” behavioral data and how the cutoffs for that were selected?

Response: Participants were excluded from our analysis if they did not meet specific behavioral criteria: Go accuracy below 75%, or Stop accuracy below 25% or above 75%. These exclusions were based on the premise of the Race model, which posits that independent “Go” and “Stop” processes compete against each other. Participants who employed a waiting strategy or indiscriminately pressed buttons, regardless of the Go or Stop signals, could achieve Stop accuracy greater than 75% or less than 25%, indicating a lack of inhibitory control processes. These exclusion criteria are commonly used in stop signal paradigms and represent a standard approach in the field (Congdon et al., 2012; Verbruggen et al., 2019; White et al., 2014) We have revised the manuscript to clarify this rationale.

3.10- How long was the task in total for each run?

Response: Each run of the task lasted 5 minutes and 15 seconds. This included 32 Uncertain Go trials, 32 Certain Go trials, and 16 Stop trials, with a jittered inter-trial interval ranging from 1 to 4 seconds.

3.11- The authors excluded any pairs presented in the same run from the calculation of pattern similarity to avoid any autocorrelation issues. How could this have affected results differentially for TD and ADHD groups given that ADHD participants second run may have been impacted by impairments in sustaining attention?

Response: Thanks for raising this valuable insight. In our study, we analyzed spatial stability of neural activity patterns across runs to avoid autocorrelation issues because this is a recommended procedure for studies employing representational similarity analysis (Dimsdale-Zucker & Ranganath, 2018; Mumford et al., 2014). But we agree with the reviewer that sustained attention, fatigue, and arousal are important components factors affecting task performance especially in children with ADHD. Considering the multiple sessions nature of our experiment design, children with ADHD might demonstrate consistent sustained attention deficits across sessions or show elevated deficits in the second session due to fatigue or reduced sustained attention. Therefore, we compared behavioral performance for various indices across sessions in children with and without ADHD, our results revealed that both groups did not show significant differences in behavioral performance between sessions, please see below for **Table S10 ~ S11**.

Table S10. Performance difference across sessions in TD children.

Index	Run1	Run2	p-value
Certain Go Accuracy (%)	91.8	93.3	0.198
Uncertain Go Accuracy (%)	94.1	94.8	0.948
Stop Accuracy (%)	51.8	53.0	0.361
SSRT (ms)	285.0	289.6	0.441

RT slowing (ms)	21.8	18.8	0.671
Certain Go RT (ms)	479.3	487.8	0.241
Uncertain Go RT (ms)	501.1	506.5	0.466
ex-Gaussian modelling			
Certain Go RT (mu)	397.6	407.6	0.167
Certain Go RT (sigma)	47.2	47.9	0.903
Certain Go RT (tau)	81.6	80.2	0.856
Uncertain Go RT (mu)	419.7	425.9	0.365
Uncertain Go RT (sigma)	41.8	45.0	0.476
Uncertain Go RT (tau)	81.5	80.7	0.929

Table S11. Performance difference across sessions in ADHD children.

Index	Run1	Run2	p-value
Certain Go Accuracy (%)	92.9	92.8	0.946
Uncertain Go Accuracy (%)	93.3	93.4	0.908
Stop Accuracy (%)	53.6	54.3	0.523
SSRT (ms)	303.2	308.4	0.724
RT slowing (ms)	22.6	24.9	0.835
Certain Go RT (ms)	510.5	520.6	0.252
Uncertain Go RT (ms)	533.1	545.4	0.126
ex-Gaussian modelling			
Certain Go RT (mu)	428.7	416.4	0.125
Certain Go RT (sigma)	48.8	41.2	0.273
Certain Go RT (tau)	81.8	104.1	0.060
Uncertain Go RT (mu)	425.0	443.0	0.065
Uncertain Go RT (sigma)	42.8	41.0	0.839
Uncertain Go RT (tau)	107.9	102.4	0.626

3.12- It is not clear how precisely randomize was used for the different types of analyses. It seems, sometimes comparisons were run between groups, sometimes between conditions or tested against zero.

Response: We appreciate the reviewer's request for clarification on how FSL's randomise was used in our analyses.

In this study, one of our main goals was to examine whether children with ADHD exhibit greater temporal and spatial variability during task performance, which has been underexplored in previous research. For this, between-group comparisons (e.g., ADHD vs. TD) were conducted using FSL's randomise, where group labels were randomly swapped between participants during the permutation process. The observed data were then compared against the distribution of permuted datasets to generate group-level

statistics. This same logic was applied to all other between-group comparisons, including those examining proactive and reactive control.

Additionally, because we hypothesized that cognitive control is encoded by distributed neural activity patterns within the brain, within-group analyses were performed to contrast proactive and reactive control conditions for each group. For these analyses, FSL's randomise was used to test the contrast maps against zero, with sign flipping applied to the contrast maps to generate permuted datasets. The same permutation logic was applied to the neural coding analyses for proactive and reactive control within each group, as well as for group-specific neural coding of cognitive control processes. We have revised the Methods and Results section to clarify the use of randomise in these different analyses.

3.13- The description of ROI definition should be revised. The first sentence is not very clear.

- More importantly, ROI definition based on resting state data in a separate dataset seems suboptimal, if the authors cannot give a clearer rationale for this. ROI selection based on published meta-analytic results or at least task-based effects in previous studies seems more warranted.

Response: We appreciate the reviewer's feedback regarding the definition of our ROIs. Following the reviewer's suggestion, we redefined the ROIs based on a meta-analysis of previous fMRI studies on inhibitory control. Specifically, we created spherical ROIs including the preSMA and rAI in the salience network, as well as the rDLPFC and rPPC in the frontoparietal network. In re-running the analyses using these meta-analytic ROIs, we found highly similar results, supporting the robustness of our original findings. Additionally, we conducted further control analyses, comparing temporal and spatial stability between the groups while controlling for age, gender, and head motion. These analyses confirmed the stability of our results, further reinforcing the validity of our conclusions.

3.14- Is there a rationale for the 125 voxel searchlight size?

Response: The choice of a 125-voxel searchlight size is based on its widespread use in previous multivariate pattern analysis (MVPA) studies, it has been shown to be sensitive in detecting both the content and cognitive processes of interest (Dimsdale-Zucker & Ranganath, 2018; Gao et al., 2021; Gao et al., 2023; Gao et al., 2022; Viganò & Piazza, 2020; Xu et al., 2021; Zheng et al., 2018). This searchlight size strikes a balance between being sufficiently large to capture meaningful patterns of neural activity, while remaining small enough to maintain spatial specificity (Dimsdale-Zucker & Ranganath, 2018).

3.15- The authors seem to correct for multiple testing within each analysis (FDR corrected p values are reported). I am wondering if they also correct globally for the large amount of tests they performed overall on this dataset?

Response: We appreciate the reviewer's concern regarding multiple testing across the various analyses. In our study, FDR correction was applied within each individual analysis to control for Type I error rates associated with multiple comparisons within those specific contexts. Each analysis in our study was designed to test distinct, hypothesis-driven questions about different neural characteristics (e.g., temporal variability and spatial stability), which assess separate aspects of brain function. These analyses are theoretically independent of each other, as they measure different neural properties, and therefore do not constitute repeated testing of the same hypothesis. This approach allows for a focused interpretation of each neural measure without unduly penalizing the results across unrelated hypotheses. Therefore, we believe the FDR corrections applied within each analysis are sufficient and appropriately control for Type I errors, while maintaining sensitivity to detect meaningful group differences. While we are open to conducting further analysis for multiple comparison correlations if the reviewer thinks it is necessary.

3.16 Figure 4: The figure title and captions include ICP modulation of RT, I suggest to phrase this more carefully. Correlational analyses cannot speak to causality and we do not know if the recruitment of brain regions in a fashion that is similar to successful go trials actually modulates RT, or if both the spatial pattern in brain activation and the RT are modulated by another, underlying process or factor.

Response: We apologize for any confusion caused. While we modeled the go event's onset with a stick function which was before the motor execution to minimize the influence of RT on our single-trial estimation, we acknowledge that the sluggish nature of the BOLD signal means that correlation analyses cannot establish causality. In response to the reviewer's concern, we have rephrased "modulation" to "association" in the figure title and captions to avoid implying a causal relationship.

Reference

- Abdulrahman, H., & Henson, R. N. (2016). Effect of trial-to-trial variability on optimal event-related fMRI design: Implications for Beta-series correlation and multi-voxel pattern analysis. *Neuroimage*, 125, 756-766.
<https://doi.org/https://doi.org/10.1016/j.neuroimage.2015.11.009>
- Albajara Sáenz, A., Septier, M., Van Schuerbeek, P., Baijot, S., Deconinck, N., Defresne, P., Delvenne, V., Passeri, G., Raeymaekers, H., & Salvesen, L. (2020). ADHD and ASD: distinct brain patterns of inhibition-related activation? *Translational Psychiatry*, 10(1), 24.
- Arazi, A., Censor, N., & Dinstein, I. (2017). Neural variability quenching predicts individual perceptual abilities. *Journal of Neuroscience*, 37(1), 97-109.
- Bein, O., Duncan, K., & Davachi, L. (2020). Mnemonic prediction errors bias hippocampal states. *Nature Communications*, 11(1), 3451.
- Bergeron, D., Sellami, L., Poulin, S., Verret, L., Bouchard, R. W., & Laforce Jr, R. (2020). The behavioral/dysexecutive variant of Alzheimer's disease: a case series with clinical, neuropsychological, and FDG-PET characterization. *Dementia and geriatric cognitive disorders*, 49(5), 518-525.

- Blank, H., & Davis, M. H. (2016). Prediction errors but not sharpened signals simulate multivoxel fMRI patterns during speech perception. *PLoS biology*, *14*(11), e1002577.
- Cai, W., Chen, T., Ide, J. S., Li, C.-S. R., & Menon, V. (2017). Dissociable fronto-operculum-insula control signals for anticipation and detection of inhibitory sensory cue. *Cerebral cortex*, *27*(8), 4073-4082.
- Cai, W., Duberg, K., Padmanabhan, A., Rehert, R., Bradley, T., Carrion, V., & Menon, V. (2019). Hyperdirect insula-basal-ganglia pathway and adult-like maturity of global brain responses predict inhibitory control in children. *Nature Communications*, *10*(1), 4798. <https://doi.org/10.1038/s41467-019-12756-8>
- Cai, W., Warren, S. L., Duberg, K., Pennington, B., Hinshaw, S. P., & Menon, V. (2021). Latent brain state dynamics distinguish behavioral variability, impaired decision-making, and inattention. *Molecular psychiatry*, *26*(9), 4944-4957.
- Cai, W., Warren, S. L., Duberg, K., Yu, A., Hinshaw, S. P., & Menon, V. (2023). Both reactive and proactive control are deficient in children with ADHD and predictive of clinical symptoms. *Translational Psychiatry*, *13*(1), 179.
- Chen, J., Leong, Y. C., Honey, C. J., Yong, C. H., Norman, K. A., & Hasson, U. (2017). Shared memories reveal shared structure in neural activity across individuals. *Nature Neuroscience*, *20*(1), 115-125.
- Congdon, E., Mumford, J. A., Cohen, J. R., Galvan, A., Canli, T., & Poldrack, R. A. (2012). Measurement and reliability of response inhibition. *Frontiers in psychology*, *3*, 37.
- Cortese, S., Aoki, Y. Y., Itahashi, T., Castellanos, F. X., & Eickhoff, S. B. (2021). Systematic review and meta-analysis: resting-state functional magnetic resonance imaging studies of attention-deficit/hyperactivity disorder. *Journal of the American Academy of Child & Adolescent Psychiatry*, *60*(1), 61-75.
- Dennis-Tiwary, T. A., Roy, A. K., Denefrio, S., & Myruski, S. (2019). Heterogeneity of the anxiety-related attention bias: A review and working model for future research. *Clinical psychological science*, *7*(5), 879-899.
- Deuker, L., Bellmund, J. L., Navarro Schröder, T., & Doeller, C. F. (2016). An event map of memory space in the hippocampus. *elife*, *5*, e16534.
- Dimsdale-Zucker, H. R., & Ranganath, C. (2018). Representational similarity analyses: a practical guide for functional MRI applications. In *Handbook of behavioral neuroscience* (Vol. 28, pp. 509-525). Elsevier.
- Dinstein, I., Heeger, D. J., & Behrmann, M. (2015). Neural variability: friend or foe? *Trends in cognitive sciences*, *19*(6), 322-328.
- Duffy, K. A., Rosch, K. S., Nebel, M. B., Seymour, K. E., Lindquist, M. A., Pekar, J. J., Mostofsky, S. H., & Cohen, J. R. (2021). Increased integration between default mode and task-relevant networks in children with ADHD is associated with impaired response control. *Developmental cognitive neuroscience*, *50*, 100980.
- Fauth-Bühler, M., de Rover, M., Rubia, K., Garavan, H., Abbott, S., Clark, L., Vollstädt-Klein, S., Mann, K., Schumann, G., & Robbins, T. W. (2012). Brain networks subserving fixed versus performance-adjusted delay stop trials in a stop signal task. *Behavioural brain research*, *235*(1), 89-97.

- Favila, S. E., Samide, R., Sweigart, S. C., & Kuhl, B. A. (2018). Parietal representations of stimulus features are amplified during memory retrieval and flexibly aligned with top-down goals. *Journal of Neuroscience*, *38*(36), 7809-7821.
- Feng, G., Gan, Z., Llanos, F., Meng, D., Wang, S., Wong, P. C., & Chandrasekaran, B. (2021). A distributed dynamic brain network mediates linguistic tone representation and categorization. *Neuroimage*, *224*, 117410.
- Gao, Z., Zheng, L., Chiou, R., Gouws, A., Krieger-Redwood, K., Wang, X., Varga, D., Ralph, M. A. L., Smallwood, J., & Jefferies, E. (2021). Distinct and common neural coding of semantic and non-semantic control demands. *Neuroimage*, *236*, 118230.
- Gao, Z., Zheng, L., Gouws, A., Krieger-Redwood, K., Wang, X., Varga, D., Smallwood, J., & Jefferies, E. (2023). Context free and context-dependent conceptual representation in the brain. *Cerebral cortex*, *33*(1), 152-166.
- Gao, Z., Zheng, L., Krieger-Redwood, K., Halai, A., Margulies, D. S., Smallwood, J., & Jefferies, E. (2022). Flexing the principal gradient of the cerebral cortex to suit changing semantic task demands. *elife*, *11*, e80368.
- Grady, C. L., & Garrett, D. D. (2014). Understanding variability in the BOLD signal and why it matters for aging. *Brain imaging and behavior*, *8*, 274-283.
- He, B. J. (2013). Spontaneous and task-evoked brain activity negatively interact. *Journal of Neuroscience*, *33*(11), 4672-4682.
- He, B. J., & Zempel, J. M. (2013). Average is optimal: an inverted-U relationship between trial-to-trial brain activity and behavioral performance. *PLoS Computational Biology*, *9*(11), e1003348.
- Heather, N. (2014). Interpreting null findings from trials of alcohol brief interventions. *Frontiers in psychiatry*, *5*, 85.
- Hong, S.-B., & Hwang, S. (2022). Resting-state brain variability in youth with attention-deficit/hyperactivity disorder. *Frontiers in psychiatry*, *13*, 918700.
- Hu, H., Li, A., Zhang, L., Liu, C., Shi, L., Peng, X., Li, T., Zhou, Y., & Xue, G. (2024). Goal-directed attention transforms both working and long-term memory representations in the human parietal cortex. *PLoS biology*, *22*(7), e3002721.
- Hu, S., Ide, J. S., Zhang, S., & Chiang-shan, R. L. (2016). The right superior frontal gyrus and individual variation in proactive control of impulsive response. *Journal of Neuroscience*, *36*(50), 12688-12696.
- Ide, J. S., Shenoy, P., Angela, J. Y., & Chiang-Shan, R. L. (2013). Bayesian prediction and evaluation in the anterior cingulate cortex. *Journal of Neuroscience*, *33*(5), 2039-2047.
- Iravani, B., Arshamian, A., Fransson, P., & Kaboodvand, N. (2021). Whole-brain modelling of resting state fMRI differentiates ADHD subtypes and facilitates stratified neuro-stimulation therapy. *Neuroimage*, *231*, 117844.
- Kennedy, D. P., Paul, L. K., & Adolphs, R. (2015). Brain connectivity in autism: the significance of null findings. *Biological psychiatry*, *78*(2), 81-82.
- Logan, G. D., & Cowan, W. B. (1984). On the ability to inhibit thought and action: A theory of an act of control. *Psychological review*, *91*(3), 295.
- Messel, M. S., Raud, L., Hoff, P. K., Skaftnes, C. S., & Huster, R. J. (2019). Strategy switches in proactive inhibitory control and their association with task-general and stopping-specific networks. *Neuropsychologia*, *135*, 107220.

- Miller, J. A., Tambini, A., Kiyonaga, A., & D'Esposito, M. (2022). Long-term learning transforms prefrontal cortex representations during working memory. *Neuron*, *110*(22), 3805-3819. e3806.
- Mizuno, Y., Cai, W., Supekar, K., Makita, K., Takiguchi, S., Silk, T. J., Tomoda, A., & Menon, V. (2023). Methylphenidate enhances spontaneous fluctuations in reward and cognitive control networks in children with attention-deficit/hyperactivity disorder. *Biological Psychiatry: Cognitive Neuroscience and Neuroimaging*, *8*(3), 271-280.
- Mochalski, L. N., Friedrich, P., Li, X., Kröll, J. P., Eickhoff, S. B., & Weis, S. (2024). Inter - and intra - subject similarity in network functional connectivity across a full narrative movie. *Human brain mapping*, *45*(11), e26802.
- Müller, V. I., Cieslik, E. C., Serbanescu, I., Laird, A. R., Fox, P. T., & Eickhoff, S. B. (2017). Altered brain activity in unipolar depression revisited: meta-analyses of neuroimaging studies. *JAMA Psychiatry*, *74*(1), 47-55.
- Mumford, J. A., Davis, T., & Poldrack, R. A. (2014). The impact of study design on pattern estimation for single-trial multivariate pattern analysis. *Neuroimage*, *103*, 130-138.
- Mumford, J. A., Turner, B. O., Ashby, F. G., & Poldrack, R. A. (2012). Deconvolving BOLD activation in event-related designs for multivoxel pattern classification analyses. *Neuroimage*, *59*(3), 2636-2643.
- Musella, K. E., & Weyandt, L. L. (2023). Attention-deficit hyperactivity disorder and youth's emotion dysregulation: A systematic review of fMRI studies. *Applied Neuropsychology: Child*, *12*(4), 353-366.
- Nguyen, M., Vanderwal, T., & Hasson, U. (2019). Shared understanding of narratives is correlated with shared neural responses. *Neuroimage*, *184*, 161-170.
- Nomi, J. S., Bolt, T. S., Ezie, C. C., Uddin, L. Q., & Heller, A. S. (2017). Moment-to-moment BOLD signal variability reflects regional changes in neural flexibility across the lifespan. *Journal of Neuroscience*, *37*(22), 5539-5548.
- Nomi, J. S., Schettini, E., Voorhies, W., Bolt, T. S., Heller, A. S., & Uddin, L. Q. (2018). Resting-state brain signal variability in prefrontal cortex is associated with ADHD symptom severity in children. *Frontiers in human neuroscience*, *90*.
- Padmala, S., & Pessoa, L. (2010). Interactions between cognition and motivation during response inhibition. *Neuropsychologia*, *48*(2), 558-565.
- Prince, J. S., Charest, I., Kurzawski, J. W., Pyles, J. A., Tarr, M. J., & Kay, K. N. (2022). Improving the accuracy of single-trial fMRI response estimates using GLMsingle. *elife*, *11*, e77599. <https://doi.org/10.7554/eLife.77599>
- Rissman, J., & Wagner, A. D. (2012). Distributed Representations in Memory: Insights from Functional Brain Imaging. *Annual Review of Psychology*, *63*(1), 101-128. <https://doi.org/10.1146/annurev-psych-120710-100344>
- Samea, F., Soluki, S., Nejati, V., Zarei, M., Cortese, S., Eickhoff, S. B., Tahmasian, M., & Eickhoff, C. R. (2019). Brain alterations in children/adolescents with ADHD revisited: A neuroimaging meta-analysis of 96 structural and functional studies. *Neuroscience & Biobehavioral Reviews*, *100*, 1-8. <https://doi.org/https://doi.org/10.1016/j.neubiorev.2019.02.011>

- Sebastian, A., Pohl, M., Klöppel, S., Feige, B., Lange, T., Stahl, C., Voss, A., Klauer, K., Lieb, K., & Tüscher, O. (2013). Disentangling common and specific neural subprocesses of response inhibition. *Neuroimage*, *64*, 601-615.
- Sebastian, A., Rössler, K., Wibrals, M., Mobascher, A., Lieb, K., Jung, P., & Tüscher, O. (2017). Neural architecture of selective stopping strategies: distinct brain activity patterns are associated with attentional capture but not with outright stopping. *Journal of Neuroscience*, *37*(40), 9785-9794.
- Shenoy, P., & Yu, A. J. (2011). Rational decision-making in inhibitory control. *Frontiers Human Neuroscience*, *5*, 48.
- Smittenaar, P., Guitart-Masip, M., Lutti, A., & Dolan, R. J. (2013). Preparing for selective inhibition within frontostriatal loops. *Journal of Neuroscience*, *33*(46), 18087-18097.
- Suskauer, S. J., Simmonds, D. J., Caffo, B. S., Denckla, M. B., Pekar, J. J., & Mostofsky, S. H. (2008). fMRI of intrasubject variability in ADHD: anomalous premotor activity with prefrontal compensation. *Journal of the American Academy of Child & Adolescent Psychiatry*, *47*(10), 1141-1150.
- Swann, N. C., Cai, W., Conner, C. R., Pieters, T. A., Claffey, M. P., George, J. S., Aron, A. R., & Tandon, N. (2012). Roles for the pre-supplementary motor area and the right inferior frontal gyrus in stopping action: Electrophysiological responses and functional and structural connectivity. *Neuroimage*, *59*(3), 2860-2870.
<https://doi.org/https://doi.org/10.1016/j.neuroimage.2011.09.049>
- van Belle, J., Vink, M., Durston, S., & Zandbelt, B. B. (2014). Common and unique neural networks for proactive and reactive response inhibition revealed by independent component analysis of functional MRI data. *Neuroimage*, *103*, 65-74. <https://doi.org/https://doi.org/10.1016/j.neuroimage.2014.09.014>
- van Hulst, B. M., de Zeeuw, P., Rijks, Y., Neggers, S. F., & Durston, S. (2017). What to expect and when to expect it: an fMRI study of expectancy in children with ADHD symptoms. *European child & adolescent psychiatry*, *26*, 583-590.
- van Rooij, S. J., Rademaker, A. R., Kennis, M., Vink, M., Kahn, R. S., & Geuze, E. (2014). Impaired right inferior frontal gyrus response to contextual cues in male veterans with PTSD during response inhibition. *Journal of Psychiatry and Neuroscience*, *39*(5), 330-338.
- Verbruggen, F., Aron, A. R., Band, G. P., Beste, C., Bissett, P. G., Brockett, A. T., Brown, J. W., Chamberlain, S. R., Chambers, C. D., & Colonius, H. (2019). A consensus guide to capturing the ability to inhibit actions and impulsive behaviors in the stop-signal task. *elife*, *8*, e46323.
- Viganò, S., & Piazza, M. (2020). Distance and direction codes underlie navigation of a novel semantic space in the human brain. *Journal of Neuroscience*, *40*(13), 2727-2736.
- Wehrheim, M. H., Faskowitz, J., Schubert, A. L., & Fiebach, C. J. (2024). *Reliability of variability and complexity measures for task and task - free BOLD fMRI* (1065-9471).
- White, C. N., Congdon, E., Mumford, J. A., Karlsgodt, K. H., Sabb, F. W., Freimer, N. B., London, E. D., Cannon, T. D., Bilder, R. M., & Poldrack, R. A. (2014). Decomposing decision components in the stop-signal task: a model-based

- approach to individual differences in inhibitory control. *Journal of Cognitive Neuroscience*, 26(8), 1601-1614.
- Winkler, A. M., Ridgway, G. R., Webster, M. A., Smith, S. M., & Nichols, T. E. (2014). Permutation inference for the general linear model. *Neuroimage*, 92, 381-397.
- Xu, S., Li, Y., & Liu, J. (2021). The neural correlates of computational thinking: Collaboration of distinct cognitive components revealed by fMRI. *Cerebral cortex*, 31(12), 5579-5597.
- Xue, G., Dong, Q., Chen, C., Lu, Z., Mumford, J. A., & Poldrack, R. A. (2010). Greater neural pattern similarity across repetitions is associated with better memory. *Science*, 330(6000), 97-101.
- Yap, K. H., Manan, H. A., & Sharip, S. (2021). Heterogeneity in brain functional changes of cognitive processing in ADHD across age: a systematic review of task-based fMRI studies. *Behavioural brain research*, 397, 112888.
- Zeithamova, D., de Araujo Sanchez, M.-A., & Adke, A. (2017). Trial timing and pattern-information analyses of fMRI data. *Neuroimage*, 153, 221-231. <https://doi.org/https://doi.org/10.1016/j.neuroimage.2017.04.025>
- Zheng, L., Gao, Z., Doner, S., Oyao, A., Forloines, M., Grilli, M. D., Barnes, C. A., & Ekstrom, A. D. (2023). Hippocampal contributions to novel spatial learning are both age-related and age-invariant. *Proceedings of the National Academy of Sciences*, 120(50), e2307884120.
- Zheng, L., Gao, Z., McAvan, A. S., Isham, E. A., & Ekstrom, A. D. (2021). Partially overlapping spatial environments trigger reinstatement in hippocampus and schema representations in prefrontal cortex. *Nature Communications*, 12(1), 6231. <https://doi.org/10.1038/s41467-021-26560-w>
- Zheng, L., Gao, Z., Xiao, X., Ye, Z., Chen, C., & Xue, G. (2018). Reduced fidelity of neural representation underlies episodic memory decline in normal aging. *Cerebral cortex*, 28(7), 2283-2296.
- Zhou, Y., Curtis, C. E., Sreenivasan, K. K., & Fougny, D. (2022). Common neural mechanisms control attention and working memory. *Journal of Neuroscience*, 42(37), 7110-7120.

REVIEWERS' COMMENTS

Reviewer #3 (Remarks to the Author):

Overall, the authors have addressed my concerns thoroughly and thoughtfully. Only few minor points remain.

Response: We thank the reviewer for their careful review and helpful comments on our manuscript. We appreciate their recognition of our efforts to address their previous concerns.

General:

- Please check language again. For instance, “significant” instead of “significantly” is used in various instances across the manuscript.

Response: We have carefully reviewed the manuscript and corrected instances of incorrect word usage.

Results:

- Line 218ff: if the authors made any changes to their statement that children with ADHD have poor reactive control and less stable performance but preserved proactive control, this is not tracked/highlighted. I still think the statement is too bold as it is currently, given that the findings were only significant in the full but not the fmri sample.

Response: We have revised the text to better acknowledge the limited robustness of the fMRI sample: “The behavioral results from the full sample indicate that children with ADHD exhibit poorer reactive control and less stable performance while maintaining proactive control compared to their TD peers. While a similar trend was observed in the fMRI sample, these effects were less robust, likely due to the reduce sample size. This reduction primarily resulted from excessive head motion, which was particularly prevalent among children diagnosed with ADHD.” (Page 6)

- In terms of structure, adding the ROI analyses directly after the respective whole brain level analyses might be helpful and more in line with the figures.

Response: We appreciate the reviewer's suggestion. In our initial submission, we structured the whole-brain and ROI results similarly as the reviewer suggested here. However, in the last revision, Reviewer #2 recommended us adapting the illustration to

the current format. This organizational structure was chosen to better emphasize the key role of the cognitive control system played in the proactive and reactive control, and to highlight their differences between children with and without ADHD. Furthermore, to maintain a degree of coherence, the ROI analyses focusing on temporal and spatial stability are still presented directly following their corresponding whole-brain analyses. We hope that this approach balances a strong emphasis on the cognitive control system with a clear and navigable presentation for our readers. We are happy to provide additional clarification or make further adjustments if needed.

- Line 371 ff: "Children with ADHD showed less distributed effect in the supplementary motor area and anterior cingulate cortex in Certain and Uncertain Go trial" please clarify that this is by visual inspection/comparison, not statistical comparison.

Response: We thank the reviewer for highlighting the need for clarification and apologize for any confusion caused by our previous wording. To clarify, the observation on Line 371 pertains solely to the ICP-RT association effect in children with ADHD. While visual inspection suggested that the spatial extent of effect in the supplementary motor area and anterior cingulate cortex appeared less distributed in children with ADHD compared to TD children during Certain and Uncertain Go trials, we did not conduct a statistical comparison at the voxel level for each participant. Such an analysis would require applying a multiple comparison correction that necessarily differs from the TFCE approach used in our group-level analysis, and therefore, this observation cannot be directly integrated into our whole-brain statistical results. Importantly, this does not impact our main findings or our conclusion that children with ADHD exhibit a weaker association between ICP and RT in the cognitive control system compared to TD children. As this conclusion is further supported by both whole-brain searchlight and ROI-based analyses, which consistently demonstrated a significantly stronger association in TD children.

To clarify this point, we have revised the text as follows: "Children with ADHD exhibited effect in the supplementary motor area and anterior cingulate cortex during Certain and Uncertain Go trials (TFCE, $p < 0.05$ corrected), though spatial extent of the effect appears to be smaller in children with ADHD than TD children." (Page 9)

Reviewer #4 (Remarks to the Author):

I co-reviewed this manuscript with one of the reviewers who provided the listed reports. This is part of the Nature Communications initiative to facilitate training in peer review

and to provide appropriate recognition for Early Career Researchers who co-review manuscripts.

For this rebuttal review, as my senior co-reviewer declined to participate in the review process, I have discussed with the editor and was advised to provide my independent assessment of the revised manuscript. I have reviewed the revised manuscript and the authors' responses to the previous comments.

Overall, I appreciate the authors' extensive efforts in addressing the previous concerns and improving the manuscript. The proposed metrics based on single-trial BOLD signals (temporal variability and spatial stability) offer novel perspectives for understanding neural dynamics in ADHD and similar disorders, showing promising potential for future research. The additional control analyses have enhanced the study's rigor and reliability. While the manuscript has been substantially improved, there are several remaining issues that require attention:

(1) There is a notable disconnect between the network-level framework emphasized in the Introduction and Discussion (focusing on SN and FPN), and the region-based reporting in the Results section. This inconsistency may pose challenges for readers less familiar with these networks. A clearer mapping between individual regions and their corresponding networks would enhance the manuscript's accessibility and coherence.

Response: We agree that a clear mapping between individual regions and their corresponding networks can enhance the manuscript's accessibility. To address this, we have revised the text to explicitly link the hypothesized regions with their associated networks, as follows: "We hypothesized that children with ADHD would show increased temporal variability and reduced spatial stability of trial-evoked responses in the AI and preSMA, which are the key nodes of the SN, as well as dlPFC and PPC, which are the key nodes of the FPN." (Please see page 5). Additionally, to better illustrate the correspondence between ROIs and their respective networks, we modified Figure 1h by using distinct colors to differentiate ROIs from different networks.

(2) Several aspects of figure presentation require improvement. While Figures 2-5 now include colorbars as requested, they lack proper labeling of the metrics being displayed. The correspondence between whole brain results and ROI-based results remains difficult to interpret due to inconsistent scaling between whole brain colorbars and ROI-based y-axes, and unclear mapping of ROI locations to whole-brain results. Additionally,

the inconsistent use of colormaps across figures lacks clear justification, and Figure 6 needs proper legend explaining the color-coding of dots. These presentation issues significantly impact the interpretability of the results.

Response: The whole-brain results presented in Figures 2–5 represent group-level statistics produced by FSL's randomise with TFCE correction, and the colorbar indicates $1-p$, which is a standard approach in neuroimaging analysis. We apologize for not including the colorbar labels in our previous submission. In contrast, the ROI-based results display values for temporal variability and spatial stability, derived from our analyses. We have now added appropriate labeling to the colorbars of the whole-brain results to clarify the metrics being displayed. Additionally, we have updated Figure 6 (now Figure 7) to include a legend explaining the color coding of dots. We hope these revisions improve the accessibility of our figures and address the reviewer's concerns.

(3) While the Discussion section has been revised, it still contains substantial overlap with the Results and Methods sections, and has actually increased in length. This makes it more challenging to follow the logical flow of arguments. The discussion of within-group spatial similarity and its relationship to heterogeneity in ADHD symptoms and cognitive deficits requires more careful consideration, as these interpretations currently lack direct supporting evidence from the results.

Response: We thank the reviewer for their careful consideration of the relationship between within-group spatial similarity and ADHD symptoms/cognitive deficits. Because no previous studies have examined the heterogeneity of distributed neural coding during cognitive control, we first performed a within-group inter-subject pattern similarity analysis. We acknowledge that directly relating within-group neural measures to clinical symptoms across both cohorts would be highly informative. However, such cross-group comparisons necessitate a common representational reference, which did not exist in this analysis. To address this, we implemented an ADHD-like neural pattern as a shared anchor, enabling robust symptom correlations while maintaining methodological rigor. This approach also aligns with recent advances in dimensional psychiatry and strengthens the clinical relevance of our findings (Cuthbert, 2014; Morris et al., 2022; Yee et al., 2015).

Regarding the overall Discussion, we appreciate the reviewer's suggestions for streamlining and clarification. However, based on the consensus of the other reviewers, we feel that the current integrated format provides a clear and comprehensive narrative.

We have made minor clarifications for enhanced clarity while retaining the structure that we consider essential for fully conveying our findings and their implications.

(4) The specification of event onsets in the fMRI GLM analysis, while included in the response letter, should be explicitly detailed in the main text. The current description is too brief for replication purposes. More detailed information about event onset definitions, timing parameters, and relevant processing steps would enhance the reproducibility of the study.

Response: We thank the reviewer for this valuable suggestion. In response, we have added detailed information about event onset definitions, timing parameters, and relevant processing steps to the main text to enhance the reproducibility of the study. “ Each trial began with a white or green cross (Cue) in the center of the screen for 300 milliseconds, followed by a green arrow (go signal). Participants were demanded to make an accurate and speedy button press in response to the pointing direction of the green arrow within 1.5s after the onset of the Go stimuli. Occasionally, the green arrow turned to red (stop signal), and participants had to withhold their responses when the color changed. The color of the cue indicated the probability of the stop signal in the coming trial. A green cross indicates that no stop signal would occur after the coming go signal, which is defined as Certain Go trial. A white cross indicates that a stop signal may occur after the coming go signal (33% chance), which is defined as Uncertain Go trial. If a stop signal is presented, the trial is defined as Stop trial. The stop-signal delay (SSD) was initialized at 200ms and adjusted in a staircase fashion. If the participant successfully canceled a prepotent response, the SSD increased by 50ms in the next Stop trial. If the participant failed in stopping, the SSD decreased by 50ms in the next Stop trial. Participants completed two runs of the CSST in the scanner, and each run included 32 Certain Go trials, 32 Uncertain Go trials, and 16 Stop trials with jittered inter-trial intervals (ITIs) between 1 and 4 seconds.” (Page 18)

“Image pre-processing and statistical analysis were performed using SPM12 (<https://www.fil.ion.ucl.ac.uk/spm/software/spm12>). The first 12 volumes before the task were discarded to allow for T1 equilibrium. The remaining images were then realigned to correct for head movements and underwent slice-timing correction. EPI images were registered to the MNI standard space. Data were spatially smoothed using a 2 mm FWHM Gaussian kernel as recent studies suggested that minimal spatial smoothing could enhance signal-to-noise ratio while preserving distributed pattern information (Dimsdale-Zucker & Ranganath, 2018).” (Page 19)

“Conventional GLM was conducted to estimate trial averaged neural responses elicited by task conditions in the CSST, including Certain Go, Uncertain Go, Successful Stop (SuccStop), Unsuccessful Stop (UnsuccStop) and Go error. Trial onset was locked to the Go stimulus. Six motion parameters were entered as covariates of no interest.” (Page 19)

Reviewer #5 (Remarks to the Author):

This is a novel and well executed study examining proactive and reactive control in children with ADHD compared to neurotypical children. The authors utilised a novel computational method to explore neural temporal and spatial stability/instability in ADHD of trial-evoked activity in a stop signal reaction time task. Via this technique authors could explore temporal stability by examining measures such as standard deviation and kurtosis in neural activity and spatial stability using representational similarity analysis (RSA). Overall this study provides new exciting insights into cognitive control in ADHD. My questions regarding the analysis and meaning of the variables such as the standard deviation and kurtosis were addressed by responses to previous reviewers. I have a number of minor remaining questions but overall the manuscript is an important, well executed one.

Response: We sincerely thank Reviewer for their thoughtful and supportive evaluation of our manuscript. We are especially grateful for your recognition of the novelty of our computational approach and the study’s contribution to understanding proactive and reactive control in ADHD. We are pleased that our responses to the previous reviewers have adequately clarified your questions regarding key variables such as standard deviation and kurtosis.

I am not sure how accurate it is to state that inhibitory dysregulation is a core feature of ADHD – this is highlighted by the relatively old references provided to support that statement. In fact, many children with ADHD have no issues with inhibitory control.

Response: We respectfully acknowledge the reviewer’s perspective and agree that inhibitory control deficits are not universally present across all children with ADHD. However, a substantial body of evidence, including recent studies (Cai et al., 2023; Mirabella, 2021; Nejati et al., 2021), supports the view that inhibitory dysregulation is one of the core feature of the disorders.

In response to the reviewer's comment, we have updated the statement and included updated references. We hope these revisions can address the reviewer's concern.

"Inhibitory dysregulation, the impaired ability to suppress context-inappropriate responses, is hypothesized to be one of the core mechanisms underlying these behavioral phenotypes (Barkley, 1997; Cai et al., 2023; Lipszyc & Schachar, 2010; Willcutt et al., 2005)." (Page 3).

It is somewhat misleading to state that participants were actually age and gender matched when the case is that there were no significant differences in groups between age and gender.

Response: We thank the reviewer for pointing this out. We have revised the text to clarify that there were no significant differences between the groups in age and gender, rather than stating they were explicitly matched.

*"There were no significant differences in age or gender between the two groups (all $p > 0.2$, two-sample t-test, **Table 1**)." (Page 5)*

What was the criterion for ADHD diagnosis? Were clinical interviews conducted and what cut-off was used for the Conners'?

Response: Clinical diagnosis of ADHD was informed by participants' parents or legal guardians and further confirmed by clinical questionnaire administered by clinical assessors under supervision of a clinical psychologist. We have added these information in the Methods. (Page 18).

Reference

- Barkley, R. A. (1997). Behavioral inhibition, sustained attention, and executive functions: constructing a unifying theory of ADHD. *Psychological bulletin*, 121(1), 65.
- Cai, W., Warren, S. L., Duberg, K., Yu, A., Hinshaw, S. P., & Menon, V. (2023). Both reactive and proactive control are deficient in children with ADHD and predictive of clinical symptoms. *Translational Psychiatry*, 13(1), 179.
- Cuthbert, B. N. (2014). The RDoC framework: facilitating transition from ICD/DSM to dimensional approaches that integrate neuroscience and psychopathology. *World Psychiatry*, 13(1), 28-35.

- Dimsdale-Zucker, H. R., & Ranganath, C. (2018). Representational similarity analyses: a practical guide for functional MRI applications. In *Handbook of behavioral neuroscience* (Vol. 28, pp. 509-525). Elsevier.
- Lipszyc, J., & Schachar, R. (2010). Inhibitory control and psychopathology: a meta-analysis of studies using the stop signal task. *Journal of the International Neuropsychological Society*, 16(6), 1064-1076.
- Mirabella, G. (2021). Inhibitory control and impulsive responses in neurodevelopmental disorders. *Developmental Medicine & Child Neurology*, 63(5), 520-526.
- Morris, S. E., Sanislow, C. A., Pacheco, J., Vaidyanathan, U., Gordon, J. A., & Cuthbert, B. N. (2022). Revisiting the seven pillars of RDoC. *BMC medicine*, 20(1), 220.
- Nejati, V., Alavi, M. M., & Nitsche, M. A. (2021). The impact of attention deficit-hyperactivity disorder symptom severity on the effectiveness of transcranial direct current stimulation (tDCS) on inhibitory control. *Neuroscience*, 466, 248-257.
- Willcutt, E. G., Doyle, A. E., Nigg, J. T., Faraone, S. V., & Pennington, B. F. (2005). Validity of the executive function theory of attention-deficit/hyperactivity disorder: a meta-analytic review. *Biological psychiatry*, 57(11), 1336-1346.
- Yee, C. M., Javitt, D. C., & Miller, G. A. (2015). Replacing DSM categorical analyses with dimensional analyses in psychiatry research: the research domain criteria initiative. *JAMA Psychiatry*, 72(12), 1159-1160.

co-reviewed this manuscript with one of the reviewers who provided the listed reports. This is part of the Nature Communications initiative to facilitate training in peer review and to provide appropriate recognition for Early Career Researchers who co-review manuscripts. For this rebuttal review, as my senior co-reviewer declined to participate in the review process, I have discussed with the editor and was advised to provide my independent assessment of the revised manuscript.

I have reviewed the revised manuscript and the authors' responses to the previous comments. Overall, I appreciate the authors' extensive efforts in addressing the previous concerns and improving the manuscript. The proposed metrics based on single-trial BOLD signals (temporal variability and spatial stability) offer novel perspectives for understanding neural dynamics in ADHD and similar disorders, showing promising potential for future research. The additional control analyses have enhanced the study's rigor and reliability.

While the manuscript has been substantially improved, there are several remaining issues that require attention:

(1) There is a notable disconnect between the network-level framework emphasized in the Introduction and Discussion (focusing on SN and FPN), and the region-based reporting in the Results section. This inconsistency may pose challenges for readers less familiar with these networks. A clearer mapping between individual regions and their corresponding networks would enhance the manuscript's accessibility and coherence.

(2) Several aspects of figure presentation require improvement. While Figures 2-5 now include colorbars as requested, they lack proper labeling of the metrics being displayed. The correspondence between whole-brain results and ROI-based results remains difficult to interpret due to inconsistent scaling between whole-brain colorbars and ROI-based y-axes, and unclear mapping of ROI locations to whole-brain results. Additionally, the inconsistent use of colormaps across figures lacks clear justification, and Figure 6 needs proper legend explaining the color-coding of dots. These presentation issues significantly impact the interpretability of the results.

(3) While the Discussion section has been revised, it still contains substantial overlap with the Results and Methods sections, and has actually increased in length. This makes it more challenging to follow the logical flow of arguments. The discussion of within-group spatial similarity and its relationship to heterogeneity in ADHD symptoms and cognitive deficits requires more careful consideration, as these interpretations currently lack direct supporting evidence from the results.

(4) The specification of event onsets in the fMRI GLM analysis, while included in the response letter, should be explicitly detailed in the main text. The current description is too brief for replication purposes. More detailed information about event onset definitions, timing parameters, and relevant processing steps would enhance the reproducibility of the study.

These modifications would further strengthen the manuscript and improve its utility for the research community. Despite these concerns, I believe this study makes a valuable contribution to our understanding of neural dynamics in ADHD, particularly through its innovative methodological approach.